# INPP5K and Atlastin-1 maintain the nonuniform distribution of ER–plasma membrane contacts in neurons

Jingbo Sun[1],*, Raihanah Harion[1],*, Tomoki Naito[1], Yasunori Saheki[1,2]

In neurons, the ER extends throughout all cellular processes, forming multiple contacts with the plasma membrane (PM) to fine-tune neuronal physiology. However, the mechanisms that regulate the distribution of neuronal ER-PM contacts are not known. Here, we used the *Caenorhabditis elegans* DA9 motor neuron as our model system and found that neuronal ER-PM contacts are enriched in soma and dendrite and mostly absent in axons. Using forward genetic screen, we identified that the inositol 5-phosphatase, CIL-1 (human INPP5K), and the dynamin-like GTPase, ATLN-1 (human Atlastin-1), help to maintain the non-uniform, somatodendritic enrichment of neuronal ER-PM contacts. Mechanistically, CIL-1 acts upstream of ATLN-1 to maintain the balance between ER tubules and sheets. In mutants of CIL-1 or ATLN-1, ER sheets expand and invade into the axon. This is accompanied by the ectopic formation of axonal ER-PM contacts and defects in axon regeneration following laser-induced axotomy. As INPP5K and Atlastin-1 have been linked to neurological disorders, the unique distribution of neuronal ER-PM contacts maintained by these proteins may support neuronal resilience during the onset and progression of these diseases.

## Introduction

The ER is the largest membranous organelle, comprising a single continuous network of interconnected tubules and cisternae that contain multiple domains with different functions (Shibata et al, 2009; Westrate et al, 2015). Extending from the nuclear envelope, the peripheral ER consists of flat cisternae sheets and reticulated tubules that include two functionally segregated domains: rough and smooth ER (Shibata et al, 2006). Rough ER, characterized by ribosome-rich areas, is a major site for the synthesis of secreted and integral membrane proteins and is found primarily around the nucleus as stacked sheets connected by twisted membrane surfaces (Terasaki et al, 2013). By contrast, ribosome-free smooth ER is a major site for the synthesis of membrane lipids and for $Ca^{2+}$ storage (Schwarz & Blower, 2016). In addition, the ER extends throughout the cell, interacting with all other cellular organelles and membranes without membrane fusion via membrane contact sites to control cell physiology, including $Ca^{2+}$ homeostasis, organelle dynamics, lipid exchange, and cell signaling (Phillips & Voeltz, 2016; Valm et al, 2017; Wu et al, 2018).

In neurons, a continuous network of ER tubules and cisternae can be found throughout the neuronal soma, axon, and dendrites (Tsukita & Ishikawa, 1976; Terasaki et al, 1994; Wu et al, 2017; Farias et al, 2019). Whereas soma and dendrites contain both rough and smooth ER, axonal ER comprises a network of largely smooth tubular ER with a very small diameter (~30 nm) (ER tubules typically have a diameter of ~60 nm in most cell types) (Wu et al, 2017; Yalcin et al, 2017; Terasaki, 2018). At the cell periphery, the ER is in close contact (within 10–30 nm) with the plasma membrane (PM) (Orci et al, 2009; Friedman & Voeltz, 2011; West et al, 2011; Fernandez-Busnadiego et al, 2015; Bayer et al, 2017; Wu et al, 2017; Collado et al, 2019; Hoffmann et al, 2019). Growing evidence suggests that ER-PM contacts contribute to general cell physiology, as well as to the unique functional properties of neurons, including the control of neuronal excitability via local regulation of ion channel function and the facilitation of non-vesicular lipid transport between the ER and PM (Moriguchi et al, 2006; Kakizawa et al, 2007; de Juan-Sanz et al, 2017; Saheki & De Camilli, 2017a; Johnson et al, 2018; Chen et al, 2019; Kirmiz et al, 2019; Sun et al, 2019; Vierra et al, 2019; Stefan, 2020). Thus, neuronal ER-PM contacts likely play important roles in neuronal survival and functional homeostasis. However, the molecular mechanisms by which the distribution of neuronal ER-PM contacts is maintained are currently unknown.

ER-PM contacts are mediated by various tethering proteins, including the evolutionarily conserved ER-resident proteins, the extended-synaptotagmins (E-Syts). In mammals, these are E-Syt1, E-Syt2, and E-Syt3; they are called tricalbins in yeast (Manford et al, 2012; Toulmay & Prinz, 2012; Saheki, 2017; Saheki & De Camilli, 2017a, 2017b; Collado et al, 2019; Hoffmann et al, 2019). E-Syts are anchored to ER membranes through their N-terminal hydrophobic stretch and form homo- and hetero-meric complexes. They tether the ER to the PM via interactions between their cytosolic C2 domains and phosphatidylinositol 4,5-bisphosphate [PI(4,5)P_2] within the PM

[1]Lee Kong Chian School of Medicine, Nanyang Technological University, Singapore   [2]Department of Molecular Physiology, Faculty of Life Sciences, Kumamoto University, Kumamoto, Japan

Correspondence: yasunori.saheki@ntu.edu.sg
*Jingbo Sun and Raihanah Harion are co-first authors

(Giordano et al, 2013; Fernandez-Busnadiego et al, 2015). E-Syts additionally possess a lipid-harboring synaptotagmin-like mitochondrial-lipid-binding (SMP) domain and mediate transport/exchange of glycerolipids between the ER and PM (Schauder et al, 2014; Saheki et al, 2016; Jeyasimman & Saheki, 2019).

In contrast, the unique structure of the ER is maintained by a number of different proteins that function to control ER shape (i.e., ER-shaping proteins). Each ER-shaping protein participates in unique steps during the formation and maintenance of the peripheral ER network, including maintenance of ER tubules via curvature stabilization, connection of ER tubules via facilitation of homotypic ER membrane fusion, and maintenance of three-way junctions that result from the fusion of ER tubules (Shibata et al, 2009; Westrate et al, 2015). Many of ER-shaping proteins are evolutionarily conserved, and studies from various model organisms, including yeast, mammalian cells, flies, and worms, have demonstrated that functional dysregulation of ER-shaping proteins, or imbalances in their activities, results in abnormal ER structures, and in many cases, alters the abundance of cortical ER (i.e., the ER that mediates ER-PM contacts). In yeast, depletion of Reticulons/RTNs and/or REEPs/DP1/Yop1p family members, which are evolutionarily conserved ER-shaping proteins involved in curvature stabilization/generation of ER tubules, leads to loss of ER tubules, expansion of ER sheets, and an increase in ER-PM contacts (De Craene et al, 2006; Voeltz et al, 2006; Hu et al, 2008; West et al, 2011). In *Drosophila* neurons, these proteins are required for maintaining the tubular structure of the ER, including axonal ER (O'Sullivan et al, 2012; Yalcin et al, 2017; Espadas et al, 2019). On the other hand, depletion of Atlastins/ATLs, which are dynamin-like GTPases that facilitate homotypic fusion of ER tubules, results in unbranched ER tubules, loss of three-way junctions, and fragmentation of the ER in mammalian cells, flies, and worms (Hu et al, 2009; Orso et al, 2009; Summerville et al, 2016; Wang et al, 2016; Liu et al, 2019). In yeast, depletion of Sey1p (the dynamin-like GTPase that resembles the Atlastin) has no effect, but simultaneous depletion of Sey1p with either Yop1p or Rtn1p results in loss of ER tubules and expansion of cortical ER sheets, suggesting the potential importance of Atlastin family proteins in regulating ER tubule formation as well as cortical ER abundance (Hu et al, 2009; Anwar et al, 2012). A recent study demonstrated that INPP5K, a metazoan specific inositol 5-phosphatase that can be targeted to the ER, is required for the maintenance of ER tubules in HeLa cells as well as in *Caenorhabditis elegans* PVD neurons (Dong et al, 2018). In the absence of INPP5K, ER sheets dramatically expand, although the precise mechanism of this process is still unclear (Dong et al, 2018). Notably, mutations in humans that impair the phosphatase activity of INPP5K lead to congenital muscular dystrophy with additional clinical manifestations, including intellectual impairments, suggesting that INPP5K plays an important role in the nervous system to maintain proper function of neurons (Osborn et al, 2017; Wiessner et al, 2017). Mutations in Atlastin-1, RTN2, or REEP1, also result in neurological problems; deleterious mutations in these proteins lead to hereditary spastic paraplegia, which is characterized by progressive loss of axons associated with motor neurons (Blackstone, 2012). Thus, proteins that are involved in the maintenance of proper ER shape are important for neuronal function and survival in humans. However, the relationship between these ER-shaping proteins and the distribution and function of neuronal ER-PM contacts has remained elusive.

In this study, we explored the molecular mechanisms controlling the distribution of neuronal ER-PM contacts and found that ER-shaping proteins play a critical role in this process. Using the *C. elegans* cholinergic DA9 motor neuron as a model system, we visualized the distribution of neuronal ER-PM contacts in vivo by fluorescence microscopy. In DA9, ER-PM contacts contain ESYT-2, the sole E-Syt homolog in *C. elegans*, and localize predominantly to somatodendritic regions, with very few ER-PM contacts present in the dorsal axon. We performed an unbiased forward genetic screen to identify mutants that mislocalized ER-PM contacts to the dorsal axon, and isolated mutations in *atln-1* and *cil-1*, which encode homologs of mammalian Atlastin-1 and INPP5K, respectively. Our genetic and cell biological analysis revealed that CIL-1 acts upstream of ATLN-1 to maintain the balance between tubules and sheets at the cortical ER as well as to restrict the ER sheets to somatodendrites. In the absence of these proteins, cortical ER sheets expand and rough ER proteins localize ectopically to axonal ER. Supporting the importance of maintaining the balance between ER tubules and sheets, mutants that lack RET-1 (the sole reticulon homolog in *C. elegans*), whose depletion is known to result in ER sheet expansion, phenocopy the ectopic formation of ER-PM contacts observed in *atln-1* and *cil-1* mutants. Furthermore, we performed laser axotomy experiments in *C. elegans*, and found that regeneration of the DA9 axon was significantly reduced in both *cil-1* and *atln-1* mutants compared with wild type. These results suggest that the non-uniform distribution of neuronal ER-PM contacts that is maintained by CIL-1 and ATLN-1 is critical for the function of these contacts, potentially contributing to the resilience of neurons against neuronal damage.

# Results

### ER-PM contacts are non-uniformly distributed in the *C. elegans* DA9 neuron

To visualize neuronal ER-PM contacts in live neurons in vivo, we chose *C. elegans* as a model system because of its highly stereotypic cell lineage, well-defined neuroanatomy, and transparent nature. We focused our analysis on the DA9 cholinergic motor neuron, which localizes to the tail of the worm. DA9 has a simple cytoarchitecture. Its cell body resides in the ventral side of the tail; a dendrite extends anteriorly along the ventral nerve cord; and an axon extends into the dorsal nerve cord forming "en passant" presynaptic nerve terminals (Fig 1A) (Klassen & Shen, 2007; Saheki & Bargmann, 2009).

To label ER-PM contacts, a split GFP approach was used (Kamiyama et al, 2016; Romei & Boxer, 2019) (Fig 1B and C). Previous studies have demonstrated that membrane contact sites can be labelled and visualized using this approach (Cieri et al, 2018; Kakimoto et al, 2018; Shai et al, 2018). We further extended this system to label ER-PM contacts in *C. elegans* neurons. We generated transgenic worms that co-expressed two components of split GFP under the DA9-selective *itr-1pB* promoter and performed imaging analysis using spinning disc confocal (SDC) microscopy or SDC structural illumination microscopy (SDC-SIM). One component of

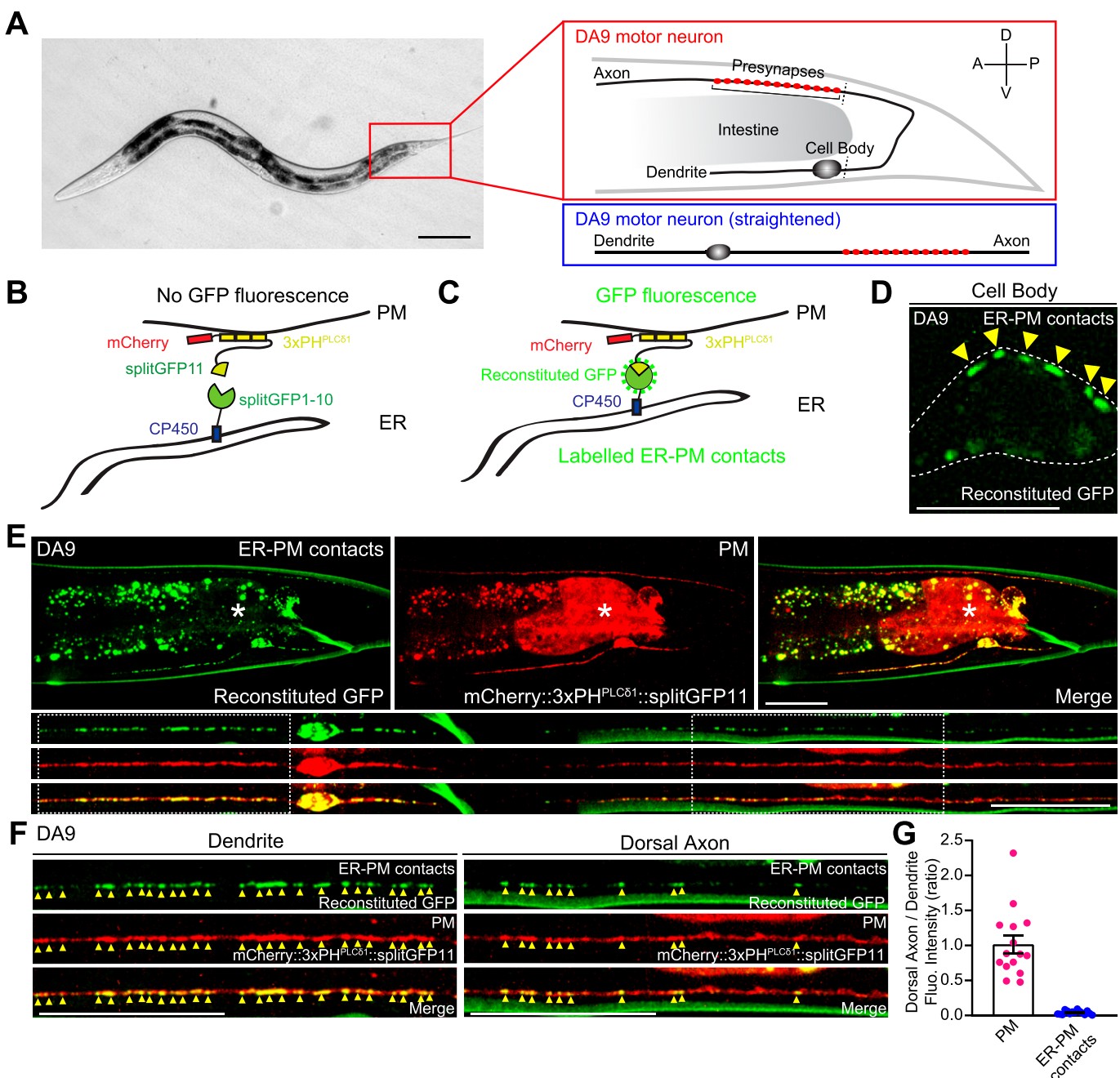

**Figure 1. ER-PM contacts are non-uniformly distributed in the *C. elegans* DA9 neuron.**
**(A)** Left: A hermaphrodite worm at L4 stage. Right: A schematic of the anatomy of DA9 motor neuron, located at the tail (top), and a schematic of straightened DA9 (bottom). Presynaptic nerve terminals present in the dorsal axon are indicated by red circles. Scale bar, 100 $\mu$m. **(B, C)** Schematics of the split GFP strategy for labelling the contacts between the ER and the plasma membrane (PM) in vivo. One component of split GFP (splitGFP11) was tagged to mCherry-fused with tandem pleckstrin homology (PH) domains of phospholipase C (PLC) $\delta$1 (mCherry::3xPH$^{PLC\delta1}$::splitGFP11) that bind to PI(4,5)P$_2$ on the PM, and the other split GFP component (splitGFP1-10) was tagged to the cytosolic region of ER resident protein CP450 (CP450::splitGFP1-10). Close proximity of the two split GFP components results in GFP reconstitution, labelling ER-PM contact sites as green puncta. **(D)** A representative live spinning disc confocal structural illumination microscopy (SDC-SIM) image of the cell body of a DA9 neuron, co-expressing both CP450::splitGFP1-10 and mCherry::3xPH$^{PLC\delta1}$::splitGFP11 under DA9 specific *itr-1pB* promoter. Reconstituted GFP marks ER-PM contacts at the cell periphery (yellow arrowheads). A single focal plane of the center of the cell body is shown. White dashed line outlines the shape of the DA9 cell body. Scale bar, 5 $\mu$m. **(E)** Representative live SDC images of a DA9 neuron showing ER-PM contacts (green: reconstituted GFP) and the PM (red: mCherry::3xPH$^{PLC\delta1}$::splitGFP11) labelled by the split GFP approach. Bottom rows show straightened images of the same DA9 neuron. Dendrite and dorsal axon, as indicated by white dashed boxes, are enlarged in (F). The central autofluorescent region is the intestine (asterisk). Scale bars, 20 $\mu$m. **(F)** ER-PM contacts that are marked by the split GFP approach (yellow arrowheads) are non-uniformly distributed in DA9. Note their enrichment in dendrite compared to dorsal axon. Scale bars, 20 $\mu$m. **(G)** Quantification of the ratio of fluorescence intensity of dorsal axon to that of dendrite of DA9 neuron. mCherry::3xPH$^{PLC\delta1}$::splitGFP11 signals were used for the PM; reconstituted GFP signals were used for ER-PM contacts (mean ± SEM, n = 15 for both PM and ER-PM contacts).
Source data are available for this figure.

the split GFP (fragments 1–10 of GFP) was fused to the cytosolic region of the ER membrane protein, CP450. The other GFP fragment (fragment 11) was fused to an mCherry-tagged PM binding module [3 × pleckstrin homology (PH) domains derived from phospholipase C (PLC) $\delta$1, which bind to PM PI(4,5)P$_2$] (mCherry::3xPH$^{PLC\delta1}$:: splitGFP11) (Fig 1B and C). Close apposition of these two fragments resulted in the assembly of fluorescent GFP as distinct puncta (observed by SDC-SIM) at the periphery of the DA9 cell body (Fig 1D). Thus, we were able to successfully label neuronal ER-PM contacts. Notably, reconstituted GFP signals labelling ER-PM contacts showed a non-uniform distribution; they were more abundant in the soma, dendrite, and proximal/ventral axon (all within the ventral region of the animal) compared with the circumferential axon and distal/dorsal axon. In contrast, mCherry-tagged PH$^{PLC\delta1}$ signal, corresponding to the PM, was present throughout the DA9 neuron (Fig 1E–G). Replacing mCherry::3xPH$^{PLC\delta1}$::splitGFP11 with cytosolic mCherry-tagged split GFP (fragment 11) (mCherry::splitGFP11) resulted in uniform distribution of GFP fluorescence throughout all neuronal processes of the DA9 neuron, demonstrating that CP450 is distributed throughout neuronal processes as a general ER marker (thus ruling out the biased enrichment of CP450 in the dendrite or ventral axon) (Fig S1A–D). These results suggest that ER-PM contacts are non-uniformly distributed in the *C. elegans* DA9 neuron.

## ESYT-2 localizes to neuronal ER-PM contacts

The split GFP approach mostly labels ER-PM contacts that depend on the presence of PM PI(4,5)P$_2$ [as mCherry::3xPH$^{PLC\delta1}$::splitGFP11 recognizes PM PI(4,5)P$_2$]. In addition, this approach may also force the formation of ectopic ER-PM contacts that may not reflect endogenous distribution of these contacts. ER-PM contacts are mediated by a number of ER-resident tethering proteins, including the evolutionarily conserved family of E-Syt proteins (Saheki & De Camilli, 2017a, 2017b). In *C. elegans*, ESYT-2 is the sole ortholog of the E-Syt proteins and is structurally similar to mammalian E-Syt2 and E-Syt3, which localize constitutively to ER-PM contacts (Giordano et al, 2013). However, the subcellular distribution of ESYT-2 in neurons has not yet been studied. To further confirm the non-uniform distribution of ER-PM contacts that we observed with the split GFP approach in DA9 neuron, we visualized the endogenous localization of ESYT-2 using a cell-type specific endogenous labelling approach (Schwartz & Jorgensen, 2016). In this approach, "FLP-on GFP" was fused to the N terminus of ESYT-2 by CRISPR/ Cas9-based gene editing. We visualized endogenous ESYT-2 (endoGFP::ESYT-2) specifically in the DA9 neuron by expressing FLP via the DA9-selective *itr-1pB* promoter. Puncta of endoGFP:: ESYT-2 were found primarily in the soma, dendrite, and ventral axon of DA9 (Fig 2A and B). Quantification of the fluorescence intensity showed that endoGFP::ESYT-2 puncta were much more abundant in the dendrite compared to the dorsal axon, similar to the distribution of ER-PM contacts observed using the split GFP approach (Fig 2A–C, compare with Fig 1E–G). A few endoGFP::ESYT-2 puncta were also present in the dorsal axon. To determine if they localized to presynaptic regions of the DA9 axon, a synaptic vesicle marker [mCherry fused to RAB-3 (mCherry::RAB-3)] was co-expressed in DA9. Co-localization of endoGFP::ESYT-2 and mCherry::RAB-3 was assessed by line scan analysis. This revealed that endoGFP::ESYT-2

puncta in the dorsal axon localized outside of the presynaptic regions labelled by mCherry::RAB-3 (Fig 2B). The endogenous localization of ESYT-2 in DA9 was further examined at different developmental stages (larval L2, L3, and L4 stages) (Fig S2A–C). We did not detect endoGFP::ESYT-2 at the larval L2 stage, most likely due to very low expression of ESYT-2 at this stage of development (Fig S2A). At the larval L3 stage, endoGFP::ESYT-2 puncta were present in the ventral DA9 process (Fig S2B), and the number of the puncta increased as animals grew to L4 larva (Fig S2C). Few endoGFP::ESYT-2 puncta were detected in the dorsal axon regardless of the development stages of the animal. Thus, the non-uniform distribution of endoGFP::ESYT-2 in the DA9 neuron is established at the larval L3 stage and maintained through development to L4 and adult stages. In the AWC olfactory sensory neurons, which reside in the head region, endoGFP::ESYT-2 also localized predominantly to the dendrite, with very little signal within presynaptic regions, indicating that the non-uniform distribution of ESYT-2 is a general feature of *C. elegans* neurons (Fig S2D and E).

To gain further insights into the localization of ESYT-2, endoGFP:: ESYT-2 puncta within the dendrite and axon were individually examined by tracking their movement along the processes via kymograph analysis. ESYT-2 puncta in the dendrite were immobile over the 2-min imaging period. By contrast, ESYT-2 puncta in the dorsal axon were highly mobile and dynamic. After 2 min, most of the ESYT-2 puncta in the dorsal axon had moved away from their time 0 position (Fig 2D). These results suggest that dendritic ESYT-2 localizes to cortical ER and stably associates with ER-PM contacts, whereas the mobile ESYT-2 possibly indicates minor fractions of ESYT-2 that may participate in membrane contact sites formed between the ER and other organelles. To examine whether dendritic ESYT-2 localizes to ER-PM contacts, wrmScarlet tagged ESYT-2 (wrmScarlet::ESYT-2) and split GFP components (to label ER-PM contacts) were co-expressed in the DA9 neuron and their association was assessed by line-scan analysis. Although wrmScarlet:: ESYT-2 did not co-localize perfectly with GFP puncta (i.e., split GFP-labelled ER-PM contact sites), the majority of wrmScarlet::ESYT-2 co-localized with GFP puncta, confirming that dendritic ESYT-2 localizes primarily to ER-PM contacts (Fig 2E and F). These results are consistent with the heterogeneity of the ER-PM contacts as shown in recent studies (Saheki & De Camilli, 2017a; Besprozvannaya et al, 2018).

Collectively, these results demonstrate that ER-PM contacts in DA9 neurons are highly enriched in the somatodendritic region and that these contacts contain an evolutionarily conserved ER-PM tethering protein, ESYT-2.

## ESYT-2 distribution is altered in *cil-1* mutants

To investigate the underlying molecular mechanism regulating the non-uniform distribution of neuronal ER-PM contacts, we performed an unbiased, forward genetic screen using a transgenic worm that expressed mNeonGreen-tagged ESYT-2 (mNeonGreen:: ESYT-2) together with a synaptic vesicle marker (mCherry::RAB-3) under the control of the DA9-selective *itr-1pB* promoter. This strain enhanced our ability to detect ESYT-2 to efficiently visualize ER-PM contacts in the DA9 neuron. mNeonGreen::ESYT-2 localized predominantly to the somatodendritic region (zone A) and the ventral

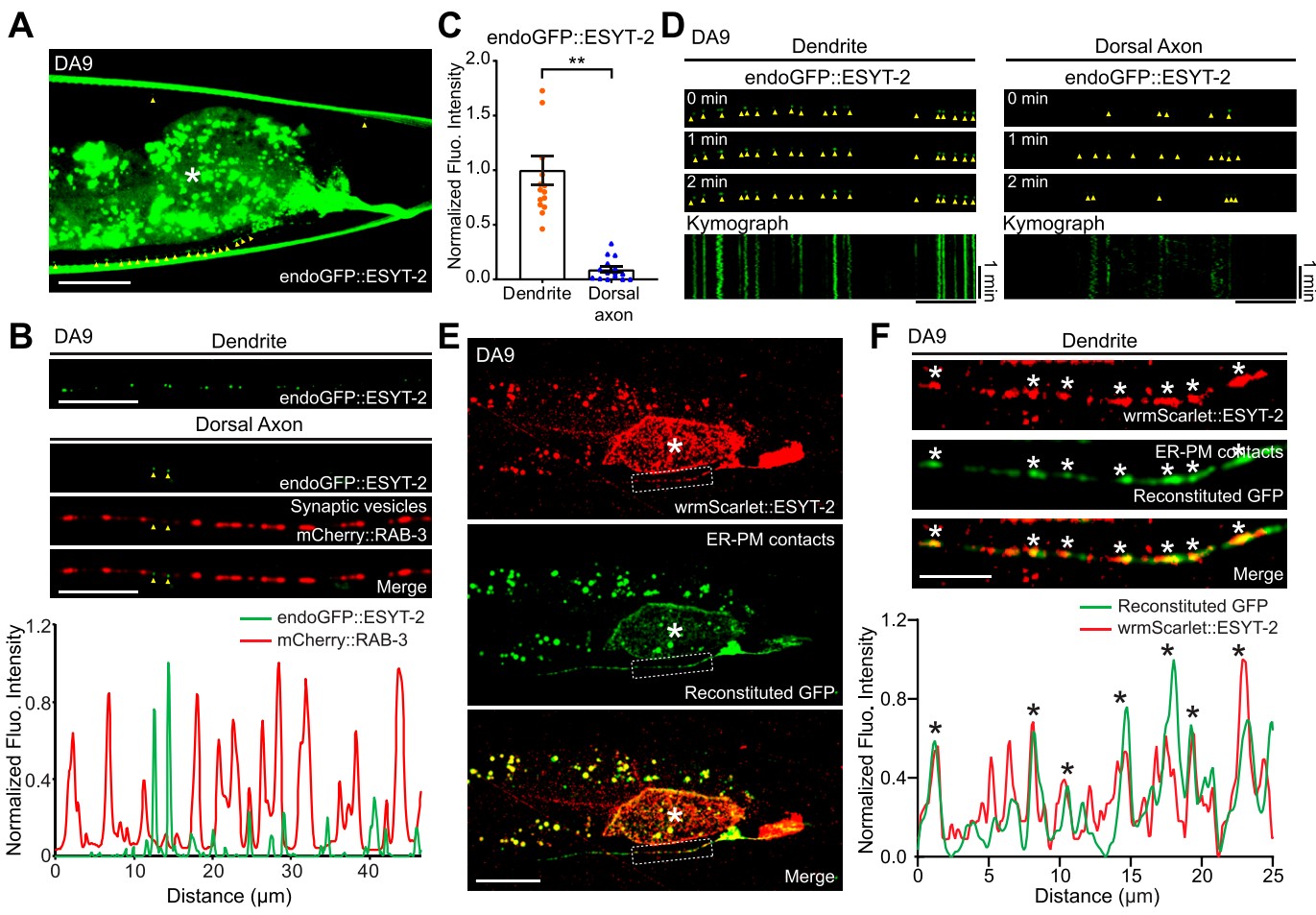

**Figure 2. ESYT-2 localizes to neuronal ER-PM contacts.**
**(A)** A representative live spinning disc confocal (SDC) image of a DA9 neuron, carrying endogenously tagged ESYT-2 (endoGFP::ESYT-2), showing enrichment of ESYT-2 in dendrite compared to dorsal axon. Yellow arrowheads indicate endoGFP::ESYT-2 puncta. The central autofluorescent region is the intestine (asterisk). Scale bar, 20 μm. **(B)** Representative live SDC images of dendrite and dorsal axon of a DA9 neuron, co-expressing endoGFP::ESYT-2 and synaptic vesicle marker [mCherry-tagged with RAB-3 (mCherry::RAB-3)]. Yellow arrowheads indicate endoGFP::ESYT-2 puncta in dorsal axon. The line scan analysis of fluorescent intensity of endoGFP::ESYT-2 (green) and mCherry::RAB-3 (red) along the dorsal axon is shown below. Scale bars, 10 μm. **(C)** Quantification of normalized fluorescence intensity of endoGFP::ESYT-2 signals in dendrites and dorsal axons of DA9 neurons (mean ± SEM; n = 14 for dendrite and n = 15 for dorsal axon. Two-tailed unpaired $t$ test **$P < 0.0001$). **(D)** Kymographs showing the dynamics of endoGFP::ESYT-2 in dendrite (left) and dorsal axon (right) of a DA9 neuron. Images captured at different time points (0, 1, or 2 min) are shown as representative images. Yellow arrowheads indicate endoGFP::ESYT-2 puncta. Note the disappearance of some endoGFP::ESYT-2 puncta in dorsal axon during the 2 min imaging period. Scale bars, 10 μm and 1 min. **(E)** Representative live SDC images of a DA9 neuron, co-expressing wrmScarlet-tagged ESYT-2 (wrmScarlet::ESYT-2) and the components for labelling ER-PM contacts (3xPH$^{PLCδ1}$::splitGFP11 and CP450::splitGFP1-10). Note the extensive overlap of wrmScarlet::ESYT-2 and reconstituted GFP (ER-PM contacts). Dendrite of the DA9 neuron, as indicated by a white dashed box, is enlarged in (F). The central autofluorescent region is the intestine (asterisk). Scale bars, 20 μm. **(F)** Extensive overlap of wrmScarlet::ESYT-2 and reconstituted GFP (ER-PM contacts) in dendrite. The line scan analysis of normalized fluorescent intensity of reconstituted GFP (green) and wrmScarlet::ESYT-2 (red) along the dendrite is shown below. White and black asterisks mark the overlap. Scale bar, 5 μm.
Source data are available for this figure.

axon, but was generally excluded from the circumferential axon (zone B) and the dorsal axon (zone C), similar to the distribution of endoEGFP::ESYT-2 (Fig 3A).

From ~1,000 mutated haploid genomes, we isolated two mutants, *yas37* and *yas38*, that exhibited increased levels of mNeonGreen::ESYT-2 in the circumferential and dorsal axon (zone B and zone C), but normal RAB-3 puncta. *yas37* was sterile, whereas *yas38* produced a reduced brood size compared with controls (Fig S3A–E). Using balancer mapping and whole genome sequencing, we identified that *yas37* carries a causative mutation in the *cil-1* gene, which is the ortholog of mammalian INPP5K, and *yas38* carries a causative mutation in the *atln-1* gene, which is the ortholog of mammalian Atlastin-1.

In *cil-1(yas37)* mutants, mNeonGreen::ESYT-2 was uniformly distributed throughout the entire axon (Fig 3B). Furthermore, the dendritic ER, which was assessed by either mNeonGreen::ESYT-2 (Fig 3B and G) or a general ER marker CP450 (Fig S5A), was shortened compared with wild-type control. The fluorescence intensity of mNeonGreen::ESYT-2 was measured for each zone, and the ratio of fluorescence intensity in zone C to that in zone A was compared between control animals and *cil-1(yas37)* mutants (Fig 3C–F). In control animals, the ratio was ~0.1, indicating ~10-fold enrichment of mNeonGreen::ESYT-2 in the dendrite compared to the dorsal axon covered by presynaptic regions. This ratio was significantly increased in *cil-1(yas37)* mutants (~1), indicating the increased presence of

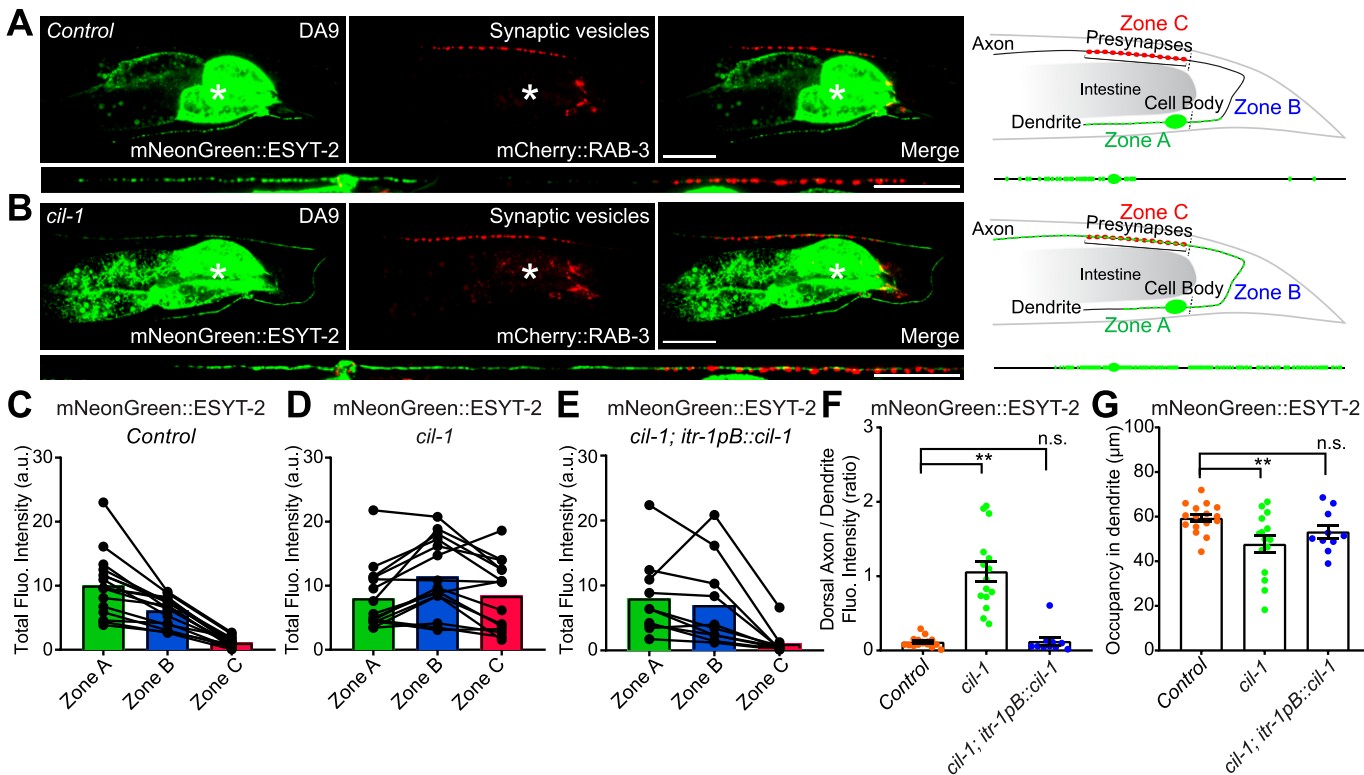

**Figure 3. ESYT-2 distribution is altered in *cil-1(yas37)* mutants.**
**(A, B)** Left: Representative live spinning disc confocal images of a DA9 neuron, co-expressing mNeonGreen-tagged ESYT-2 (mNeonGreen::ESYT-2) and mCherry::RAB-3 in wild-type control (A) and *cil-1 (yas37)* mutants (B). Straightened images of the DA9 neurons are shown below. Right: Schematics showing the distribution of mNeonGreen::ESYT-2 (marked as small green circles) in DA9 neuron. Zone A denotes dendrite; Zone B denotes ventral axon and circumferential axon up to the most proximal region of presynaptic nerve terminals labelled by mCherry::RAB-3 (flanked by the two black dashed lines); Zone C denotes the region of dorsal axon that is covered by presynaptic nerve terminals. Note the altered distribution of mNeonGreen::ESYT-2 in *cil-1 (yas37)* mutants. The central autofluorescent region is the intestine (asterisk). Scale bars, 20 μm. **(C, D, E)** Quantification of total fluorescence intensity of mNeonGreen::ESYT-2 puncta present in zones A, B, and C in wild-type control (n = 16) (C), *cil-1 (yas37)* mutants (n = 15) (D), and *cil-1 (yas37)* mutants expressing CIL-1 under DA9 specific *itr-1pB* promoter (n = 10) (E). Each connected dot represents one animal. **(F)** Quantification of the ratio of mNeonGreen::ESYT-2 fluorescence intensity of dorsal axon (zone C) to that of dendrite (zone A) from the same animals as shown in (C, D, E). (mean ± SEM; Dunnett's multiple comparisons test **P < 0.0001 [*Control* versus *cil-1 (yas37)*], n.s. denotes not significant). **(G)** Quantification of the occupancy of mNeonGreen::ESYT-2 signals in dendrites (zone A) from the same animals as shown in (C, D, E). Note the rescue of the phenotype by DA9 specific re-expression of CIL-1 in *cil-1(yas37)* mutants in (E, F, G). (mean ± SEM; Dunnett's multiple comparisons test; **P = 0.0080 [*Control* versus *cil-1 (yas37)*], n.s. denotes not significant). Source data are available for this figure.

mNeonGreen::ESYT-2 in the distal axon. Importantly, the mCherry::RAB-3 signal was not affected in *cil-1(yas37)* mutants (Fig S3F), indicating that the mechanisms controlling the distribution of ESYT-2 is different from those that regulate the formation and/or assembly of presynaptic terminals. In addition, DA9-specific expression of *cil-1* under the DA9-selective *itr-1pB* promoter restored the distribution of mNeonGreen::ESYT-2 (Fig 3E–G). These results suggest that *cil-1* acted cell-autonomously in the DA9 neuron to maintain the distribution of ESYT-2.

**CIL-1 regulates the cortical ER network and maintains the distribution of ER-PM contacts in neurons**

*cil-1* encodes an evolutionarily conserved inositol 5-phosphatase that acts primarily on phosphoinositide PI(4,5)P$_2$ (INPP5K in human). It contains a 5-phosphatase domain followed by a SKICH domain (Bae et al, 2009; Dong et al, 2018) (Fig 4A). Sequencing *cil-1(yas37)* DNA revealed a G to A mutation, which results in an early stop codon in the

CIL-1 open reading frame. *cil-1* mutants are recessive, suggesting that *cil-1(yas37)* is a loss-of-function allele of CIL-1 (Fig 4A).

To examine whether 5-phosphatase activity and/or the SKICH domain are necessary for CIL-1 function, we tested mutant versions of CIL-1 proteins for their abilities to rescue mNeonGreen::ESYT-2 localization in DA9. A missense mutation in a critical residue of the 5-phosphatase domain (N175A) eliminated its activity, failing to rescue the mNeonGreen::ESYT-2 mislocalization in *cil-1(yas37)* mutant (Figs 4A and C and S4A and C). Deletion of the SKICH domain also inactivated CIL-1 (Figs 4A and C and S4A and C). These results suggest that both the 5-phosphatase activity and the SKICH domain are essential for CIL-1 function. Another *cil-1* mutant, *cil-1(my15)*, causes a premature stop in the CIL-1 open reading frame, resulting in the deletion of the SKICH domain (Bae et al, 2009). *cil-1(my15)* mutants displayed uniform distribution of mNeonGreen::ESYT-2 along the dorsal axon and dendrite, similar to that seen in *cil-1(yas37)* null mutants, highlighting the importance of the SKICH domain for CIL-1 function (Figs 4A and C and S4B and C).

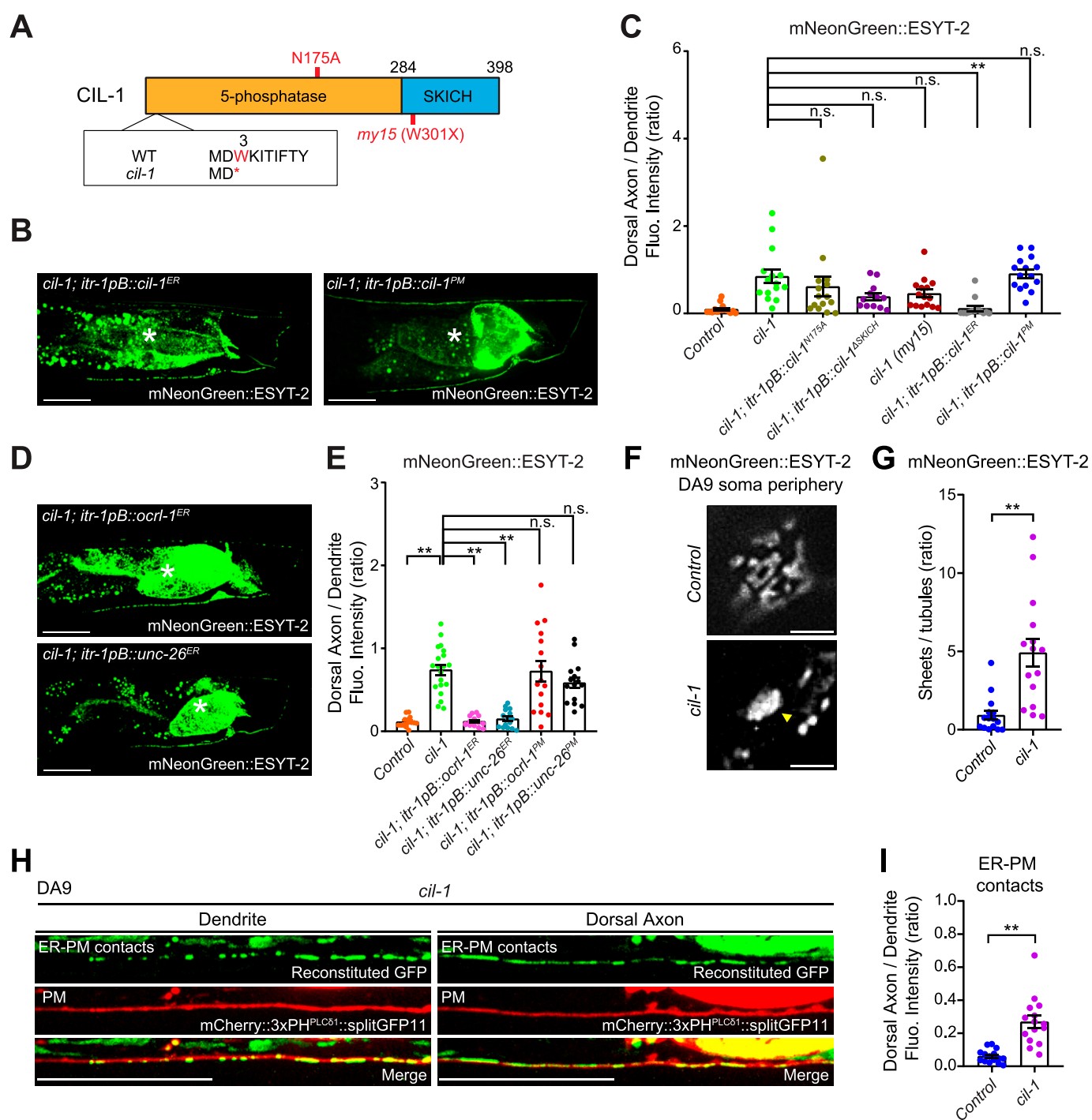

**Figure 4. CIL-1 regulates the cortical ER network and maintains the distribution of ER-PM contacts in neurons.**
**(A)** Domain structure of CIL-1 in *C. elegans*. N175A phosphatase-dead and early termination mutations (W3X [indicated by an asterisk], corresponding to *cil-1 (yas37)* mutation, and W301X, corresponding to *my15*) are indicated. **(B)** Representative live spinning disc confocal (SDC) images of DA9 neurons, expressing mNeonGreen::ESYT-2, from *cil-1 (yas37)* mutants that additionally express either ER-targeted CIL-1 (left) or PM-targeted CIL-1 (right) under DA9 specific *itr-1pB* promoter. Note that expression of ER-targeted CIL-1 restored the distribution of mNeonGreen::ESYT-2 in *cil-1 (yas37)* mutants. Scale bars, 20 μm. **(C)** Quantification of the ratio of mNeonGreen::ESYT-2 fluorescence intensity of dorsal axon (zone C) to that of dendrite (zone A) in wild-type control (n = 15), *cil-1 (yas37)* (n = 15), *cil-1 (yas37); itr-1pB::cil-1^N175A* (n = 15), *cil-1 (yas37); itr-1pB::cil-1^ΔSKICH* (n = 12), *cil-1 (my15)* (n = 15), *cil-1 (yas37); itr-1pB::cil-1^ER* (n = 13), and *cil-1 (yas37); itr-1pB::cil-1^PM* (n = 15) (mean ± SEM; Dunnett's multiple comparisons test; **P = 0.0006 [*cil-1 (yas37)* versus *cil-1 (yas37); itr-1pB::cil-1^ER*], n.s. denotes not significant). **(D)** Representative live SDC images of DA9 neurons, expressing mNeonGreen::ESYT-2, from *cil-1 (yas37)* mutants that additionally express ER-targeted phosphatase domains of either OCRL-1 or UNC-26 under DA9 specific *itr-1pB* promoter. Scale bars, 20 μm. **(E)** Quantification of the ratio of mNeonGreen::ESYT-2 fluorescence intensity of dorsal axon (zone C) to that of dendrite (zone A) in wild-type control (n = 15), *cil-1 (yas37)* (n = 20), *cil-1 (yas37); itr-1pB::ocrl-1^ER* (n = 15), *cil-1 (yas37); itr-1pB::unc-26^ER* (n = 15), *cil-1 (yas37); itr-1pB::ocrl-1^PM* (n = 16), and *cil-1 (yas37); itr-1pB::unc-26^PM* (n = 15) (mean ± SEM; Dunnett's multiple comparisons test; **P < 0.0001 *cil-1 (yas37)* versus *Control*, *cil-1 (yas37)* versus *cil-1 (yas37); itr-1pB::ocrl-1^ER*, and *cil-1 (yas37)* versus *cil-1 (yas37);*

The SKICH domain is required to recruit INPP5K/CIL-1 to various cellular compartments, including the ER, the PM, and the nucleus (Gurung et al, 2003; Dong et al, 2018). To determine the site at which CIL-1 functions to maintain the distribution of ESYT-2 in DA9, the 5-phosphatase domain of CIL-1 was selectively targeted to the ER or to the PM by fusing it with either the ER-resident CP450 protein or the PM-targeting motif of Lck. DA9-specific expression of the ER-targeted CIL-1 5-phosphatase domain restored the distribution of mNeonGreen::ESYT-2 in *cil-1(yas37)* mutants, whereas the PM-targeted CIL-1 5-phosphatase had no effect (Fig 4B and C). These results suggest that CIL-1 may function at the ER to modulate PI(4,5)$P_2$ levels of this organelle to maintain non-uniform distribution of ESYT-2 in neurons (see the Discussion section), although the possibility that PM PI(4,5)$P_2$ plays some roles in regulating ESYT-2 distribution in neurons cannot be excluded.

To further investigate the role of PI(4,5)$P_2$ in the distribution of ESYT-2 in neurons, we targeted 5-phosphatase domains from other PI(4,5)$P_2$ phosphatases, namely, OCRL-1 and UNC-26 (orthologues of human OCRL and synaptojanin-1, respectively), to the ER or to the PM by the same strategy that we used for targeting CIL-1 5-phosphatase domain to these cellular compartments. HeLa cells were used to confirm the activities of these chimeric 5-phosphatase domains (Fig S4D–G). Expression of either PM-targeted OCRL-1 5-phosphatase domain (mScarlet-I-OCRL-1$^{PM}$) or PM-targeted UNC-26 5-phosphatase domain (mScarlet-I-UNC-26$^{PM}$) strongly reduced the levels of PM PI(4,5)$P_2$, as assessed by iRFP-tagged with the PH domain of PLC$^{δ1}$ (iRFP-PH$^{PLCδ1}$) (Fig S4D and E), demonstrating that the 5-phosphatase domains of OCRL-1 and UNC-26 efficiently dephosphorylate PI(4,5)$P_2$ in cells. In contrast, expression of ER-targeted versions of the same 5-phosphatase domains (OCRL-1$^{ER}$-EGFP and UNC-26$^{ER}$-EGFP) had not effects on the levels of PM PI(4,5)$P_2$ (Fig S4F and G). These results suggest that the chimeric 5-phosphatase domains primarily act in cis without affecting PI(4,5)$P_2$ levels of other cellular compartments.

Finally, we expressed these chimeric 5-phosphatase domains specifically in DA9 neuron and examined whether forced reduction of PI(4,5)$P_2$ levels in the PM or ER membranes restore the non-uniform distribution of mNeonGreen::ESYT-2 in *cil-1(yas37)* mutants. Targeting the 5-phosphatase domain of either UNC-26 or OCRL-1 to the ER but not to the PM restored the distribution of mNeonGreen::ESYT-2 to the similar levels to control animals (Fig 4D and E). These results suggest the importance of PI(4,5)$P_2$ levels in ER membranes for maintaining the non-uniform distribution of ESYT-2.

Notably, CIL-1/INPP5K function is critical for maintaining the balance between ER tubules and sheets (Dong et al, 2018). Depletion of INPP5K in HeLa cells leads to the reduction of ER tubules and expansion of ER sheets, whereas loss of CIL-1 activity in *C. elegans* disrupts the formation of ER tubules and impairs their

extension into dendrites of the PVD neuron (Dong et al, 2018). However, the role of CIL-1/INPP5K in maintaining the structure of cortical ER (i.e., the ER that is engaged in ER-PM contacts) remains unknown. To examine if CIL-1 is required for regulating cortical ER structure in neurons, we imaged mNeonGreen::ESYT-2 under SDC-SIM at the periphery of the DA9 soma as an estimate of the structure of cortical ER in DA9. In control animals, cortical ER, as assessed by mNeonGreen::ESYT-2, formed network of sheets and tubules (Fig 4F) (see the Materials and Methods section for quantification details). In *cil-1(yas37)* mutants, however, cortical ER was visibly less complex, with fewer tubules, consisting primarily of larger sheets that lacked fenestration (Fig 4F and G).

These results indicated that the abnormal expansion of cortical ER sheets in DA9 may have contributed to the ectopic localization of ESYT-2 to dorsal axons in *cil-1(yas37)* mutants. To confirm whether ER-PM contacts are indeed mislocalized to dorsal axons, we used the split GFP approach in DA9 neuron of *cil-1(yas37)* mutants. Remarkably, ER-PM contacts were abundant in the dorsal axon of DA9 in *cil-1(yas37)* mutants (Fig 4H and I). In addition, the reconstituted GFP signal was largely diffuse, indicating that the ectopically formed ER-PM contacts were mediated by expanded cortical ER sheets. Collectively, these results suggest that ER shape may play a role in maintaining the distribution of neuronal ER-PM contacts.

## CIL-1 functions upstream of ATLN-1 to maintain the distribution of ER-PM contacts in neurons

The other mutant from the forward genetic screen, *atln-1(yas38)*, showed defects in mNeonGreen::ESYT-2 distribution with normal mCherry::RAB-3 puncta that resembled those of *cil-1(yas37)* mutants (Figs 5A and B and S3F). In *atln-1(yas38)* mutants, mNeonGreen::ESYT-2 was present uniformly throughout the entire axon (Fig 5A and B). In addition, the dendritic ER, as assessed by either mNeonGreen::ESYT-2 or a general ER marker CP450, was significantly shorter than in wild-type controls or *cil-1(yas37)* mutants (Figs 5A, E, and G and S5A). Expressing wild-type ATLN-1 via the DA9-selective *itr-1pB* promoter restored the mNeonGreen::ESYT-2 localization pattern in DA9, suggesting that *atln-1* acts cell autonomously, as seen with *cil-1* (Fig S5B–D, G, and H). ER-PM contacts, labelled by the split GFP approach, were also ectopically present in the dorsal axon in *atln-1(yas38)* mutants, further supporting that ATLN-1 and CIL-1 mediate similar functions (Fig 5C and D).

Sequencing the *atln-1(yas38)* mutant revealed a G to A mutation, which results in an amino acid substitution (E338K) in the ATLN-1 open reading frame (Fig S5B). *atln-1* mutants are recessive, suggesting that *atln-1(yas38)* is a possible reduction-of-function allele of ATLN-1. *atln-1* encodes a member of the evolutionarily conserved family of Atlastin proteins, which are ER-localized dynamin-like

*itr-1pB::unc-26$^{ER}$*, n.s. denotes not significant). **(F)** Representative live SDC-SIM images of the periphery of DA9 cell body, expressing mNeonGreen::ESYT-2, from wild-type control and *cil-1 (yas37)* mutants. Single focal planes are shown. A yellow arrowhead in a *cil-1 (yas37)* mutant soma denotes the accumulation of mNeonGreen::ESYT-2 in expanded sheet-like ER. Scale bars, 2 $\mu$m. **(G)** Quantification of the ratio of the total area of ER sheets to that of ER tubules, as assessed by mNeonGreen::ESYT-2 signals at the periphery of DA9 cell body, in wild-type control (n = 15) and *cil-1 (yas37)* (n = 15) (mean ± SEM; Two-tailed unpaired *t* test, **P = 0.0003). **(H)** Representative live SDC images of a DA9 neuron, showing ER-PM contacts (green: reconstituted GFP) and the PM (red: mCherry::3xPH$^{PLCδ1}$::splitGFP11) that are labelled by the split GFP approach, in dendrite and dorsal axon of *cil-1 (yas37)* mutants. Note the ectopic formation and expansion of ER-PM contacts in dorsal axon of *cil-1 (yas37)* mutants compared to wild-type control (compare with Fig 1E). Scale bars, 20 $\mu$m. **(I)** Quantification of the ratio of reconstituted GFP fluorescence intensity (ER-PM contacts) of dorsal axon to that of dendrite in DA9 neuron of wild-type control (n = 15) and *cil-1 (yas37)* (n = 15) (mean ± SEM; Two-tailed unpaired *t* test, **P < 0.0001).
Source data are available for this figure.

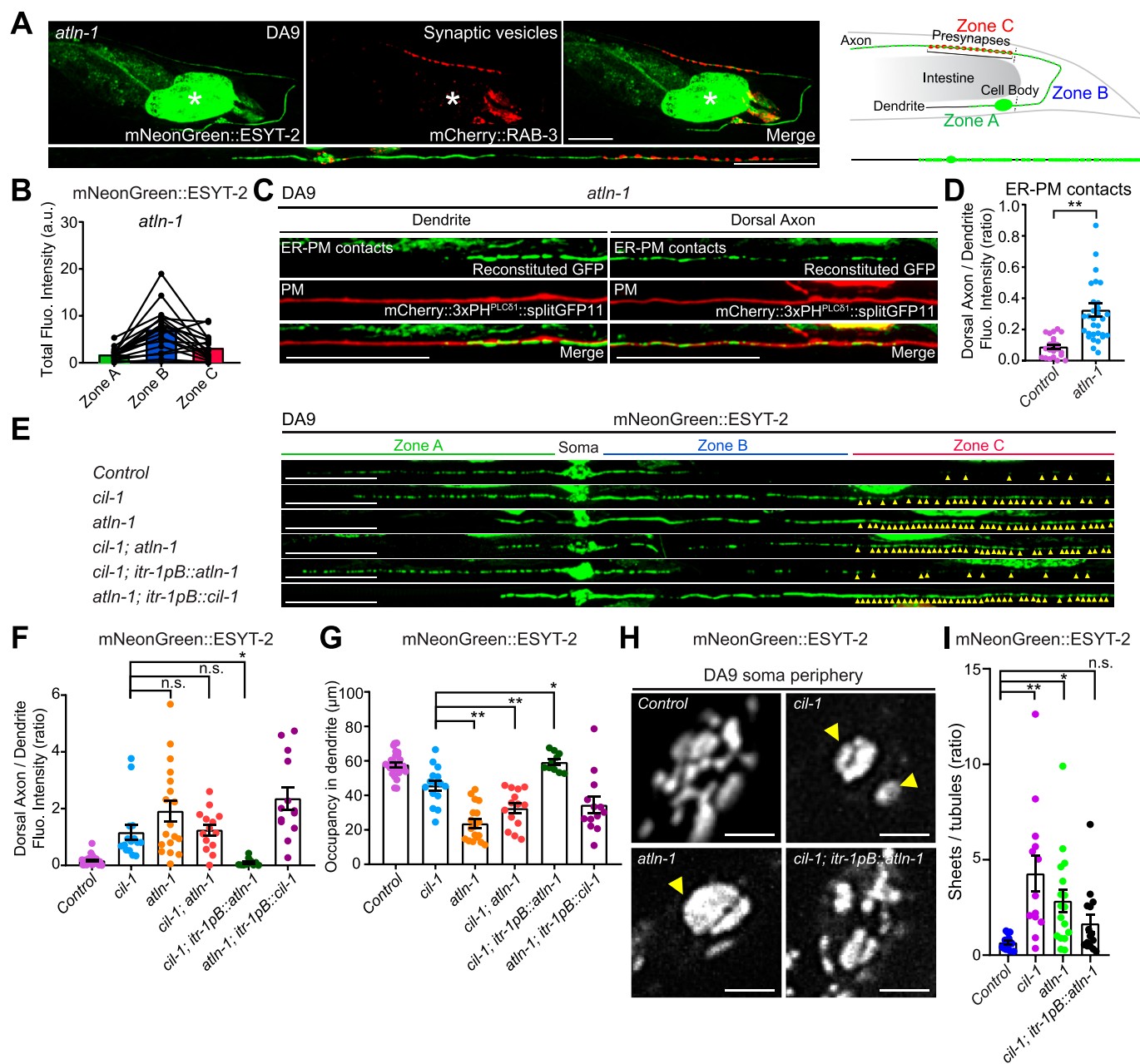

**Figure 5. CIL-1 and ATLN-1 have related functions in maintaining ER shape in neurons.**
**(A)** Left: Representative live spinning disc confocal (SDC) images of a DA9 neuron, co-expressing mNeonGreen::ESYT-2 and mCherry::RAB-3, in *atln-1 (yas38)* mutants. A straightened image of the DA9 neuron is shown below. Right: A schematic showing the distribution of mNeonGreen::ESYT-2 (marked as small green circles) in DA9 neuron. Note the altered distribution of mNeonGreen::ESYT-2 in *atln-1 (yas38)* mutants compared to wild-type control (compare to Fig 3A). Scale bars, 20 μm. **(B)** Quantification of the total fluorescence intensity of mNeonGreen::ESYT-2 present in different zones of *atln-1 (yas38)* mutants (n = 18) shows increased abundance of ESYT-2 in zone C compared to Fig 3C). **(C)** Representative live SDC images of a DA9 neuron, showing ER-PM contacts (green: reconstituted GFP) and the PM (red: mCherry::3xPH^PLCδ1::splitGFP11) that are labelled by the split GFP approach, in dendrite and dorsal axon of *atln-1 (yas38)* mutants. Note the ectopic formation and expansion of ER-PM contacts in dorsal axon compared to wild-type control (compare to Fig 1E). Scale bars, 20 μm. **(D)** Quantification of the ratio of reconstituted GFP fluorescence intensity (ER-PM contacts) of dorsal axon to that of dendrite in DA9 neuron of wild-type control (n = 21) and *atln-1 (yas38)* (n = 31) (mean ± SEM; Two-tailed unpaired *t* test, **P < 0.0001). **(E)** Representative live SDC images of straightened DA9 neurons of animals with indicated genotype. Yellow arrowheads mark mNeonGreen::ESYT-2 signals in zone C. Note that overexpression of ATLN-1 in *cil-1 (yas37)* mutants suppressed the ectopic localization of mNeonGreen::ESYT-2 in zone C.
**(F)** Quantification of the ratio of mNeonGreen::ESYT-2 fluorescence intensity of dorsal axon (zone C) to that of dendrite (zone A) in DA9 neurons of animals as shown in (E): wild-type control (n = 26), *cil-1 (yas37)* (n = 15), *atln-1 (yas38)* (n = 18), *cil-1 (yas37);atln-1 (yas38)* (n = 14), *cil-1 (yas37); itr-1pB::atln-1* (n = 10), and *atln-1 (yas38); itr-1pB::cil-1* (n = 13) (mean ± SEM; Dunnett's multiple comparisons test; *P = 0.0432 [*cil-1 (yas37)* versus *cil-1 (yas37); itr-1pB::atln-1*], n.s. denotes not significant). **(G)** Quantification of the occupancy of mNeonGreen::ESYT-2 signals in dendrites (zone A) of DA9 neurons from the same animals as in (F). (mean ± SEM; Dunnett's multiple comparisons test; **P < 0.0001 [*cil-1 (yas37)* versus *atln-1 (yas38)*], **P = 0.0054 [*cil-1 (yas37)* versus *cil-1 (yas37);atln-1 (yas38)*], *P = 0.0108 [*cil-1 (yas37)* versus *cil-1 (yas37); itr-1pB::atln-1*]). **(H)** Representative live SDC-SIM images of the periphery of DA9 cell body, expressing mNeonGreen::ESYT-2, from animals with indicated genotype. Single focal planes are shown. Yellow arrowheads in *cil-1 (yas37)* and *atln-1 (yas38)* mutants denote the accumulation of mNeonGreen::ESYT-2 in expanded sheet-like ER. Note the partial

GTPases. Similar to other Atlastin proteins, ATLN-1 consists of a GTPase domain followed by a three-helix bundle (3HB), transmembrane (TM) segments, and a cytosolic tail (C-Tail) (Liu et al, 2019) (Fig S5B). The glutamic acid that is mutated to lysine in the *atln-1(yas38)* mutant (E338K) is located near the 3HB domain, which is required for oligomerization of Atlastins (Bian et al, 2011; Byrnes & Sondermann, 2011). Overexpression of the ATLN-1 (E388K) mutant protein partially restored the distribution of mNeonGreen::ESYT-2 in DA9 neurons, suggesting that ATLN-1 (E388K) retains some function (Fig S5B–D, G, and H). DA9-specific overexpression of an ATLN-1 variant (K80A) with dominant-negative GTPase activity (Liu et al, 2019) in wild-type animals resulted in altered distribution of mNeonGreen::ESYT-2, supporting the critical role of ATLN-1's GTPase activity in maintaining a normal distribution of ESYT-2 in DA9 neurons (Fig S5B and E–H).

To gain further insight into the relationship between *cil-1* and *atln-1,* we performed genetic epistasis analysis between these mutants. *cil-1(yas37); atln-1(yas38)* double mutants displayed defects in the distribution of mNeonGreen::ESYT-2, similar to *atln-1(yas38)* single mutants (Fig 5E–G), consistent with related functions of *cil-1* and *atln-1.* Strikingly, overexpression of *atln-1* in DA9 of *cil-1(yas37)* mutants suppressed the ectopic accumulation of mNeonGreen::ESYT-2 in the dorsal axon (zone C) and partially restored the distribution of mNeonGreen::ESYT-2 in the dendrite (zone A) (Fig 5E–G). In contrast, *cil-1* overexpression failed to rescue mNeonGreen::ESYT-2 distribution in *atln-1(yas38)* mutants (Fig 5E–G). These results demonstrate that ATLN-1 acts genetically downstream of CIL-1.

Genetic interaction between *cil-1* and *atln-1* suggests that proteins encoded by these genes may function together to maintain the structure of cortical ER. Thus, we further examined the relationship between these proteins by imaging cortical ER. We used SDC-SIM to visualize mNeonGreen::ESYT-2 at the periphery of the DA9 cell body. Notably, *atln-1(yas38)* mutants displayed dramatic expansion of cortical ER sheets, similar to *cil-1(yas37)* mutants (Fig 5H and I). Overexpression of ATLN-1 in *cil-1(yas37)* mutants partially restored the balance between ER tubules and sheets and suppressed the expansion of cortical ER sheets. These results indicate that ATLN-1 overexpression, at least partially, bypasses the function of CIL-1 in maintaining the network of cortical ER tubules and sheets (Fig 5H and I). Collectively, our results demonstrate that ATLN-1 functions downstream of CIL-1 to maintain the structure of cortical ER, and further suggest that the expansion of cortical ER sheets, as observed in *cil-1* and *atln-1* mutants, underlies ectopic formation of ER-PM contacts in neuronal axons.

### Cortical ER shape is important for the normal distribution of ER-PM contacts in neurons

Our results suggested that cortical ER shape is potentially important for maintaining the distribution of neuronal ER-PM contacts. To further test this idea, we analyzed mutants that lacked the curvature-stabilizing ER-shaping protein, reticulon. Yeast mutants that lack reticulons have fewer cortical ER tubules and more cortical ER sheets (De Craene et al, 2006; Voeltz et al, 2006; West et al, 2011). Consistent with these yeast studies, *C. elegans* mutants lacking reticulon, *ret-1(tm390)* (RET-1 is the sole homolog of reticulons in *C. elegans*), had fewer cortical ER tubules and enlarged cortical ER sheets (as assessed by mNeonGreen::ESYT-2) in the DA9 soma, similar to *cil-1* or *atln-1* mutants (Fig 6A). Furthermore, the dendritic ER, which was assessed by mNeonGreen::ESYT-2, was shortened compared with wild-type control similar to these mutants (Fig S6B). Importantly, *ret-1(tm390)* mutants also exhibited an altered distribution of mNeonGreen::ESYT-2 in DA9 neuron, where mNeonGreen::ESYT-2 was present uniformly throughout the entire axon (Figs 6B and C and S6A). ER-PM contacts, labelled by the split GFP approach, were also ectopically present in the dorsal axon in *ret-1(tm390)* mutants (Fig 6D and E).

To further confirm the role of cortical ER shape in the distribution of ER-PM contacts in neurons, we overexpressed CLIMP-63, a luminal ER spacer whose overexpression is known to result in expansion of ER sheets in mammalian cells (Shibata et al, 2010) (there is no homolog of CLIMP-63 in *C. elegans*), in DA9 neurons and examined its effect on the distribution of ESYT-2. CLIMP-63 overexpression resulted in expansion of cortical ER sheets and reduction in cortical ER tubules, as assessed by mNeonGreen::ESYT-2, in the DA9 soma, similar to *cil-1, atln-1,* and *ret-1* mutants (Fig S6C). Notably, this was accompanied by altered distribution of mNeonGreen::ESYT-2 in DA9 axons; mNeonGreen::ESYT-2 became more abundant in the commissure region of DA9 (and less restricted to the somatodendritic region of DA9) compared to wildtype control (Fig S6D–F). These results suggest that the abundance of cortical ER sheets affects the distribution of ER-PM contacts in neurons. However, we could not detect the increased presence of mNeonGreen::ESYT-2 in the dorsal axons of DA9 in this condition (Fig S6G). This could be possibly due to preferential targeting of CLIMP-63 in somatodendritic regions of neurons as previously reported (Farias et al, 2019).

Taken together, these data support the notion that cortical ER shape, and more specifically the proper balance of ER tubules and sheets, play a role in restricting the distribution of neuronal ER-PM contacts to specific regions within neurons.

### Expansion and invasion of ER sheets in dorsal axon contribute to abnormal distribution of neuronal ER-PM contacts

Using electron microscopy to visualize ER structure in *C. elegans* neurons poses a challenge due to its small size. In *C. elegans* neurons, the ribosome-rich domain of the ER, or rough ER, is found primarily in the soma and is generally absent from the axon (Rolls et al, 2002; Saheki & Bargmann, 2009). Thus, we visualized the distribution of fluorescent protein-labelled rough ER proteins via fluorescent microscopy and examined whether ER sheets invaded into the dorsal axons of DA9 in *cil-1, atln-1,* and *ret-1* mutants.

---

rescue of cortical ER structure in *cil-1 (yas37)* mutants overexpressing ATLN-1 under DA9 specific *itr-1pB* promoter. Scale bars, 2 μm. **(I)** Quantification of the ratio of the total area of ER sheets to that of ER tubules, as assessed by mNeonGreen::ESYT-2 signals at the periphery of DA9 cell body, in wild-type control (n = 14), *cil-1 (yas37)* (n = 13), *atln-1 (yas38)* (n = 17), and *cil-1 (yas37); itr-1pB::atln-1* (n = 15) (mean ± SEM; Dunnett's multiple comparisons test; **P = 0.0004 [*Control* versus *cil-1 (yas37)*], *P = 0.0265 [*Control* versus *atln-1 (yas38)*], n.s. denotes not significant).
Source data are available for this figure.

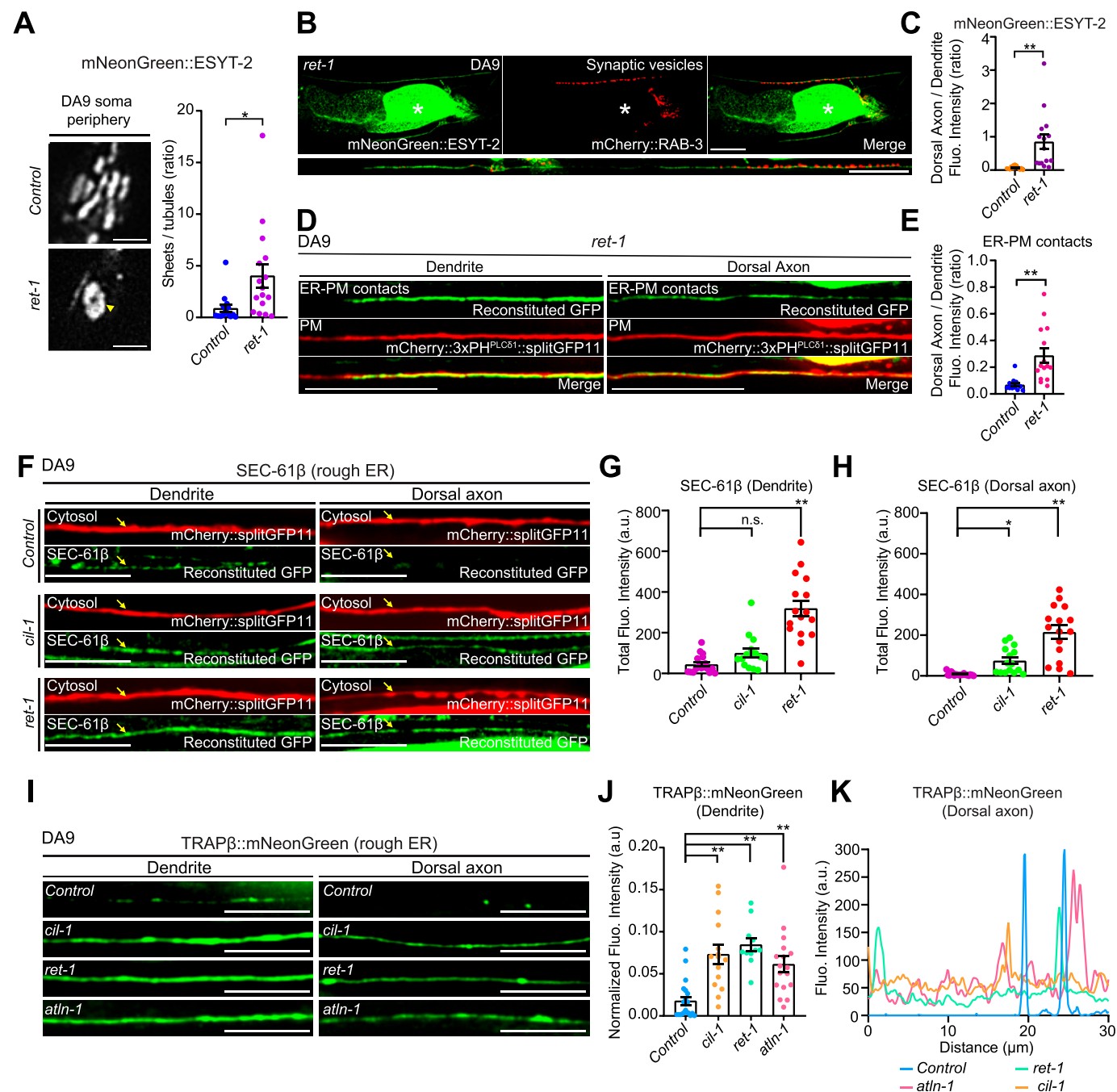

**Figure 6. Abnormal distribution of neuronal ER-PM contacts is accompanied by expansion of ER sheets in dorsal axon.**
**(A)** Left: Representative live spinning disc confocal (SDC)-SIM images of the periphery of DA9 cell body, expressing mNeonGreen::ESYT-2, in wild-type control and *ret-1 (tm390)* mutants. Single focal planes are shown. A yellow arrowhead denotes the accumulation of mNeonGreen::ESYT-2 in expanded sheet-like ER. Scale bars, 2 μm. Right: Quantification of the ratio of the total area of ER sheets to that of ER tubules, as assessed by mNeonGreen::ESYT-2 signals at cell periphery, in wild-type control (n = 15) and *ret-1 (tm390)* (n = 16) (mean ± SEM; two-tailed unpaired *t* test, *P = 0.0146). **(B)** Representative live SDC images of a DA9 neuron, co-expressing mNeonGreen::ESYT-2 and mCherry::RAB-3, from *ret-1 (tm390)* mutants. A straightened image of the DA9 neuron is shown below. The central autofluorescent region is the intestine (asterisk). Scale bars, 20 μm. **(C)** Quantification of the ratio of mNeonGreen::ESYT-2 fluorescence intensity of dorsal axon (zone C) to that of dendrite (zone A) of DA9 in wild-type control (n = 15) and *ret-1 (tm390)* (n = 15) (mean ± SEM; Two-tailed unpaired *t* test, **P = 0.0015). **(D)** Representative live SDC images of dendrite and dorsal axon of DA9, showing ER-PM contacts (green: reconstituted GFP) and the PM (red: mCherry::3xPH^PLC^δ1::splitGFP11) labelled by the split GFP approach in *ret-1 (tm390)* mutants. **(E)** Quantification of the ratio of reconstituted GFP fluorescence intensity (ER-PM contacts) of dorsal axon to that of dendrite in DA9 of wild-type control (n = 15) and *ret-1 (tm390)* (n = 15) (mean ± SEM; Two-tailed unpaired *t* test, **P = 0.0004). **(F)** Representative live SDC images of DA9 neurons, carrying endogenously tagged splitGFP1-10::SEC-61β and additionally expressing mCherry-tagged splitGFP11 (mCherry::splitGFP11) under DA9 selective *mig-13* promoter, showing the endogenous localization of SEC-61β (green: reconstituted GFP) and the cytosol (red: mCherry::splitGFP11) in animals of indicated genotype. Yellow arrows indicate the neuronal processes. Scale bars, 10 μm. **(G)** Quantification of the total fluorescence intensity of reconstituted GFP (endogenous SEC-61β) in dendrite of wild-type control (n = 18), *cil-1 (yas37)* (n = 15), and *ret-1 (tm390)* (n = 16) (mean ± SEM; Dunnett's multiple comparisons test; **P < 0.0001 [*Control* versus *ret-1 (tm390)*], n.s. denotes not significant). **(H)** Quantification of the total

In the first set of experiments, we endogenously tagged the cytosolic N terminus of rough ER marker SEC-61$\beta$ (SEC61 translocon $\beta$ subunit) with one component of split GFP (fragments 1–10) using CRISPR-based gene-editing. We then expressed the other component of split GFP (fragment 11) together with soluble mCherry using the DA9-specific *mig-13* promoter. Thus, we could visualize endogenously tagged SEC-61$\beta$ via GFP reconstitution specifically in the mCherry-labelled DA9 neuron (Fig 6F–H). In wild-type controls, SEC-61$\beta$ was mostly present in the soma, with little in the dendrite and none in the axon (Figs 6F–H and S6H). However, in *ret-1(tm390)* and *cil-1(yas37)* mutants, SEC-61$\beta$ was clearly present in the axon, and was also increased in the dendrite. In contrast, SEC-61$\beta$ levels were not affected in the soma (Fig S6H). As the *sec-61.B* gene is less than 1 centimorgan from the *atln-1* gene, we were unable to establish a strain carrying the *sec-61.B* gene tagged with the split GFP component in the *atln-1* mutant.

In the second set of experiments, we expressed a version of translocon-associated protein $\beta$ (TRAP$\beta$), another rough ER marker, in which its C-terminus was tagged with mNeonGreen, in DA9 via the DA9-specific *itr-1pB* promoter (Fig 6I–K). In control animals, TRAP$\beta$::mNeonGreen primarily localized to the cell body with little presence in the dendrite or axon. In the dorsal axon, TRAP$\beta$::mNeon-Green localized to a few distinct puncta, suggesting that it accumulates in specific compartments within axonal ER (Fig 6I–K). In *cil-1(yas37)*, *ret-1(tm390)*, and *atln-1(yas38)* mutants, however, TRAP$\beta$::mNeonGreen was much more abundant in the dendrite and more diffusely localized within the axon compared with control animals. Levels of TRAP$\beta$::mNeonGreen in the soma were not affected (Fig S6I).

Collectively, these results show that ER sheets expanded and invaded into the dorsal axon and became more abundant in dendrite of DA9 neuron in *cil-1(yas37)*, *ret-1(tm390)*, and *atln-1(yas38)* mutants. Our data strongly suggest that these mutants exhibited ectopically formed ER-PM contacts in the dorsal axon because of expansion and invasion of cortical ER sheets in this region.

### ER-PM contacts regulated by CIL-1 and ATLN-1 are important for efficient axon regeneration

Axon regeneration is an important physiological response to various insults to neurons. The role of ER-PM contacts in this process remains unclear. After laser-induced damage to the DA9 dorsal axon, it has previously been shown that the axon proximal to the damage grows back (i.e., regenerates). In *ric-7(n2657)* mutants, the axon distal to the damage degenerates much more quickly than wild-type animals, eliminating potential interference from the remaining distal axon fragments after the damage (Ding & Hammarlund, 2018) (Fig 7A–C).

Metazoan ER-PM contacts are populated by various tethering proteins, including E-Syts and Junctophilins. Interestingly, the absence of Junctophilin (JPH-1 in *C. elegans*), which is a metazoan-specific ER-PM tethering protein, negatively affected axon regeneration for the *C. elegans* mechanosensory PLM neuron, after laser-induced axonal injury. In contrast, *esyt-2* is dispensable for axon regeneration of the PLM neuron (Kim et al, 2018). We investigated axon regeneration in DA9 neurons, using the *ric-7(n2657)* mutation as a background mutation. To sever the DA9 dorsal axon without damaging surrounding tissues, a pulsed laser was applied to a region of the axon near the presynaptic region (see the Materials and Methods section), and axon regrowth was assessed 24 and 48 h after axotomy.

Similar to PLM axon, DA9 axon regeneration was normal in the absence of *esyt-2*, but it was significantly reduced in the absence of *jph-1* (Fig 7D and E). In control and *esyt-2* mutant animals, the axon grew back to ~150 and ~190 $\mu$m in length from the cut-site at 24 and 48 h, respectively. However, in *jph-1 (ok2823)* mutants, the axon regrew to ~100 and ~150 $\mu$m from the cut-site at 24 and 48 h, respectively (Fig 7D and E). These results suggest that proper function/assembly of neuronal ER-PM contacts is important for maintaining robust axon regeneration in DA9 neurons. We also examined the distribution of ER-PM contacts in regenerating axons. To this end, the distribution of ER-PM contacts was assessed 24 h after axotomy either by the split GFP approach or mNeonGreen:: ESYT-2. No significant changes in the distribution of ER-PM contacts were observed in axons or in dendrites (Fig S7A–D). These results indicate that the overall distribution of neuronal ER-PM contacts is maintained during axon regeneration after laser-induced injury.

The distribution of neuronal ER-PM contacts in *cil-1* and *atln-1* mutants is severely altered, and thus, it is possible that these contacts are functionally abnormal. To examine such possibility, we determined whether *cil-1* and *atln-1* mutants show any changes in resilience to axonal damage. Remarkably, similar to *jph-1 (ok2823)* mutants, *cil-1(yas37)* and *atln-1(yas38)* mutants exhibited significantly reduced axon regeneration, supporting the notion that ER-PM contacts in these mutants are functionally abnormal. Even 48 h after axotomy, the axon was only ~130 $\mu$m from the cut-site in these mutants (Fig 7F and G). DA9-specific re-expression of CIL-1 in *cil-1(yas37)* mutants and ATLN-1 in *atln-1(yas38)* mutants (via the *itr-1pB* promoter) rescued these phenotypes, confirming that CIL-1 and ATLN-1 function cell autonomously to maintain robust axon regeneration (Fig 7F and G).

These results suggest that proper distribution (non-uniform distribution) of neuronal ER-PM contacts is essential for maintaining robust axon regeneration. If this is indeed the case, restoring the distribution of neuronal ER-PM contacts alone should be sufficient to restore axon regeneration in *cil-1(yas37)* mutants. To test this hypothesis, we performed the laser axotomy experiment in

---

fluorescence intensity of reconstituted GFP (endogenous SEC-61$\beta$) in dorsal axon of the same animals as in (G) (mean ± SEM; Dunnett's multiple comparisons test; *$P$ = 0.0468 [*Control* versus *ret-1 (tm390)*], **$P$ < 0.0001 [*Control* versus *cil-1 (yas37)*]). **(I)** Representative live SDC images of dendrite and dorsal axon of DA9 neurons, expressing mNeonGreen-tagged rough ER marker TRAP$\beta$ (TRAP$\beta$::mNeonGreen) in animals with indicated genotype. Note the increased abundance of TRAP$\beta$:: mNeonGreen in both dendrite and dorsal axon in *cil-1(yas37)*, *ret-1(tm390)*, and *atln-1(yas38)* mutants. Scale bars, 20 $\mu$m. **(J)** Quantification of normalized fluorescence intensity of TRAP$\beta$::mNeonGreen in dendrite of DA9 neuron from wild-type control (n = 21), *cil-1 (yas37)* (n = 15), *ret-1 (tm390)* (n = 11), and *atln-1 (yas38)* (n = 17) (mean ± SEM; Dunnett's multiple comparisons test; **$P$ < 0.0001 [*Control* versus *cil-1 (yas37)* and *Control* versus *ret-1 (tm390)*], **$P$ = 0.0006 [*Control* versus *atln-1 (yas38)*]). **(K)** The representative line scan profiles of TRAP$\beta$::mNeonGreen along the dorsal axons of animals with indicated genotype. Source data are available for this figure.

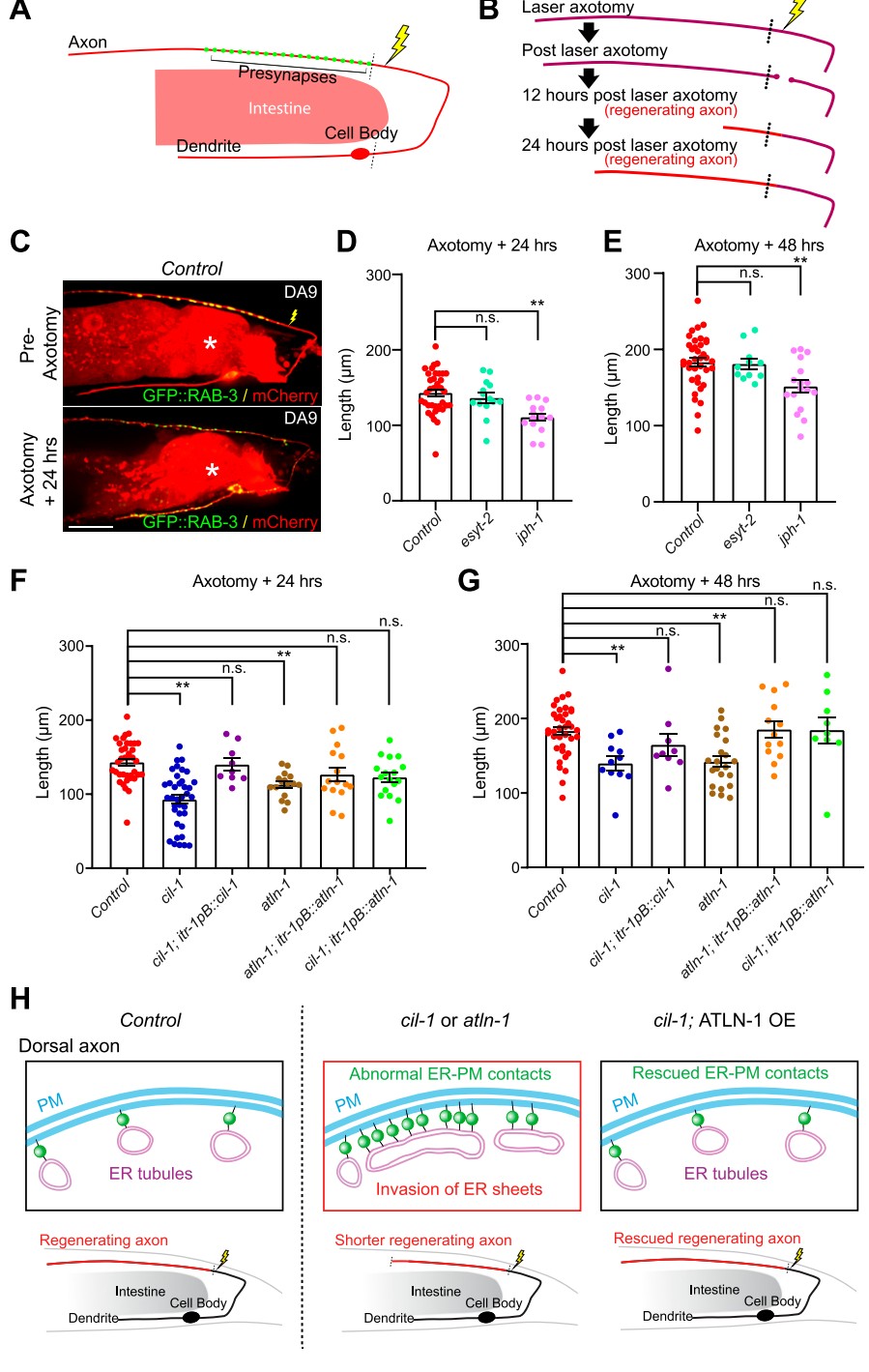

**Figure 7. CIL-1 and ATLN-1 are both required for efficient axon regeneration.**

**(A)** A schematic diagram showing the laser ablation site (a yellow lightning mark) proximal to the presynaptic region of DA9 neuron. **(B)** Cartoon depicting axon regeneration process at different time points after laser-induced axotomy. Red indicates newly regenerated axon after laser axotomy. Note that the axotomy was considered successful when two clear bulbs were observed at both ends of a severed axon. **(C)** Representative live spinning disc confocal images of DA9 neurons, co-expressing cytosolic soluble mCherry (red) and GFP::RAB-3 (green), from wild-type control animals before and 24 h after laser axotomy. The lightning marks denote the axotomy sites. The central autofluorescent region is the intestine (asterisk). Scale bar, 20 $\mu m$. **(D, E)** Quantification of regenerated DA9 dorsal axon length at 24 h (D) and 48 h (E) after laser axotomy in wild-type control and mutants as indicated. For 24 h as shown in (D): wild-type control (n = 37), esyt-2 (syb709) (n = 14), and jph-1 (ok2823) (n = 15) (mean ± SEM; Dunnett's multiple comparisons test; **$P$ = 0.0002 [Control versus jph-1(ok2823)], n.s. denotes not significant). For 48 h (E): wild-type control (n = 37), esyt-2 (syb709) (n = 11), and jph-1 (ok2823) (n = 17) (mean ± SEM; Dunnett's multiple comparisons test; **$P$ = 0.0029 [Control versus jph-1 (ok2823)], n.s. denotes not significant). Note the reduced axon regeneration compared to wild-type control in jph-1 (ok2823) mutants. **(F, G)** Quantification of regenerated DA9 dorsal axon length at 24 h (F) and 48 h (G) after laser axotomy in wild-type control and mutants as indicated. For 24 h (F): wild-type control (n = 37, same control as in D), cil-1 (yas37) (n = 37), cil-1 (yas37); itr-1pB::cil-1 (n = 9), atln-1 (yas38) (n = 16), atln-1 (yas38); itr-1pB::atln-1 (n = 15), and cil-1 (yas37); itr-1pB:: atln-1 (n = 17) (mean ± SEM; Dunnett's multiple comparisons test; **$P$ < 0.0001 [Control versus cil-1 (yas37)], **$P$ = 0.0057 [Control versus atln-1 (yas38)], n.s. denotes not significant). For 48 h (G): wild-type control (n = 37, same control as in E), cil-1 (yas37) (n = 11), cil-1 (yas37); itr-1pB::cil-1 (n = 9), atln-1 (yas38) (n = 22), atln-1 (yas38); itr-1pB::atln-1 (n = 13), and cil-1 (yas37); itr-1pB::atln-1 (n = 9) (mean ± SEM; Dunnett's multiple comparisons test; **$P$ = 0.0057 [Control versus cil-1 (yas37)], **$P$ = 0.0005 [Control versus atln-1 (yas38)], n.s. denotes not significant). **(H)** A model depicting the importance of the balance of ER sheets and tubules in maintaining the distribution of ER-PM contacts in DA9 neurons. In wild-type control animals, the axonal region is largely occupied with smooth tubular ER, whereas the somatodendritic region is occupied by both rough and smooth ER that consist of the network of ER sheets and tubules. The balance of ER tubules and sheets restrict the formation of neuronal ER-PM contacts primarily in somatodendritic region. In cil-1 and atln-1 mutants, ER sheets expand and invade into the dorsal axon as seen by ectopic presence of rough ER proteins in this region. This is accompanied by abnormal distribution of ER-PM contacts and disruption of their functions in neurons. Such disruption, at least in part, contributes to reduced axon regeneration in these mutants. Supporting this notion, overexpression of ATLN-1 in cil-1 (yas37) mutants, which rescues the distribution of ER-PM contacts, is sufficient to restore axon regeneration. Source data are available for this figure.

cil-1(yas37) mutants overexpressing ATLN-1 in DA9 neuron, which exhibited a normal distribution of ER-PM contacts despite the complete absence of CIL-1 (see Fig 5E–G). Overexpression of ATLN-1 in cil-1(yas37) mutants restored axon regeneration, as the axon regrew to ~140 and ~190 $\mu m$ from the cut-site 24 and 48 h after axotomy, respectively, similar to controls (Fig 7F and G).

Taken together, our results demonstrate the critical role of CIL-1 and ATLN-1 in axon regeneration following axonal damage and

suggest the importance of maintaining the non-uniform distribution of neuronal ER-PM contacts in this process.

## Discussion

In this study, we used the *C. elegans* motor neuron DA9 to understand how the distribution of neuronal ER-PM contacts is established and maintained. We showed that the distribution of neuronal ER-PM contacts critically depends on the activities of proteins that are involved in shaping the ER. Furthermore, they are essential for maintaining neuronal resilience against insults. Key findings of our current study are the following:

(1) We successfully visualized the distribution of ER-PM contacts in neurons in vivo using a split GFP approach and found that neuronal ER-PM contacts were non-uniformly distributed (enriched in somatodendritic regions and rare in axon) in *C. elegans*. Remarkably, some of these contacts contained the evolutionarily conserved ER-PM tethering protein, ESYT-2 (E-Syts).

(2) Using an unbiased forward genetic screen, we identified mutants that exhibited abnormal distribution of ER-PM contacts. Analysis of these mutants—namely, *cil-1(yas37)* and *atln-1(yas38)*—revealed the importance of cortical ER shape in maintaining the non-uniform distribution of ER-PM contacts in neurons. These mutants showed defects in ER tubule formation that was accompanied by expansion of cortical ER sheets and the ectopic formation of ER-PM contacts in dorsal axons.

(3) We found that the inositol 5-phosphatase, CIL-1 (human INPP5K), functioned at the ER, regulated the cortical ER network, and maintained the distribution of ER-PM contacts in neurons. In the absence of CIL-1, cortical ER sheets expanded with a concomitant reduction in ER tubules. Importantly, our results reveal that CIL-1 acts genetically upstream of ER-shaping ATLN-1 (human Atlastin-1). Overexpression of ATLN-1 could bypass the requirement of CIL-1 to regulate cortical ER structure and maintain the distribution of ER-PM contacts, indicating their related functions.

(4) We found that the restriction of cortical ER sheets to somatodendritic regions was maintained by proteins that promote ER tubule formation, including RET-1 (human reticulons), CIL-1, and ATLN-1. In the mutants with reduced functions of any of these proteins, rough ER sheets, which normally reside in the somatodendritic compartment, ectopically localized to the axonal compartment. Thus, the coordinated action of multiple ER-shaping proteins is critical for maintaining the identity of axonal ER and restricting cortical ER primarily to somatodendritic regions.

(5) Our results demonstrate the importance of CIL-1 and ATLN-1 in the resilience of neurons after neuronal damage. Using laser axotomy, we showed that efficient axon regeneration required the functions of these proteins. When the distribution of ER-PM contacts was restored in *cil-1* mutants (by ATLN-1 overexpression), axon regeneration, which relies on normal function of ER-PM contacts (Kim et al, 2018), was also restored. This suggests that the normal distribution of ER-PM contacts is critical for the function of these contacts and for maintaining the ability of axons to regenerate after an insult (Fig 7H).

Our results demonstrate that in the *C. elegans* DA9 neuron, the ER forms a network of sheets and tubules, similar to those seen in yeast and mammalian cells. Furthermore, we found that ER-PM contacts are most abundant in somatodendritic regions and scarce in the axon of this neuron. Similar to what we showed in the present study, previous high-resolution 3D reconstruction of ER structure in mammalian neurons via electron microscopy has reported a non-uniform distribution of ER-PM contacts in different regions of the cell, with particular enrichment in the soma (Wu et al, 2017). Thus, the molecular mechanisms responsible for maintaining neuronal ER-PM contacts are likely evolutionarily conserved. Axonal ER is mostly tubular and smooth, whereas somatodendritic ER is a mixture of rough and smooth ER and consists of a network of both sheets and tubules (Wu et al, 2017; Yalcin et al, 2017; Terasaki, 2018; Farias et al, 2019). Although how exactly CIL-1/INPP5K and ATLN-1/Atlastin control the distribution of neuronal ER-PM contacts remains elusive, our study suggests that the balance between ER sheets and tubules may determine where ER-PM contacts are distributed in neurons based on the reported functions of INPP5K and Atlastin to regulate ER morphologies (see below). Axonal ER tubules may physically prevent ER-PM contacts from forming. In support of this notion, mutant worms lacking proteins involved in ER tubule formation, including RET-1/Reticulon, CIL-1, and ATLN-1, show ectopic formation of ER-PM contacts in the dorsal axon. We further provide evidence that smooth ER tubules in axons become more sheet-like with the ectopic presence of rough ER proteins in the reduced/eliminated activities of these proteins, resulting in the expansion of cortical ER sheets and ectopic formation of ER-PM contacts in axons (Fig 7H). We acknowledge the heterogeneity of ER-PM contacts. Because of the difficulty in determining precise ultrastructure of the ER by electron microscopy in *C. elegans* neurons, we used fluorescence-based detection of neuronal ER-PM contacts (i.e., the split GFP approach and ESYT-2). However, both the split GFP approach and ESYT-2 do not label all ER-PM contacts. Further studies are needed to examine whether the proteins that we investigated in the current study have general roles in maintaining the distribution of all types of ER-PM contacts.

Our genetic and cell biological data suggest that CIL-1 acts on the ER and regulates ATLN-1 activities. However, it is still not clear how CIL-1 function is related to ATLN-1 function. There is no strong evidence that supports the presence of $PI(4,5)P_2$ in ER membranes. It is possible that the presence of CIL-1/INPP5K is the exact reason why $PI(4,5)P_2$ is not abundant in ER membranes; removal of $PI(4,5)P_2$ from ER membranes by CIL-1-dependent $PI(4,5)P_2$ dephosphorylation may be essential for ATLN-1 to regulate the balance of ER tubules and sheets. INPP5K was also reported to hydrolyze $PI(3,4,5)P_3$, a phosphoinositide that is normally produced by phosphoinositide 3-kinase (PI3K) at the PM (Ijuin et al, 2000; Gurung et al, 2003; Schmid et al, 2004; Dong et al, 2018). Thus, whether $PI(4,5)P_2$ is the major substrate of CIL-1 deserves further investigation. Atlastins are primarily implicated in maintaining three-way ER junctions by promoting homotypic fusion of ER tubules. However, apart from its well-known function in homotypic ER fusion, there is evidence that Atlastins may play a role in ER tubule formation. First, Atlastins physically interact with curvature-stabilizing ER-shaping proteins, including reticulons and DP1 (Hu et al, 2009). Second, in yeast the depletion of Sey1p (the dynamin-like GTPase that resembles the Atlastin) together with Yop1p or Rtn1p results in loss of ER tubules and an increase in ER sheets (Hu et al, 2009; Anwar et al, 2012). Interestingly, the depletion of Sey1p together with Yop1p or Rtn1p

also leads to the expansion of ER-PM contacts, as seen in other yeast mutants lacking ER tubule-shaping proteins (Rtn1p, Rtn2p, and DP1/Yop1p) (West et al, 2011). Thus, ATLN-1/Atlastins may also function to regulate ER tubule formation in certain circumstances. In fact, our newly isolated *atln-1* reduction-of-function mutant showed a reduction in ER tubules and an expansion of cortical ER sheets, supporting this notion.

Our studies also provide novel insights into the mechanisms of ER domain segregation in neurons. Rough ER, which consists of ER sheets, localizes predominantly to the somatodendritic region but is generally absent from the dorsal axon in the DA9 neuron (Fig 6F). The lack of rough ER in most axonal processes has been reported for other *C. elegans* neurons (Rolls et al, 2002). Similar patterns of rough ER has also been seen in mammalian neurons, where rough ER localizes predominantly to the soma and dendrite, but is absent from the distal axon (Wu et al, 2017; Ozturk et al, 2020). Neurons are polarized cells, thus the limited presence of rough ER in the somatodendritic region is critical for the polarized transport of transmembrane and secreted proteins in neurons (Ramirez & Couve, 2011; Cui-Wang et al, 2012). However, the mechanisms that regulate the segregation of rough and smooth ER domains in neurons has been a mystery. Using SEC-61β and TRAPβ as rough ER markers, we found that ER-shaping proteins play critical roles in restricting the distribution of rough ER proteins to the somatodendritic region in the DA9 neuron. In mutants with reduced/eliminated activities of ATLN-1, RET-1, or CIL-1, these proteins are no longer restricted to the somatodendritic region, invading into axonal ER. As ATLN-1, RET-1, and CIL-1 are all involved in ER tubule formation in neurons (as shown in the current study), the formation of smooth tubular ER in axons may normally prevent SEC-61β and TRAPβ (and potentially other rough ER proteins) from invading the axonal ER, contributing to the limited presence of rough ER in the somatodendritic region.

ESYT-2 localized to ER-PM contacts in the DA9 neuron, suggesting that E-Syts/ESYT-2 are involved in the function of neuronal ER-PM contacts. Although ESYT-2 is dispensable for axon regeneration in the DA9 (as shown in the current study) and PLM neurons in *C. elegans* (Kim et al, 2018), a recent study reported that E-Syts are required for the structure and function of ER-PM contact sites in *Drosophila* photoreceptors (Nath et al, 2020). In mice, however, depletion of all the three E-Syts does not result in major abnormalities (Sclip et al, 2016; Tremblay & Moss, 2016). Thus, it is likely that ESYT-2 functions redundantly with other proteins, such as Junctophilin, at neuronal ER-PM contacts in *C. elegans*, although a recent study also suggested a non-cell autonomous function of Junctophilin in axon regeneration (Piggott et al, 2021). In yeast, tricalbins (yeast homologs of E-Syts) are distributed non-homogenously at cortical ER, localizing preferentially to highly curved regions of cortical ER sheets (Collado et al, 2019; Hoffmann et al, 2019). In this organism, ER shape contributes to the organization of ER-PM contacts into compositionally distinct subdomains, positioning tricalbins and other ER-anchored proteins within distinct regions of the cortical ER (Hoffmann et al, 2019). Whether the organization of cortical ER subdomains is altered in *C. elegans* mutants with reduced/eliminated activities of CIL-1, ATLN-1, or RET-1 needs further investigation. Importantly, E-Syt-mediated ER-PM tethering, as well as lipid transport, are both controlled by levels of cytosolic $Ca^{2+}$ (Chang et al, 2013; Giordano et al, 2013; Fernandez-Busnadiego et al, 2015; Idevall-Hagren et al, 2015; Saheki et al, 2016; Bian et al, 2018).

Thus, cytosolic $Ca^{2+}$-dependent regulation of the properties of neuronal ER-PM contacts also deserves further investigation.

Finally, our study suggests that the unique distribution of neuronal ER-PM contacts is associated with the resilience of neurons against neuronal damage. Laser axotomy experiments revealed that axon regeneration is significantly reduced in mutant worms with reduced/eliminated activities of CIL-1 or ATLN-1 compared to controls. Mutants lacking CIL-1 or ATLN-1 show ectopic formation of axonal ER-PM contacts. Petkovic et al (2014), suggested that lipid delivery at neuronal ER-PM contacts promote neurite outgrowth (Petkovic et al, 2014). Thus, proper compositions of ER-PM contacts rather than their abundance may be important for regeneration or outgrowth of axons. Our study does not rule out the possibility that the axonal ER morphology (i.e., tubular versus sheet) and motility as well as other related functions of the axonal ER, such as maintenance of $Ca^{2+}$ homeostasis and regulation of the functions of various other organelle contact sites, may also play a role in proper axon regeneration. INPP5K, a human homolog of CIL-1, is highly expressed in the developing and adult brain, eye, muscle, and kidney (Ijuin et al, 2008). Interestingly, overexpression of INPP5K has been shown to promote axon regeneration in mammalian neurons (Fink et al, 2017). Human bi-allelic point mutations that impair INPP5K's phosphatase activity lead to congenital muscular dystrophy with additional clinical manifestations that include cataracts, short stature, and intellectual impairments (Wiessner et al, 2017). Mutations in a number of well-established ER-shaping proteins, such as ATL1 (SPG3A) and RTN2 (SPG12), cause hereditary spastic paraplegia, a syndrome characterized by progressive weakness and spasticity mainly caused by the degeneration of motor and sensory axons (Blackstone, 2012). More recently, autosomal-dominant missense mutations in ATL3 were reported to cause hereditary sensory and autonomic neuropathy (HSAN) (Fischer et al, 2014; Kornak et al, 2014). These mutations disrupt fusion of the ER and induce aberrant ER tethering, causing major changes in ER morphology. Interestingly, cells carrying a disease-causing ATL3 mutation show pleiotropic defects in ER-related functions, including defects in vesicular transport and autophagosome formation (Krols et al, 2018, 2019; Behrendt et al, 2019). Our study shows that CIL-1/INPP5K and ATLN-1/Atlastin-1 help to maintain the non-uniform distribution of neuronal ER-PM contacts. These proteins likely contribute to the maintenance of the unique distribution of neuronal ER-PM contacts through their properties to control ER morphology, although we do not exclude the possibility that they control the distribution of ER-PM contacts indirectly through the regulation of other ER-related functions. Maintenance of the distribution of neuronal ER-PM contacts may be essential for the normal development and resilience of neurons in humans. Whether the development and progression of various neurological disorders is associated with the abnormal distribution and function of neuronal ER-PM contacts deserves further investigation.

## Materials and Methods

### *C. elegans* strains and maintenance

All *C. elegans* strains were maintained under standard conditions at room temperature (22°C–24°C) on nematode growth media (NGM) agar plates seeded with the bacterial *Escherichia coli* (*E. coli*) strain

OP50 as a food source, as previously described (Brenner, 1974). Wild-type was N2 Bristol strain. All experiments were performed on L4 larva hermaphrodite animals at room temperature, unless stated otherwise. Germline transformation was carried out as described (Mello & Fire, 1995). Some strains were provided by the Caenorhabditis Genetics Center and the National Bioresource Project. Mutants and balancers used were *jph-1(ok2823) I, cil-1(my15) III, ret-1(tm390) V, tmC25 [unc-5(tmIs1241)] IV, qC1 [dpy-19(e1259) glp-1(q339)] nIs189 III*.

## Molecular cloning and strains

Primers and plasmids used in this study are listed in Table S1. Strains used in this study are listed in Table S2.

### *For C. elegans expression*

**Cloning of *itr-1pB::CP450::splitGFP1-10*** Genomic DNA fragment corresponding to *itr-1pB* was ligated into *odr-3p::CP450::mCherry* (Saheki & Bargmann, 2009) in the FseI and AscI sites, to generate *itr-1pB::CP450::mCherry*. cDNA corresponding to splitGFP1-10 (see below) was PCR amplified and ligated into *itr-1pB::CP450::mCherry* in the XhoI and EcoRI sites, using the following primer set, splitGFP1-10_XhoI_F and splitGFP1-10_EcoRI_R, to generate *itr-1pB::CP450::splitGFP1-10*.

cDNA encoding codon optimized splitGFP1-10 with synthetic introns (lower case) for *C. elegans* expression:

ATGTCCAAGGGAGAGGAGCTTTTCACCGGAGTCGTCCCAATCCTTGTCGA GCTTGACGGAGACGTCAACGGACACAAGTTCTCCGTCCGCGGAGAGGGAGA GGGAGACGCCACCATCGGAAAGCTTACCCTTAAGTTCATCTGCACCACCGGA AAGCTTCCAGTCCCATGGCCAACCCTTGTCACCACCCTTACCTACGGAGTCCA ATGCTTCTCCCGCTACCCAGACCACATGAAGCGCCACGACTTCTTCAAGTCCG CCATGCCAGAGGGATACGTCCAAGAGCGCACCATCTCCTTCAAGgtaagtttaa acatatatatactaactactgattatttaaattttcagGACGACGGAAAGTACAAGACCC GCGCCGTCGTCAAGTTCGAGGGAGACACCCTTGTCAACCGCATCGAGCTTAA GgtaagtttaaacagttcggtactaactaaccatacatatttaaattttcagGGAACCGACT TCAAGGAGGACGGAAACATCCTTGGACACAAGCTTGAGTACAACTTCAACTC CCACAACGTCTACATCACCGCCGACAAGCAGAAGAACGGAATCAAGGCCAAC TTCACCgtaagtttaaacatgatttttactaactaactaatctgatttaaattttcagGTCCGC CACAACGTCGAGGACGGATCCGTCCAACTTGCCGACCACTACCAGCAAAACA CCCCAATCGGAGACGGACCAGTCCTTCTTCCAGACAACCACTACCTTTCCACC CAAACCGTCCTTTCCAAGGACCCAAACGAGAAG.

**Cloning of *itr-1pB::splitGFP11::3xPH^PLCδ1^::mCherry*** cDNA corresponding to *splitGFP11_3xPH^PLCδ1^* was PCR amplified and ligated in the NheI and KpnI sites of *itr-1pB::CP450::mCherry*, using a gBlock (IDT) and the following primer set, splitGFP11_3xPH-PLCδ1_NheI_F and splitGFP11_3xPH-PLCδ1_KpnI_R, to generate *itr-1pB::splitGFP11:: 3xPH^PLCδ1^::mCherry*.

gBlock encoding codon optimized splitGFP11_ 3xPH^PLCδ1^ for *C. elegans* expression (cDNA encoding codon optimized splitGFP11 is underlined):

<u>ATGCGAGATCATATGGTTCTGCACGAATACGTCAACGCTGCCGGCATTAC</u> <u>A</u>GGTGGCAAATTCATGTGTACAAGAGATCTtGAGCTCAAGCTTCACGGCCTGC AAGACGATCCGGATCTTCAGGCCTTGTTAAAAGGCTCGCAACTGTTAAAAGTC AAGTCATCATCTTGGCGAAGAGAGAGATTTTACAAATTACAGGAAGACTGTA AAACCATATGGCAAGAGTCTCGAAAGGTTATGAGATCCCCTGAATCCCAGCT GTTTTCGATAGAAGACATTCAAGAGGTACGTATGGGCCACCGTACTGAGGGA TTGGAAAAATTTGCTCGTGACATACCGGAGGATAGATGCTTTTCAATTGTTTT

CAAAGACCAAAGAAACACGCTTGACCTCATTGCGCCGTCGCCTGCTGATGCG CAACACTGGGTCCAGGGCCTCAGAAAAATAATACATCATTCTGGCTCGATGG ATCAGAGACAAAAGGGTGGTTCTGGTGGCCACGGCTTGCAGGACGACCCTGA CCTCCAAGCACTGTTAAAGGGCTCTCAGTTGTTAAAAGTAAAATCATCCAGTT GGAGACGAGAGAGATTTTATAAACTTCAGGAAGATTGTAAGACCATCTGGCA AGAATCTAGAAAAGTCATGCGATCGCCAGAATCTCAGCTCTTTTCGATTGAGG ATATTCAGGAAGTGAGAATGGGCCACCGAACAGAGGGATTGGAGAAATTTGC CCGAGATATTCCGGAAGATCGTTGTTTTTCAATTGTGTTTAAGGATCAAAGAA ATACCTTAGACTTGATAGCTCCTTCCCCAGCCGATGCTCAGCACTGGGTACAG GGTCTCCGAAAAATCATCCATCATTCGGGCTCTATGGACCAACGTCAAAAAG GTGGAAGTGGTGGTCATGGCCTTCAAGATGATCCGGACCTCCAGGCGTTACT CAAGGGTTCTCAATTGCTGAAAGTGAAATCGTCTAGTTGGAGAAGAGAAAGA TTCTATAAACTTCAAGAGGACTGCAAGACAATCTGGCAAGAGAGTCGTAAGG TTATGCGTTCGCCTGAGTCTCAACTGTTCAGTATCGAGGGATATCCAAGAGGTG CGAATGGGTCACCGAACCGAGGGCCTGGAGAAGTTCGCGCGAGACATCCCT GAGGATCGATGTTTCTCAATTGTGTTTAAAGATCAGCGTAATACCCTGGATCT TATTGCACCGAGTCCGGCGGATGCACAGCACTGGGTTCAAGGTCTCAGAAAG ATTATACACCATTCTGGTTCAATGGATCAAAGACAGAAGCA.

**Cloning of *itr-1pB::splitGFP11::3xPH^PLCδ1^*** cDNA corresponding to *splitGFP11_ 3xPH^PLCδ1^* was PCR amplified and ligated in the NheI and EcoRI sites of *itr-1pB::CP450::mCherry*, using a gBlock (IDT) (see above) and the following primer set, splitGFP11_3xPH-PLCδ1_NheI_F and splitGFP11_3xPH-PLCδ1 _EcoRI_R, to generate *itr-1pB::splitGFP11:: 3xPH^PLCδ1^*.

**Cloning of *itr-1pB::mNeonGreen::ESYT-2*** Genomic DNA fragment corresponding to *itr-1pB* was ligated into *esyt-2p::mNeonGreen::ESYT-2* in the FseI and AscI sites, to generate *itr-1pB::mNeonGreen::ESYT-2*.

**Cloning of *itr-1pB::wrmScarlet::ESYT-2*** cDNA corresponding to wrmScarlet (El Mouridi et al, 2017) was PCR amplified and ligated into the *itr-1pB::mNeonGreen::ESYT-2* in the AscI and EcoRI sites, using the following primer set, wrmScarlet_AscI_F and wrmScarlet_EcoRI_R, to generate *itr-1pB::wrmScarlet::ESYT-2*.

**Cloning of *itr-1pB::splitGFP11::mCherry*** cDNA corresponding to 3xPH^PLCδ1^ was deleted by site-directed mutagenesis in *itr-1pB:: splitGFP11::3xPH^PLCδ1^::mCherry*, using the following primer set, splitGFP11_mutagenesis_F and splitGFP11_mutagenesis_R, to generate *itr-1pB::splitGFP11::mCherry*.

**Cloning of *itr-1pB::FLP* and *odr-3p::FLP*** Genomic DNA fragment corresponding to *itr-1pB* was PCR amplified and ligated into the pMLS262 vector in the AvrII and FseI sites, using *C. elegans* genomic DNA as a template and the following primer set, pMLS262_itr-1pB_AvrII_F and pMLS262_itr-1pB_FseI_R, to generate *itr-1pB::FLP*.

Genomic DNA fragment corresponding to *odr-3p* was PCR amplified and ligated into the pMLS262 vector in the AvrII and FseI sites, using *C. elegans* genomic DNA as a template and the following primer set, pMLS262_odr-3p_AvrII_F and pMLS262_odr-3p_FseI_R, to generate *odr-3p::FLP*.

**Cloning of *itr-1pB::cil-1*** cDNA of *cil-1* (C50C3.7) was PCR amplified and ligated into *itr-1pB::jph-1::mCherry* in the NheI and KpnI sites, using *C. elegans* cDNA library as a template and the following primer set, cil-1_NheI_F and cil-1_stop_KpnI_R, to generate *itr-1pB::cil-1*. Stop codon was placed between *cil-1* cDNA and mCherry to prevent mCherry expression.

**Cloning of *itr-1pB::cil-1^N175A^*** The amino-acid present in the phosphatase domain of CIL-1 that was reported to be essential for its catalytic activity (N175) (Bae et al, 2009) was mutated to alanine using site-directed mutagenesis in *itr-1pB::cil-1* with the primer set, cil-1_N175A_mutagenesis_F and cil-1_N175A_mutagenesis_R, to generate *itr-1pB::cil-1^N175A^*.

**Cloning of *itr-1pB::cil-1^ΔSKICH^*** Genomic DNA fragment corresponding to *itr-1pB* was ligated into *pSM_SL2_mCherry* (Pokala et al, 2014) in the FseI and AscI sites, to generate *itr-1pB::SL2_mCherry*. cDNA corresponding to CIL-1 lacking the SKICH domain (amino acids 1-284 of CIL-1) was PCR amplified and ligated into *itr-1pB::SL2_mCherry* in the NheI and KpnI sites, using *itr-1pB::cil-1* as a template and the following primer set, cil-1_NheI_F and cil-1_ΔSKICH_Kpn1_R, to generate *itr-1pB::cil-1^ΔSKICH^*.

**Cloning of *itr-1pB::cil-1^PM^*** cDNA corresponding to the 5-phosphatase domain of *C. elegans cil-1* was PCR amplified using the following primer sets, KpnI-cil-1_S and BamHI-stop-cil-1_AS. The PCR products were then ligated at KpnI and BamHI sites in the Lck-mScarlet-I vector to generate *mScarlet-I-CIL-1^PM^*. Lck-mScarlet-I (mScarlet-I^PM^ control for mammalian expression) was a gift from Dorus Gadella (plasmid # 98821; Addgene). Genomic DNA fragment corresponding to *itr-1pB* was ligated into *odr-3::cb5::mCherry* (Saheki & Bargmann, 2009) in the FseI and AscI sites, to generate *itr-1pB::cb5::mCherry*. cDNA corresponding to Lck-mScarlet-I-CIL was then PCR amplified from mScarlet-I-CIL-1^PM^ using the following primer set, Lck_mScarlet_5ptase_CIL-1_F and Lck_mScarlet_5ptase_CIL-1_R and ligated into *itr-pB::cb5::mCherry* via the ligation-independent cloning method to generate *itr-1pB::cil-1^PM^*.

**Cloning of *itr-1pB::cil-1^ER^*** cDNA corresponding to CP450 was PCR amplified and ligated into *itr-1pB::cil-1^ΔSKICH^* in the AscI and NheI sites, using *odr-3p::CP450::mCherry* (Saheki & Bargmann, 2009) as a template and the following primer set, CP450_AscI_F and CP450_NheI_R, to generate *itr-1pB::cil-1^ER^*.

**Cloning of *itr-1pB::TRAPbeta::mNeonGreen*** cDNA of *trap-2* (T04G9.5) was PCR amplified and ligated into *itr-1pB::CP450::mNeonGreen* in the AscI and SmaI sites, using *C. elegans* cDNA library as a template and the following primer set, TRAPbeta_T04G9.5_AscI_F and TRAPbeta_T04G9.5_SmaI_R, to generate *itr-1pB::TRAPbeta::mNeonGreen*.

**Cloning of *mig-13p::split-GFP11::mCherry*** Genomic DNA corresponding to *mig-13p* was PCR amplified and ligated into *itr-1pB::splitGFP11::mCherry* in the FseI and AscI sites, using *C. elegans* genomic DNA as a template and the following primer set, mig-13p_FseI_F and mig-13p_AscI_R, to generate *mig-13p::split GFP11::mCherry*.

**Cloning of *itr-1pB::atln-1*** cDNA of *atln-1* (Y54G2A.2a) was PCR amplified and ligated into *itr-1pB::jph-1::mCherry* in the AscI and KpnI sites, using *C. elegans* cDNA library as a template and the following primer set, atln-1_AscI_F and atln-1_stop_KpnI_R, to generate *itr-1pB::atln-1*. Stop codon was placed between *atln-1* cDNA and mCherry to prevent mCherry expression.

**Cloning of *itr-1pB::atln-1^E338K^*** E338K mutation in ATLN-1 [a mutation mimicking *atln-1(yas38)*] was introduced using site-directed mutagenesis in *itr-1pB::atln-1* with the primer set, atln-1_E338K_site mutation_F and atln-1_E338K_site mutation_R, to generate *itr-1pB::atln-1^E338K^*.

**Cloning of *itr-1pB::atln-1^K80A^*** K80A mutation in ATLN-1 (a mutation that makes the GTP-binding activity of ATLN-1 defective [Liu et al, 2019]) was introduced using site-directed mutagenesis in *itr-1pB::atln-1* with the primer set atln-1_K80A_site mutation_F and atln-1_K80A_site mutation_R, to generate *itr-1pB::atln-1^K80A^*.

**Cloning of *itr-1pB::CP450::mCherry*** Genomic DNA fragment corresponding to *itr-1pB* was ligated into *odr-3::CP450::mCherry* (Saheki & Bargmann, 2009) in the FseI and AscI sites, to generate *itr-1pB::CP450::mCherry*.

**Cloning of *itr-1pB::ocrl-1^ER^*** cDNA corresponding to *ocrl-1* 5-phosphatase domain was PCR amplified and ligated into *itr-1pB::cil-1^ER^* in the NheI and XhoI sites, using *C. elegans* cDNA library as a template and the following primer set, ocrl-1_IPPc_NheI-F and ocrl-1_IPPc_XhoI-R, to generate *itr-1pB::ocrl-1^ER^*.

**Cloning of *itr-1pB::unc-26^ER^*** cDNA corresponding to *unc-26* 5-phosphatase domain was PCR amplified and ligated into *itr-1pB::cil-1^ER^* in the NheI and KpnI sites, using *C. elegans* cDNA library as a template and the following primer set, unc-26_IPPc_NheI-F and unc-26_IPPc_Kpnl-R, to generate *itr-1pB::unc-26^ER^*.

**Cloning of *itr-1pB::ocrl-1^PM^*** cDNA corresponding to Lck-mScarlet-I-OCRL-1 was PCR amplified from *mScarlet-I-OCRL-1^PM^* using the following primer set, Lck_mScarlet_5ptase_CIL-1_F and Lck_mScarlet_5ptase_OCRL-1_R and ligated into *itr-pB::cb5::mCherry* via the ligation-independent cloning method to generate *itr-1pB::ocrl-1^PM^*.

**Cloning of *itr-1pB::unc-26^PM^*** cDNA corresponding to Lck-mScarlet-I-UNC-26 was PCR amplified from *mScarlet-I-UNC-26^PM^* using the following primer set, Lck_mScarlet_5ptase_CIL-1_F and Lck_mScarlet_5ptase_UNC-26_R and ligated into *itr-pB::cb5::mCherry* via the ligation-independent cloning method to generate *itr-1pB::unc-26^PM^*.

**Cloning of *itr-1pB::mCherry::CLIMP-63*** cDNA corresponding to *CLIMP-63* was PCR amplified and ligated into *itr-1pB::mCherry::JPH-1* in the NheI and KpnI sites, using CLIMP-63 pcDNA (plasmid #80977; Addgene) as a template and the following primer set, CLIMP-63_NheI-F and CLIMP-63_KpnI-R, to generate *itr-1pB::mCherry::CLIMP-63*.

***For mammalian expression***
**Cloning of OCRL-1^ER^-EGFP and UNC-26^ER^-EGFP** cDNA corresponding to ER-targeted 5-phosphatase domain of *C. elegans ocrl-1* [*itr-1pB::ocrl-1^ER^*] or *unc-26* [*itr-1pB::unc-26^ER^*] was PCR amplified using the following primer sets, XhoI-CP450_S and ApaI-ocrl1_AS for OCRL-1^ER^; XhoI-CP450_S and ApaI-unc-26_AS for UNC-26^ER^. The PCR products were then ligated at XhoI and ApaI sites in pEGFP-N1 (EGFP control) to generate *OCRL-1^ER^-EGFP* and *UNC-26^ER^-EGFP*.

**Cloning of mScarlet-I-OCRL-1^PM^ and mScarlet-I-UNC-26^PM^** cDNA corresponding to the 5-phosphatase domain of *C. elegans ocrl-1* or *unc-26* was PCR amplified using the following primer sets, BsrGI-ocrl1_S

and BamHI-stop-ocrl1_AS for OCRL-1$^{PM}$, KpnI_unc-26_S and BglII-stop-unc-26_AS for UNC-26$^{PM}$. The PCR products were then digested with BsrGI and BamHI for OCRL-1$^{PM}$ or KpnI and BglII for UNC-26$^{PM}$ and ligated at Acc65I and BamHI sites or KpnI and BamHI sites in the Lck-mScarlet-I vector to generate *mScarlet-I-OCRL-1$^{PM}$* and *mScarlet-I-UNC-26$^{PM}$*, respectively.

### Isolation and characterization of *cil-1 (yas37)* and *atln-1 (yas38)*

A strain expressing mNeonGreen::ESYT-2 in DA9 neuron (*sybIs50*) was mutagenized with EMS according to standard protocols (Anderson, 1995). 1000 F1s were cloned into different plates, and 30–50 F2 animals from individual F1 animals were subjected to a direct visual screen under a microscope. The mutants were isolated based on the changes in mNeonGreen::ESYT-2 distribution in DA9 neuron as observed with a 40x objective on either a Leica DMi8 microscope or a Nikon Ti2 inverted microscope.

### Mapping of *cil-1* and *atln-1*

*cil-1(yas37)* and *atln-1(yas38)* were mapped to chromosome III and chromosome IV, respectively, using balancer strains (Dejima et al, 2018). Genomic DNA of the mutants were extracted and sent to GENEWIZ, Inc. for next generation sequencing. cDNA encoding CIL-1 was PCR amplified and ligated into a vector that contains DA9 specific *itr-1pB* promoter and injected at 5 ng/µl into *cil-1(yas37)* mutants; cDNA encoding ATLN-1 was PCR amplified and ligated into a vector that contains DA9 specific *itr-1pB* promoter and injected at 5 ng/µl into *atln-1(yas38)* mutants. The constructs rescued mNeonGreen::ESYT-2 localization in DA9 neurons. To identify the *cil-1* and *atln-1* mutations, the *cil-1* genomic coding region in *yas37* and *atln-1* genomic region coding *yas38* were amplified by PCR, and PCR products were sequenced.

### Cell-type specific endogenous tagging of ESYT-2 by CRISPR/cas9

A cell-specific CRISPR protocol, a SapTrap approach (Schwartz & Jorgensen, 2016), was used to endogenously tag GFP to the N terminus of ESYT-2 in specific neurons. Upon injection of FLPase, the off cassette was removed, leaving the N-terminal GFP and a single FRT site that encodes a 12-amino-acid flexible linker (GSSYSLESIGTS) in front of the endogenous ESYT-2. To achieve DA9 or AWC specific expression of endogenously tagged GFP::ESYT-2 (endoGFP::ESYT-2), FLPase was expressed under DA9-selective *itr-1pB* promoter or AWC-selective *odr-3* promoter.

### Generation of *esyt-2* knock-out strain

Knock-out allele of *esyt-2 (syb709)* was generated by SunyBiotech Corporation using CRISPR/Cas9 genome editing method. 1,902 bp (627 bp–2,528 bp) of *esyt-2* were deleted from the respective gene of N2 worms. The strain was backcrossed 6 times to N2 before use.

### Endogenous tagging of SEC-61.B with splitGFP1-10

*sec-61.B (knu560)* was generated by Knudra Transgenics. cDNA encoding splitGFP1-10 was inserted before the initiation codon of SEC-61.B to generate N-terminal splitGFP1-10-tagged SEC-61.B. The strain was backcrossed six times to N2 before use.

### Brood size assay

Brood size assay was conducted for control N2 and *atln-1 (yas38)* animals at 20°C. L4 worms of N2 and *atln-1 (yas38)* were singled out to NGM plates that had been seeded with OP50. The number of eggs that are laid and hatched were counted manually for the next three consecutive days. The number of progeny laid during the three consecutive days were plotted against strain type (Fig S3D). The average number of eggs laid and hatched for each strain per day for each strain was plotted against the day number (Fig S3E). At least eight independent plates for each strain were assayed.

### Cell culture and transfection

HeLa cells were cultured in DMEM containing 10% FBS and 1% penicillin/streptomycin at 37°C and 5% CO$_2$. Transfection of plasmids was carried out with Lipofectamine 2000 (Thermo Fisher Scientific). Cells were routinely verified as free of mycoplasma contamination at least every 2 mo, using MycoGuard Mycoplasma PCR Detection Kit (Genecopoeia). No cell lines used in this study were found in the database of commonly misidentified cell lines that is maintained by ICLAC and NCBI Biosample.

### Live imaging by fluorescence microscopy

L4 hermaphrodite animals with fluorescently tagged proteins were transferred to a glass slide and immobilized on 3% agarose pads using 2–3 µl 1 mg/µl levamisole diluted in M9 buffer. Multiple transgenic lines of each transgene were examined for fluorescent expression and localization patterns. HeLa cells were washed twice and incubated with Ca$^{2+}$ containing buffer (140 mM NaCl, 5 mM KCl, 1 mM MgCl$_2$, 10 mM Hepes, 10 mM glucose, and 2 mM CaCl$_2$, pH 7.4) before imaging. All images were captured under a 100× objective.

SDC microscopy and super-resolution SDC-structured illumination microscopy (SDC-SIM) were performed on a setup built around a Nikon Ti2 inverted microscope equipped with a Yokogawa CSU-W1 confocal spinning head, a Plan-Apo objective (100 × 1.45 NA), a back-illuminated sCMOS camera (Prime 95B; Photometrics), and a super-resolution module (Live-SR; Gataca Systems) that was based on structured illumination with optical reassignment and image processing (Roth & Heintzmann, 2016). The method, known as multifocal structured illumination microscopy (York et al, 2012), makes it possible to double the resolution and the optical sectioning capability of confocal microscopy simultaneously. The maximum resolution is 128 nm with a pixel size in super-resolution mode of 64 nm. Excitation light was provided by 488 nm/150 mW (Coherent) (for GFP/mNeonGreen) and 561 nm/100 mW (Coherent) (for mCherry/wrmScarlet/RFP/mScarlet-I) and 647-nm/125 mW (for iRFP) (power measured at optical fiber end) DPSS laser combiner (iLAS system; Gataca systems). All image acquisition and processings were controlled by MetaMorph (Molecular Device) software. Images were acquired with exposure times in the 400–500 ms range.

### Laser axotomy

Axotomy experiments were performed as described previously (Hammarlund et al, 2009). Strains were crossed into XE1931, a kind gift from Marc Hammarlund, before performing axotomy. Before

laser ablation experiments, L4 animals were immobilized without anesthetics by using 0.1 μm diameter polystyrene microsphere polybeads (Polysciences) and mounted onto 3% agarose pads on a glass slide. UV ablation was carried out on a GATACA iPulse system based on a dual galvanometer scanning system. After laser equipment calibration, DA9 neurons were cut at the posterior part of the asynaptic region using 355 nm pulsed laser. Animals were then recovered to OP50-seeded NGM plates and scored later with SDC microscopy.

## Image analysis and quantification

All images were analyzed off-line using Fiji (http://fiji.sc/wiki/index.php/Fiji). Quantification of fluorescence signals was performed using Excel (Microsoft) and Prism 7 or 8 (GraphPad Software).

### Quantification of ER-PM contacts, PM marker, and ER marker

For quantification of ER-PM contacts, PM marker and ER marker that were visualized by the split-GFP approach (extrachromosomal array for Figs 1G and S1C and integrated array for Figs 4I, 5D, 6E, and S7B), z-stack images of the side view of a DA9 neuron were taken by SDC microscopy and were projected into a single plane by maximum projection. Background fluorescence was measured in a region close to a DA9 neuron and subtracted from each maximum projected image. The background-subtracted image was then converted to a 10-bit image, and the DA9 neuron was straightened using the "Straighten" function. Regions of 40 μm in width from the start of dendrite (for dendrite) and from the start of dorsal axon (for dorsal axon) were selected for analysis. The clusters/particles containing mCherry (for PM marker) or reconstituted GFP signals (for ER-PM contacts and ER marker) above an arbitrary threshold were segmented, and area and mean fluorescent intensity of individual clusters/particles were measured using the "Analyze particles" command. The total fluorescence intensity was calculated as a sum of the product of the area and mean fluorescent intensity of individual clusters/particles, and then, the ratio of total fluorescence intensity of dorsal axon to that of dendrite was plotted.

### Quantification of endoGFP::ESYT-2

For quantification of endoGFP::ESYT-2 (Fig 2C), worms were rotated on the agar pad to orient neuronal processes at the top plane (dorsoventral axis) before SDC microscopy. For dendrite, z-stack images covering the entire dendrite region were taken. For dorsal axon, z-stack images covering the presynaptic terminals (labelled by mCherry::RAB-3) were taken. The images were then projected by maximum projection. Background fluorescence was measured and subtracted from each maximum projected image, and entire dendrite and dorsal axon were selected for analysis. The clusters/particles containing endoGFP::ESYT-2 signals above an arbitrary threshold were segmented, and area and mean fluorescent intensity of individual clusters/particles were measured using the "Analyze particles" command. The total fluorescence intensity was calculated as a sum of the product of the area and mean fluorescent intensity of individual clusters/particles and then plotted with normalization.

### Line scan analysis

For analysis of the colocalization between endoGFP::ESYT-2 and mCherry::RAB-3 puncta (synaptic vesicles) (Fig 2B) and between Reconstituted GFP (ER-PM contacts) and wrmScarlet::ESYT-2 (Fig 2F), line scan analysis was performed. A line of 50 μm in length (Fig 2B) or 25 μm in length (Fig 2F) was manually drawn along dorsal axons of DA9, and fluorescence intensity of GFP and mCherry/wrmScarlet along the manually drawn line was measured. The value was normalized by the maximum fluorescence intensity and plotted.

For analysis of the localization of TRAPβ::mNeonGreen (Fig 6K), a line of 30 μm in length was manually drawn along a dorsal axon of DA9, and fluorescence intensity of mNeonGreen along the manually drawn line was measured and plotted.

### Quantification of mNeonGreen::ESYT-2

For quantification of the fluorescence intensity of mNeonGreen::ESYT-2 along the different zones of DA9 neurons, z-stack images of the side view of a DA9 neuron were taken by SDC microscopy and projected into a single plane by maximum projection. Background fluorescence was measured in a region close to a DA9 neuron and subtracted from each maximum projected image. The background-subtracted image was then converted to a 10-bit image, and the DA9 neuron was straightened using the "Straighten" function. A region covering the entire dendrite and a region of 70 μm in width from the beginning of presynaptic terminals (marked by mCherry::RAB-3) was selected for analysis of dendrite and dorsal axon, respectively. The clusters/particles containing mNeonGreen::ESYT-2 signals above an arbitrary threshold were segmented, and area and mean fluorescent intensity of individual clusters/particles were measured using the "Analyze particles" command. The same threshold was used for all images. The total fluorescence intensity of mNeonGreen::ESYT-2 for different zones of a DA9 neuron was then calculated as a sum of the product of the area and mean fluorescent intensity of fluorescence clusters/particles (Figs 3C–E–E, 5B, S4C, S5D and F, and S6A). For some figures, the ratio of total fluorescence intensity of dorsal axon (or indicated regions of zone B for Fig S6F) to that of dendrite was plotted (Figs 3F, 4C and E, 5F, and 6C, S5G, S6F and G, and S7D). Occupancy of mNeonGreen::ESYT-2 in dendrites was quantified by measuring the length of regions in dendrites that contained mNeonGreen::ESYT-2 (Figs 3G, 5G, S5H, and S6B).

### Quantification of mCherry::RAB-3

For quantification of mCherry::RAB-3 puncta along the dorsal axon (Fig S3F), z-stack images of the side view a DA9 neuron were taken by SDC microscopy and projected into a single plane by maximum projection. Background fluorescence was measured in a region close to a DA9 neuron and subtracted from each maximum projected image. The background-subtracted image was then converted to a 10-bit image, and the DA9 neuron was straightened using the "Straighten" function. A region of dorsal axon covered by mCherry::RAB-3 puncta was selected for analysis. The clusters/particles containing mCherry::RAB-3 signals above an arbitrary threshold were segmented, and the number of clusters/particles as well as total fluorescence intensity were measured using the "Analyze particles" command.

### Quantification of cortical ER structure

For quantification of ER structure (Figs 4G, 5I, 6A, and S6C), images of the periphery of DA9 neurons, expressing mNeonGreen::ESYT-2 (as a marker for cortical ER), were taken by SDC-SIM. The clusters/particles containing mNeonGreen::ESYT-2 signals above an arbitrary threshold were segmented, and area of individual clusters/particles were measured using the "Analyze particles" command. mNeonGreen::ESYT-2 fluorescence clusters that are less than 0.242 $\mu m^2$ were considered as "tubules," and mNeonGreen::ESYT-2 fluorescence clusters that are more than 0.242 $\mu m^2$ were further analyzed based on their aspect ratio (AR) values; AR values more than 2.2 were considered as "tubules," whereas AR values less than 2.2 were considered as "sheets." The ratio of the area of cortical ER sheets to that of cortical ER tubules were then calculated and plotted.

### Quantification of ER length in dendrite

For quantification of ER length in dendrites of DA9 neurons (Fig S5A), z-stack images of the side view a DA9 neuron were taken by SDC microscopy and projected into a single plane by maximum projection. ER length was quantified by measuring the length of regions in dendrites that contained CP450::mCherry.

### Quantification of rough ER marker

For quantification of the fluorescence intensity of reconstituted GFP::SEC-61$\beta$ (Figs 6G and H and S6H) or TRAP$\beta$::mNeonGreen (Figs 6J and S6I), z-stack images of the side view of a DA9 neuron were taken by SDC microscopy and were projected into a single plane by maximum projection. Background fluorescence was measured in a region close to a DA9 cell body and subtracted from each maximum projected image. After background subtraction, a region of 20 $\mu m$ in length was manually selected for dendrite (for both reconstituted GFP::SEC-61$\beta$ and TRAP$\beta$::mNeonGreen) and dorsal axon (for reconstituted GFP::SEC-61$\beta$). DA9 cell body was manually outlined for both reconstituted GFP::SEC-61$\beta$ and TRAP$\beta$::mNeonGreen. The clusters/particles containing fluorescence signals above an arbitrary threshold were segmented, and area and mean fluorescence intensity of individual clusters/particles were measured using the "Analyze particles" command. The total fluorescence intensity was then calculated as sum of the product of the area and mean fluorescent intensity of fluorescence cluster/particles and plotted.

### Quantification of axon length after laser axotomy

For quantification of axon length after laser axotomy (Fig 7D–G), z-stack images of the side view of a laser axotomized DA9 neuron were taken by SDC microscopy and were projected into a single plane by maximum projection. The full length of regenerated DA9 axons (labelled by SL2::mCherry) of recovered worms were measured from the beginning of dorsal axons. For some worms (e.g., wild-type control animals), multiple maximum projected images were used to measure the length of regenerated axons.

### Quantification of iRFP-PH$^{PLC\delta1}$ signals at the PM

For analysis of the signal of iRFP-PH$^{PLC\delta1}$ at the PM via SDC microscopy (Fig S4E and G), line scan analysis was performed. A line of 5 $\mu m$ in length was manually drawn around the PM, and iRFP fluorescence intensity along the manually drawn line was measured. The peak intensity around the PM region was normalized with the intensity of cytoplasmic region and then plotted for quantification.

### Statistical methods

No statistical method was used to predetermine sample size, and the experiments were not randomized for live cell imaging. For comparing the mean of two groups, two-tailed unpaired $t$ test was performed. To compare the mean of multiple groups to the mean of an indicated group, one-way ANOVA followed by Dunnett's multiple comparisons test was performed. $P$-values > 0.05 were considered not significant. "n" represents the number of animals shown in graphs. All data are shown as mean ± SEM. In dot plots, each dot represents the value from a single worm (or a single cell) with the bar as the mean. At least eight animals (or cells) are scored in each experiment.

## Data Availability

The authors declare that the data supporting the findings of this study are available within the paper and its supplementary information files. Source data for figures (Figs 1G, 2B, C, and F, 3C–G, 4C, E, G, and I, 5B, D, F, G, and I, 6A, C, E, G, H, J, and K, 7D–G, S1C, S3D–F, S4C, E, and G, S5A, D, and F–H, S6A–C and F–I, and S7B and D) are provided with the paper. Other data are available from the corresponding author upon reasonable request. Reagents and strains generated for this study are available directly from the authors upon request.

## Supplementary Information

## Acknowledgements

We thank Darshini Jeyasimman, Bilge Ercan, Dylan Hong Zheng Koh, and Dennis Dharmawan for sharing reagents and discussion. This work was supported in part by the Singapore Ministry of Education Academic Research Fund Tier 2 (MOE2017-T2-2-001), a Nanyang Assistant Professorship, a Lee Kong Chian School of Medicine startup grant (LKCMedicine-SUG), and an Ageing Research Institute for Society and Education (ARISE) seed grant (ARISE/2017/7) to Y Saheki.

### Author Contributions

J Sun: data curation, formal analysis, validation, investigation, visualization, methodology, and writing—review and editing.
R Harion: data curation, formal analysis, validation, investigation, visualization, methodology, and writing—review and editing.
T Naito: data curation, formal analysis, validation, investigation, and visualization.

Y Saheki: conceptualization, resources, data curation, supervision, funding acquisition, project administration, and writing—original draft, review, and editing.

## Conflict of Interest Statement

The authors declare that they have no conflict of interest.

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
