## [Reviewer comments · Life Science Alliance]

Life Science Alliance

INPP5K and Atlastin-1 maintain the nonuniform distribution of ER-plasma membrane contacts in neurons

Jingbo Sun, Raihanah Harion, Tomoki Naito, and Yasunori Saheki

DOI: <https://doi.org/10.26508/lsa.202101092>

Corresponding author(s): Yasunori Saheki, Nanyang Technological University

Review Timeline:

Submission Date:	2021-04-12
Editorial Decision:	2021-05-19
Revision Received:	2021-08-19
Editorial Decision:	2021-09-02
Revision Received:	2021-09-03
Accepted:	2021-09-07

Transaction Report:

Please note that the manuscript was reviewed at *Review Commons* and these reports were taken into account in the decision-making process at *Life Science Alliance*.

May 19, 2021

Re: Life Science Alliance manuscript #LSA-2021-01092-T

Yasunori Saheki
Nanyang Technological University
Lee Kong Chian School of Medicine
Singapore 308232
Singapore

Dear Dr. Saheki,

Thank you for submitting your manuscript entitled "INPP5K and Atlastin-1 maintain the non-uniform distribution of endoplasmic reticulum-plasma membrane contacts in neurons" to Life Science Alliance (LSA).

For a brief overview, the manuscript was reviewed at Review Commons and the authors then transferred a text-revised version of the manuscript along with reviewers' comments, and a point-by-point rebuttal to the reviewers' comments, which included a revision plan to address most of the reviewers' points and a rebuttal / clarification to the concerns that were out of scope. LSA editors assessed the information and found the proposed revision plan to be appropriate.

We would, thus, encourage you to submit a revised manuscript including the additional experiments that have been mentioned in the pbp rebuttal. The revised manuscript will have to be re-reviewed, preferably by some or all of the same reviewers. We will notify them of the points that we overruled from their review.

Thank you for this interesting contribution to Life Science Alliance. We are looking forward to receiving your revised manuscript.

Sincerely,

Shachi Bhatt, Ph.D.
Executive Editor
Life Science Alliance
<http://www.lsajournal.org>
Tweet @SciBhatt @LSAJournal

- A letter addressing the reviewers' comments point by point.
- An editable version of the final text (.DOC or .DOCX) is needed for copyediting (no PDFs).
- High-resolution figure, supplementary figure and video files uploaded as individual files: See our detailed guidelines for preparing your production-ready images, <https://www.life-science-alliance.org/authors>
- Summary blurb (enter in submission system): A short text summarizing in a single sentence the study (max. 200 characters including spaces). This text is used in conjunction with the titles of papers, hence should be informative and complementary to the title and running title. It should describe the context and significance of the findings for a general readership; it should be written in the present tense and refer to the work in the third person. Author names should not be mentioned.

B. MANUSCRIPT ORGANIZATION AND FORMATTING:

Life Science Alliance manuscript #LSA-2021-01092-T

REBUTTAL TO THE COMMENTS RAISED BY THE REVIEWERS

We thank the editor and the reviewers for the many constructive comments and suggestions. We have carried out extensive new experimentation to address all the concerns and made necessary changes to the texts.

We have now

- 1) shown that there are no significant changes in the distribution of mNeonGreen-tagged ESYT-2-labelled or split GFP-labelled ER-PM contacts in axons or in dendrites after laser-induced axotomy (for Reviewer #1)
- 2) provided evidence that CLIMP-63 overexpression induced expansion of cortical ER sheets in DA9 cell bodies similar to *cil-1*, *atln-1*, and *ret-1* mutants and increased presence of mNeonGreen::ESYT-2 in ventral processes (for Reviewers #2 and #3)
- 3) characterized the activities of PM-targeted and ER-targeted 5 phosphatase domains of OCRL-1 and UNC-26 in HeLa cells and provided evidence that they act primarily *in cis* to modulate the levels of PI(4,5)P₂ in the cell (for Reviewer #3)

A specific list of changes is indicated below, followed by a point-by-point rebuttal from page 2.

Fig S4 and **Fig S6** are updated with additional data.

New Fig S7 is added with newly generated data.

We now have seven main figures (**Fig 1-7**) and seven supplementary figures (**Fig S1-7**).

The following data are new:

Figs S4D-G [showing the efficient depletion of PM PI(4,5)P₂, as assessed by the PH domain of PLCδ1 (iRFP-PH PLCδ1), by the PM-targeted 5 phosphatase domains but not by the ER-targeted 5 phosphatase domains]

Fig S6C [showing the expansion of mNeonGreen::ESYT-2-labelled cortical ER sheets in DA9 cell bodies in animals overexpressing CLIMP-63]

Fig S6D-G [showing the increased presence of mNeonGreen::ESYT-2 in ventral and commissure axon (Zone B) upon CLIMP-63 overexpression]

Fig S7A and S7B [showing the absence of detectable changes in the distribution of ER-PM contacts in axons or in dendrites 24 hours post laser axotomy]

Fig S7C and S7D [showing the absence of detectable changes in the distribution of mNeonGreen::ESYT-2 in axons or in dendrites 24 hours post laser axotomy]

The following *C. elegans* strains were newly generated for the revision:

- SAH656: *sybIs2669 X; ric-7(n2657) V*
- SAH666: *sybIs50 X; ric-7 (n2657) V; yasEx241 [itr-1pB::mCherry (50ng/ul)]; pCFJ421 - myo-2p::GFP::H2B (10ng/ul)]*
- SAH667: *yasEx232 [itr-1pB::mCherry::CLIMP63 (20ng/ul)]; pCFJ421 - Pmyo-2::GFP::H2B (5ng/ul)]; sybIs50 X*
- SAH675: *cil-1(yas37)/qC1 [dpy-19(e1259) glp-1(q339)] nls189 III; sybIs50 X; yasEx238 [itr-1pB::cil-1 (PM) (5ng/ul)]; elt-2p::mCherry (10ng/ul)]*
- SAH676: *cil-1(yas37)/qC1 [dpy-19(e1259) glp-1(q339)] nls189 III; sybIs50 X; yasEx239 [itr-1pB::unc-26 (PM) (5ng/ul)]; elt-2p::mCherry (10ng/ul)]*
- SAH677: *cil-1(yas37)/qC1 [dpy-19(e1259) glp-1(q339)] nls189 III; sybIs50 X; yasEx240 [itr-1pB::ocrl-1 (PM) (5ng/ul)]; elt-2p::mCherry (10ng/ul)]*

-Sentences from the reviewers' comments are *in italics*

-Our responses are in blue

Reviewer #1 (Evidence, reproducibility and clarity (Required)):

*In this article, the authors used the nematode *C. elegans* to characterize molecular and cellular mechanisms controlling formation and stability of ER-plasma membrane contact sites. One main assay in their work is based on split GFP, with tandem pleckstrin homology (PH) domains of phospholipase C (PLC) $\delta 1$ at PM and ER resident protein CP450, as a way to visualize the distribution of neuronal ER-PM contacts in vivo by fluorescence microscopy. The main findings are the following:*

- 1. ER-PM contacts are non-uniformly distributed, mostly in somatodendritic regions, with very few ER-PM contacts in the dorsal axon, in mature neuron and ESYT-2 localizes to these ER-PM contacts,*
- 2. using this observation to design and perform a forward genetic screen to identify mutants that mislocalized ER-PM contacts to the dorsal axon, they isolated mutations in *atln-1* and *cil-1*, which encode homologs of mammalian *Atlastin-1* and inositol 5-phosphatase *INPP5K*, respectively.*
- 3. cortical ER sheets expand and rough ER proteins localize ectopically to axonal ER in these mutants,*
- 4. *CIL-1* and *ATLN-1* have related functions, with *CIL-1* being upstream *ATLN-1*, to maintain the balance between tubules and sheets at the cortical ER as well as to restrict the ER sheets to dendrites,*
- 5. mutants that lack *RET-1*, reticulon homolog in *C. elegans*, whose depletion is known to result in ER sheet expansion, phenocopy the phenotypes of *CIL-1* and *ATLN-1*,*
- 6. *CIL-1* and *ATLN-1* are both required for efficient axon regeneration following laser-axotomy.*

The experiments were rigorously designed and performed, the statistical analysis is correct, the claims and conclusions of the authors are well based on their results.

We thank the reviewer for these very nice comments on our manuscript.

I only have one request which is to quantify ER-PM contacts and characterize the localization of E-Syt2 in regenerating axons in order to get a clear insight into the contribution of ER-PM contact sites in axonal regeneration thus link the final piece of data with the rest of the article. Here the authors inferred that the same defects seen in mature would occur in regenerating axons but this requires direct investigation. Given the tools and skills of the authors, this request seems achievable in a limited amount of time, possibly by analyzing already available data.

We thank the reviewer for this excellent suggestion. We generated new strains to observe ESYT-2 (via mNeonGreen-tagged ESYT-2) and ER-PM contacts (via the split GFP approach) during axon regeneration after laser-induced axotomy. The distribution of ER-PM contacts was assessed 24 hours after axotomy. We did not detect significant changes in the distribution of ER-PM contacts in axons or in dendrites (new Fig S7A-S7D). These results indicate that the overall distribution of neuronal ER-PM contacts is maintained during axon regeneration after laser-induced injury.

Reviewer #1 (Significance (Required)):

This article presents major conceptual advance because it identifies two new crucial actors of neuronal ER-PM contact sites and show their role in axonal regeneration. These new

results allow linking the balance between ER tubules and sheets and ER-PM contacts. They further identified an important role of *Atlastin-1* and *INPP5K* in axonal regeneration and because these genes were associated to degenerative disease, this piece of work provides an important pathophysiological hypothesis.

We thank this reviewer for these very positive comments on our manuscript.

****Referees cross-commenting****

To follow-up on Rev#2 comment: they focused on ER-PM contact sites because that is the essence of the split-GFP assay they used for their screen.

We fully agree with this reviewer for this point. We focused on ER-PM contacts in this manuscript because we isolated mutants based on defects in the distribution of ER-PM contacts. In brief, we mutagenized SAH553, a strain that expresses mNeonGreen-tagged ESYT-2 [a sole *C. elegans* homolog of E-Syts, well-established evolutionary conserved ER-PM tethers] under the control of DA9 neuron-specific promoter and identified mutants that showed altered distribution of ESYT-2 in DA9 neuron [i.e., *cil-1(yas37)* and *atln-1(yas38)* mutants]. We further validated the altered distribution of ER-PM contacts in these mutants by crossing *cil-1(yas37)* and *atln-1(yas38)* mutants into SAH526, a strain that expresses components of split GFP to label ER-PM contacts, under the control of DA9 neuron-specific promoter.

*I think Rev#2 criticisms mainly relate to the fact that the effects on ER-PM could be indirect given what is known on *Atlastin* and *INPP5K*. I would agree with that point but rather see it as one requesting to be discussed. I still think the data are solid.*

We agree with this reviewer for this comment. We addressed reviewer 2's concerns by discussing such possibility more clearly in discussion (page 16 of our revised manuscript).

Old sentence: "Our study strongly suggests that the balance between ER sheets and tubules determines where ER-PM contacts are distributed in neurons."

New sentence: "Although how exactly CIL-1/INPP5K and ATL-1/Atlastin control the distribution of neuronal ER-PM contacts remains elusive, our study suggests that the balance between ER sheets and tubules may determine where ER-PM contacts are distributed in neurons based on the reported functions of INPP5K and Atlastin to regulate ER morphologies (see below)."

*Rev#3 criticisms: in relation with Petkovic et al, it could be that the key point is the role of *Atl* in tubular ER. This needs to be discussed.*

We appreciate the reviewer for this comment. We think that the reviewer is referring to the reviewer 3's statement "The finding that contact sites are depleted in axons and inhibit axonal regeneration are a bit surprising and somewhat contradictory to other studies such as Petkovic et al NCB 2015 which suggested lipid delivery at contact sites promote rather than inhibit neurite outgrowth." To be clear, we showed that *atln-1* and *cil-1* mutants show "ectopic formation" of ER-PM contacts in axon rather than "depletion". We then found that these mutants are defective in axon regeneration following laser-induced axotomy.

We believe that proper compositions of ER-PM contacts rather than their abundance (i.e., quality vs. quantity) is important for regeneration or outgrowth of axons. As this reviewer pointed out, the ER axonal morphology (i.e., tubular vs. sheet) in axons,

controlled by ATLN-1, may also play an important role in axon regeneration. We added the following new sentences in discussion to discuss these possibilities (page 17 of our revised manuscript).

New sentences: "Mutants lacking CIL-1 or ATLN-1 show ectopic formation of axonal ER-PM contacts. Petkovic et al., suggested that lipid delivery at neuronal ER-PM contacts promote neurite outgrowth (Petkovic et al., 2014). Thus, proper compositions of ER-PM contacts rather than their abundance maybe important for regeneration or outgrowth of axons. Our study does not rule out the possibility that the axonal ER morphology (i.e., tubular vs. sheet) and motility as well as other related functions of the axonal ER, such as maintenance of Ca²⁺ homeostasis and regulation of the functions of various other organelle contact sites, may also play a role in proper axon regeneration."

I mostly agree with Rev#3 comments but still would like to point out that, despite our best efforts in biology, all we do is correlate events and deduce hypothetical mechanisms which withstand only based upon testing further assumptions made on their basis. I support publication with major textual changes to answer our requests and additional experiments when it is clearly needed. I would stand by my request on ESyt2.

We appreciate this reviewer for these very thoughtful comments. We have performed major textual changes according to reviewers' suggestions and performed specific experiments as outlined in this letter.

Reviewer's expertise: Role of SNAREs in membrane trafficking and ER-contact sites during neuronal differentiation.

Reviewer #2 (Evidence, reproducibility and clarity (Required)):

Here Sun and colleagues study the effect on ER structure of mutations of INPP5K and Atlastin-1, using a neuronal model system in worms. They show that INPP5K lies upstream of Atlastin-1 in regulating ER structure and they indicate that there is a differential effect of the mutations, with increased entry of ER into axons rather than dendrites.

We thank the reviewer for all the constructive comments to improve our manuscript.

The main claim as expressed in the title is that the INPP5K/Atlastin axis has specific effects on ER-PM contact rather than on the ER in general and its many functions. The main claim is therefore a quite specific, and would represent conceptual advance of some interest if it could be supported by evidence.

Our title “INPP5K and Atlastin-1 maintain the non-uniform distribution of endoplasmic reticulum-plasma membrane contacts in neurons” does not necessarily indicate that these proteins have specific effects on ER-PM contacts. We agree with this reviewer and other reviewers that INPP5K and Atlastin-1 regulate ER morphology and its related functions; we did not intend to ignore these functions in our manuscript.

It was known before that INPP5K, which lysates inositides, in particular PI45P2, is required for normal tubular ER morphology, but not how the inositide lipid(s) interacts with this process - e.g. no specific target. How Atlastins regulate tubular ER morphology has also been studied for a long time, with some mechanistic insight, including in vitro reconstitutions, lipid dependencies. Thus, finding and validating these hits in the screen is a good corroboration of what was known, but not much of an advance.

What is new is to find a link between INPP5K and Atlastin.

We agree with this reviewer that both ATLN-1/Atlastins and CIL-1/INPP5K have been shown to regulate the proper ER morphology. In our study, we found that depletion of these proteins in neurons affect the distribution of neuronal ER-PM contacts (through an unbiased forward genetic screen). We also identified a genetic link between ATLN-1/Atlastins and CIL-1/INPP5K, which had not been shown before.

A major problem is showing which of many possible aspects of Atlastin function on ER architecture is the key one for axonal growth/health. Changing the balance of ER architecture may have many downstream effects, including contact with other compartments (mitochondria, endosomes, peroxisomes etc), as well as more generic activities including motility, calcium storage (which is much affected by contacts with PM, mitochondria and end-/lyso-somes). Formation of ER-PM contact may be affected indirectly and may not be the key feature for axonal health.

We thank the reviewer for these comments. We have addressed them by discussing the possibility that the axonal ER morphology and its other related functions, which are potentially regulated by ATLN-1/Atlastins and CIL-1/INPP5K functions, may also play a role in proper axon regeneration by adding the following new sentences in discussion (page 17 of our revised manuscript).

New sentences: “Our study does not rule out the possibility that the axonal ER morphology (i.e., tubular vs. sheet) and motility as well as other related functions of the axonal ER, such as maintenance of Ca²⁺ homeostasis and regulation of the functions of various other organelle contact sites, may also play a role in proper axon regeneration.”

Thus, the manuscript's direct linking of the INPP5K-Atlastin axis to ER-PM contacts seems unwarranted, which significantly reduced the significance and claim to novelty of the manuscript.

We found, through an unbiased forward genetic screen, that depletion of ATLN-1/Atlastins and CIL-1/INPP5K affects the distribution of neuronal ER-PM contacts. Detailed analysis of the mutants lacking ATLN-1 and/or CIL-1 show that these proteins are involved in maintaining axonal ER morphology of *C. elegans* neurons (similar to what has been previously shown in other cell types and in other model systems as we discussed throughout our manuscript), which may contribute, at least in part, to the maintenance of the distribution of neuronal ER-PM contacts. We believe that our finding is an important step in understanding how a complex cell, like neuron, maintains the distribution of ER-PM contacts.

An example of where the claims are excessive is on page 9 "Collectively, these results reveal the importance of ER shape in maintaining the distribution of neuronal ER-PM contact". There is no way to establish that shape is the causative effect here. Atlastin and INPP5K have been associated with growing/moving parts of the ER network, so it may be not shape but something else about INPP5K/Atlastin.

We appreciate the reviewer's comments on this sentence. We toned it down as shown below (page 10 of our revised manuscript).

Old sentence: "Collectively, these results reveal the importance of ER shape in maintaining the distribution of neuronal ER-PM contacts."

New sentence: "Collectively, these results suggest that ER shape may play a role in maintaining the distribution of neuronal ER-PM contacts."

We also toned down our original statement in discussion (page 16 of our revised manuscript).

Old sentence: "Our study strongly suggests that the balance between ER sheets and tubules determines where ER-PM contacts are distributed in neurons."

New sentence: "Although how exactly CIL-1/INPP5K and ATLN-1/Atlastin control the distribution of neuronal ER-PM contacts remains elusive, our study suggests that the balance between ER sheets and tubules may determine where ER-PM contacts are distributed in neurons based on the reported functions of INPP5K and Atlastin to regulate ER morphologies (see below)."

A second, validating way to make ER sheets should be tested to see if it produces parallel results for example over-expressing CLIMP (Gao et al., PLoS Biol 2019, PMID: 31469817).

We thank the reviewer for this suggestion. We overexpressed mammalian CLIMP-63 (there is no homolog of CLIMP-63 in *C. elegans*) in DA9 neurons of wildtype worms and examined whether this would cause any changes in the distribution of mNeonGreen::ESYT-2 in DA9 neurons. As expected, we found CLIMP-63 overexpression induced expansion of cortical ER sheets in DA9 cell bodies similar to *cil-1*, *atln-1*, and *ret-1* mutants (**new Fig S6C**). Notably, this was accompanied by altered distribution of mNeonGreen::ESYT-2 in DA9 axons; mNeonGreen::ESYT-2 became more abundant in the commissure region of DA9 and less restricted to the somatodendritic region of DA9 compared to wildtype control (**new Fig S6D-S6F**). These results suggest that the abundance of cortical ER sheets affects the

distribution of ER-PM contacts in DA9 neurons. However, we could not detect the increased presence of mNeonGreen::ESYT-2 in the dorsal axons of DA9 in this condition (**new Fig S6G**). We think this could be due to the limited effect of CLIMP-63 overexpression in ER structures in dorsal axons compared to those in regions proximal to the cell bodies [e.g., CLIMP-63 was shown to localize preferentially to somatodendritic regions in mammalian neurons (PMID: 30772082)].

Overall, the key conclusions are not convincing. Claims linked to ER-PM are often speculative. In addition, the molecular mechanism linking INPP5K to atlastin is missing. The manuscript could improve by filling in the three major gaps in the current story:

(a) work out how INPP5K affects Atlastin-1

(b) work out how Atlastin-1 affects ER entry into cell protrusions, and determine what the difference is (if any) between dendrites and the axon

(c) work out what the specific role is for ER-PM contact compared to other ER functions that might be perturbed by INPP5K/Atlastin mutations.

Sorting this out could take a long time.

Additional experiments would be needed to specifically test whether ER-PM contact alone lies behind the effects on axonal function

We acknowledge that all these points are interesting, but we believe that they are beyond the scope of our manuscript. We would like to quote the comments from two other reviewers of this manuscript (reviewer 1 and reviewer 3) for these points from "Referees cross-commenting".

Reviewer 1:

"I think Rev#2 criticisms mainly relate to the fact that the effects on ER-PM could be indirect given what is known on Atlastin and INPP5K. I would agree with that point but rather see it as one requesting to be discussed. I still think the data are solid."

"I mostly agree with Rev#3 comments but still would like to point out that, despite our best efforts in biology, all we do is correlate events and deduce hypothetical mechanisms which withstand only based upon testing further assumptions made on their basis. I support publication with major textual changes to answer our requests and additional experiments when it is clearly needed."

Reviewer 3:

"I agree with Rev#1 that a substantial amount of work of good quality has been performed. The authors show many correlations (INPP5K, ATL, RET-1, ablation with ER-PM contact deregulation; INPP5K, ATL ablation with axonal regrowth defect) which in my opinion are still worthwhile, interesting observations that should be published."

"Rev#2 points out 3 interesting questions the data raise that are unanswered. I agree these are the interesting questions the data raise. But if it would take another year or two to answer, perhaps it would be worth it to publish the data they have now, just adjusting the wording to admit to the questions unanswered."

We agree with the reviewer 1 and the reviewer 3, regarding major concerns raised by the reviewer 2. Thus, we performed textual changes to address them (please see above and below for individual textual changes that we have made in the revised manuscript).

****Issues****

****Major****

"We found Atlastin-1 and INPP5K (ATLN-1 and CIL-1 in C. elegans) ... function together to maintain the non-uniform distribution of neuronal ER-PM contacts. " This should d be toned down to clarify that no direct link from INPP5K and Atlastin to distributing ER-PM contacts is established. However, the effect could be indirect via many steps that lead to altered movement into both dendrites and axons, and then the pathway from this altered distribution to altered contact with the PM is as yet untested, and is possibly irrelevant.

We thank the reviewer for this comment. We toned down our claims by changing the relevant sentences (page 4 of our revised manuscript) as shown below.

Old sentences: "We found Atlastin-1 and INPP5K (ATLN-1 and CIL-1 in *C. elegans*) that have been linked to hereditary spastic paraplegia and intellectual disability function together to maintain the non-uniform distribution of neuronal ER-PM contacts. CIL-1 functions upstream of ATLN-1 to prevent expansion and invasion of cortical ER sheets into the axon."

New sentences: "We found Atlastin-1 and INPP5K (ATLN-1 and CIL-1 in *C. elegans*) that have been linked to hereditary spastic paraplegia and intellectual disability function together to prevent expansion and invasion of cortical ER sheets into the axon. In their absence, the non-uniform distribution of neuronal ER-PM contacts is disrupted."

"Our results support the important role of the unique distribution of neuronal ER-PM contacts in neuronal resilience following a damaging insult. " This implies that restoration of just ER-PM contact rescues the absence of INPP5K/Atlastin. However, experiments here that show ATLN-1 over-expression rescues cil-1(yas37) mutants do not show that this is via ER-PM contact.

We appreciate the reviewer's comments. However, we do not fully agree with these points. We showed that overexpression of ATLN-1 in *cil-1 (yas37)* mutants restored both ER morphology (Fig 5H and 5I) and the distribution of ESYT-2-mediated ER-PM contacts in DA9 neuron (Fig 5E and 5F). This restoration was correlated with the rescue of axon regeneration defects in *cil-1 (yas37)* mutant (Fig 7F and 7G). Although it remains as correlation, these results at least partially suggest that restored ER-PM contacts might have contributed to the rescue of axon regeneration defects in *cil-1 (yas37)* mutants. To address the reviewer's concern, we changed the word from "support" to "suggest" in the relevant sentence (page 4 of our revised manuscript) to tone down our statement.

The link from INPP5K/Atlastin to contacts is the weakest part of this. It is based on experiments using a strain lacking the sole worm reticulon (RET-1). This may look simple, with a simplified reticulon biology compared to humans (4 RTN genes) and yeast (2 Rtns + Yop1), but it is hard to interpret. RET-1 has seven splice forms from 200 to 3300 aa length. These have been studied very little (only Pubmed hit Iwahashi et al 2002, 12054525). How many ways does the RET-1 mutation affect the strain? What are the unique interactions of each reticulon isoform (paralleling the specific interaction of human RTN3, Wu and Voeltz 2021, 33434526)? And which are lost? Without knowing any of this, the experiment in Fig 6A-E is largely invalid.

We do not agree with this reviewer that our experiments (Fig 6A-6E) are invalid. Our results shown in Fig 5H and Fig 5I suggested that the expansion of cortical ER sheets, as observed in *cil-1* and *atln-1* mutants, potentially underlies ectopic formation of ER-PM contacts. We wanted to test this hypothesis by using another

evolutionarily conserved and well-characterized ER-shaping protein, RET-1, whose absence is known to cause major expansion of cortical ER sheets in yeast (e.g., PMID: 21502358; PMID: 16469703; PMID: 16624861). As predicted based on these yeast studies, we found that *C. elegans ret-1* mutants also showed expansion of cortical ER sheets (Fig 6A), similar to *cil-1* and *atln-1* mutants (Fig 5H and 5I). The similarity of the phenotype between *ret-1*, *cil-1*, *atln-1* mutants (i.e., expansion of cortical ER sheets and ectopic formation of ER-PM contacts in axon) partially demonstrates that cortical ER shape plays an important role in maintaining the distribution of neuronal ER-PM contacts.

We would like to quote the reviewer 3's comments from "Referees cross-commenting", supporting the validity of our experiments.

Reviewer 3 stated "For me, I disagree for example to call the RET-1 experiment invalid, because the experiment was done in a correct way. We can conclude there is a correlation between RET-1 ablation and ER-PM contact differences even if we don't understand the phenotype entirely, meaning why this is so, but that it does happen I believe is clearly shown. What is invalid is the conclusion that this proves that ER shape regulates ER-PM contact distribution - with this I agree. The data support this hypothesis, but do not prove it. But to me it doesn't mean the data don't deserve to be out there, published."

As the reviewer 3 mentioned above, we believe that there is a correlation between ER shape and ER-PM contact distribution based on our results. We performed textual changes to tone down our statements regarding the interpretation of these results (pages 11-12 of our revised manuscript).

Old sentence: "Our results suggested that cortical ER shape is essential for maintaining the distribution of neuronal ER-PM contacts."

New sentence: "Our results suggested that cortical ER shape is potentially important for maintaining the distribution of neuronal ER-PM contacts." (page 11)

Old sentence: "Thus, cortical ER shape, and more specifically the proper balance of ER tubules and sheets, is essential for restricting the distribution of neuronal ER-PM contacts to specific regions within neurons (namely the somatodendritic region)."

New sentence: "Taken together, these data support the notion that cortical ER shape, and more specifically the proper balance of ER tubules and sheets, play a role in restricting the distribution of neuronal ER-PM contacts to specific regions within neurons." (page 12)

In Fig 6F-K, the results are summarised with a heavy emphasis on invasion into the dorsal axon, but the data shows equal invasion into the dendrite. This undercuts the message of much of the manuscript's current focus on the INPP5K/ATLN axis being required to main the normal state of dendrites being rich in ER but the axon poor.

We would like to clarify these points. Our results do not indicate "normal state of dendrites being rich in ER but the axon poor" as this reviewer stated. Using CP450 as a general ER marker (e.g., PMID: 19718034; PMID: 12006669), we showed in Fig S1 that the ER itself is present both in dendrite and in axon. We stated in the page 6 of our original manuscript "...uniform distribution of GFP fluorescence throughout all neuronal processes of the DA9 neuron, demonstrating that CP450 is distributed throughout neuronal processes as a general ER marker (thus ruling out the biased

enrichment of CP450 in the dendrite or ventral axon) (Figures S1A-S1D)". Thus, we did not claim anywhere in our original manuscript that dendrites are rich in the ER itself.

What we found was that rough ER proteins (visualized by endogenously-tagged SEC-61 β and overexpressed TRAP β ::mNeonGreen) were almost non-existent in axons compared to dendrites in wild-type *C. elegans* DA9 neurons (Fig 6F and 6I). In *cil-1*, *atln-1* and *ret-1* mutants, however, we observed significant presence of SEC-61 β and TRAP β in axons compared to wild-type animals. Because rough ER proteins are primarily present in ER sheets, we interpreted the elevated levels of these proteins in axons as a sign of expansion/invasion of ER sheets into axons [which may underlie the increased abundance of cortical ER sheets (i.e., ER-PM contacts) in these mutants].

We agree that the levels of SEC-61 β and TRAP β were also elevated in dendrites of *cil-1*, *atln-1* and *ret-1* mutants. Because ER-PM contacts are already abundant in dendrites in wild-type animals (as we showed in Fig 1E-1G), we could not detect further increase in these contacts in dendrites.

Fig 7 starts by repeating previous work on junctophilin, which has the domain arrangement to bridge from ER to PM and no other apparent molecular function. While axonal regrowth after cutting in DA9 neurons is reduced without junctophilin, INPP5K and ATLN-1 mutants cause more severe defects and ESYT-2 mutation has no effect. While these are interesting results, they are interpreted in an odd and very specific way. ATLN-1 has a well-known role in ER motility, so it is a simple idea that its loss would reduce ER motility say near the axonal growth tip, or reduced interaction with key organelles near there (mitochondria, endosomes etc.) contributing to reduced regrowth of the axon. It is not clear what drives the authors to only focus on ER-PM contact. Indeed, the finding that INPP5K and ATLN-1 mutants cause more severe defects than the junctophilin mutant is consistent with them affecting different pathways.

The final experiment is reported as "striking", but it was already established above that overexpression of ATLN-1 rescues INPP5K, and in no way has it been shown how ATLN-1 directly affects ER-PM contact, so Fig 7F-G do not test the hypothesis stated.

We appreciate the reviewer's comments. However, as we stated above, we do not fully agree with these points. We showed that overexpression of ATLN-1 in *cil-1* (*yas37*) mutants restored both ER morphology (Fig 5H) and the distribution of ESYT-2-mediated ER-PM contacts in DA9 neuron (Fig 5E and 5F). This restoration was correlated with the rescue of axon regeneration defects in *cil-1* (*yas37*) mutant (Fig 7F and 7G). Although it remains as correlation, these results at least partially suggest that restored ER-PM contacts might have contributed to the rescue of axon regeneration defects in *cil-1* (*yas37*) mutants.

To address the reviewer's concern, we removed "strikingly" from the relevant sentence to tone down on our statement (page 12 of our revised manuscript). As we stated above (on the page 4 of this letter), we also discussed in our revised manuscript the possibility that the axonal ER morphology and its other related functions, which are potentially regulated by ATLN-1/Atlastins and CIL-1/INPP5K functions, may also play a role in proper axon regeneration by adding the following sentences in discussion (page 17 of our revised manuscript).

New sentences: "Our study does not rule out the possibility that the axonal ER morphology (i.e., tubular vs. sheet) and motility as well as other related functions of the axonal ER, such as maintenance of Ca²⁺ homeostasis and regulation of the

functions of various other organelle contact sites, may also play a role in proper axon regeneration.”

****Minor****

I expected to see a detailed comparison between this work and previous work showing pleiotropic effects of ATL3 disease mutations, including: reduced ER complexity in neurons, and the mutated protein not entering growing axons (Behrendt et al., CMLS 2019, PMID: 30666337).

Thank you for sharing this information. We cited this paper and other related papers in discussion to mention pleiotropic effects of ATL3 disease mutations. We added the following sentence to the discussion (page 17-18 of our revised manuscript).

New sentences: “More recently, autosomal-dominant missense mutations in ATL3 were reported to cause hereditary sensory and autonomic neuropathy (HSAN) (Fischer et al., 2014; Kornak et al., 2014). These mutations disrupt fusion of the ER and induce aberrant ER tethering, causing major changes in ER morphology. Interestingly, cells carrying a disease-causing ATL3 mutation show pleiotropic defects in ER-related functions, including defects in vesicular transport and autophagosome formation (Behrendt et al., 2019; Krols et al., 2019; Krols et al., 2018). Our study shows that CIL-1/INPP5K and ATL3/Atlastin-1 help to maintain the non-uniform distribution of neuronal ER-PM contacts. These proteins likely contribute to the maintenance of the unique distribution of neuronal ER-PM contacts through their properties to control ER morphology, although we do not exclude the possibility that they control the distribution of ER-PM contacts indirectly through the regulation of other ER-related functions.”

Do the split GFP labelled contacts in DA9 neurons cause issues with over-strong ER-PM interactions?

Yes, there is a possibility that expressing the split GFP system might cause stronger ER-PM interactions. It is a limitation of this approach. Thus, we used ESYT-2 (a well-established ER-PM tether) as another marker protein of ER-PM contacts to complement our observation. We stated in the page 7 of our original manuscript “To further confirm the non-uniform distribution of ER-PM contacts that we observed with the split GFP approach in DA9 neuron, we visualized the endogenous localization of ESYT-2 using a cell-type specific endogenous labeling approach (Schwartz and Jorgensen, 2016).” to introduce this complementary approach.

We also added the following new sentences in the page 7 of our revised manuscript to further discuss the limitation of this approach.

New sentences: “The split GFP approach mostly labels ER-PM contacts that depend on the presence of PM PI(4,5)P₂ [as mCherry::3xPH^{PLCδ1}::splitGFP11 recognizes PM PI(4,5)P₂]. In addition, this approach may also force the formation of ectopic ER-PM contacts that may not reflect endogenous distribution of these contacts.”

Although split GFP-labelled ER-PM contacts (Fig 1) and endogenously-tagged ESYT-2 (Fig 2) do not perfectly co-localize, overall distribution of these two markers is similar (both endogenous ESYT-2 and the split GFP labelled contacts are enriched in dendrites and scarce in axons), supporting the complementarity of these two approaches.

Can the authors explain the imperfect colocalisation between the ER-PM split-GFP construct

and ESYT-2 (Fig 2E), in particular areas that are ESYT-2+ve GFP-ve?

The split GFP approach may not be able to label all the ER-PM contacts. It is known that a tether for a particular contact site is not evenly distributed among different contact sites of the same kind. This is also the case for ER-PM tethers (e.g., PMID: 29684786; PMID: 28301744).

Reviewer 3 also stated in his comments “It is now well documented that ER-PM MCS are heterogeneous, and that different subtypes with different compositions exists (for example 10.7554/eLife.31019)”. Thus, we do not expect perfect co-localization between ESYT-2 and ER-PM contacts that are labelled by the split GFP approach.

We also discussed this point further in the discussion by adding the following new sentences in the page 16 of our revised manuscript.

New sentences: “We acknowledge the heterogeneity of ER-PM contacts. Due to the difficulty in determining precise ultrastructure of the ER by electron microscopy in *C. elegans* neurons, we used fluorescence-based detection of neuronal ER-PM contacts (i.e., the split GFP approach and ESYT-2). However, both the split GFP approach and ESYT-2 do not label all ER-PM contacts. Further studies are needed to examine whether the proteins that we investigated in the current study have general roles in maintaining the distribution of all types of ER-PM contacts.”

In support of the point about the RET-1 mutant, it has much stronger effects on ER distribution than INPP5K (Fig 6B-H). Only for TRAPbeta are they similar - is there a possible explanation for that different between markers?

One possible explanation is that RET-1 is more essential in maintaining ER morphology compared to CIL-1.

Conclusion about Fig 6 (p11) should include not just that ER sheets expanded and invaded into the dorsal axon but also the dendrites.

Thank you for this suggestion. We included this information in the relevant sentence as shown below (page 13 of our revised manuscript).

Old sentence: “Collectively, these results show that ER sheets expanded and invaded into the dorsal axon of DA9 neuron in *cil-1(yas37)*, *ret-1(tm390)*, and *atln-1(yas38)* mutants.”

New sentence: “Collectively, these results show that ER sheets expanded and invaded into the dorsal axon and became more abundant in dendrite of DA9 neuron in *cil-1(yas37)*, *ret-1(tm390)*, and *atln-1(yas38)* mutants.”

Reviewer #2 (Significance (Required)):

****Expertise:****

mainly in membrane contact site biology with knowledge and experience of a variety of proteins that are ER-PM tethers, and to a lesser extent in ER architecture. I felt I had sufficient expertise to evaluate the results, except I have no experience of using this model system, for which I rely on the clear explanation provided by the authors.

****Audience:****

the manuscript is aiming to bring cutting edge cell biology into the field of developmental neurobiology.

****Referees cross-commenting****

Rev 2: I note that we all have different takes on this and that I am the most skeptical. I would welcome a debate on the points raised. To start, I'll extract just one comment that in many ways summarises my review (apologies that hit's quite long): "It is not clear what drives the authors to only focus on ER-PM contact."

We focused on ER-PM contacts in this manuscript because we isolated mutants based on defects in the distribution of ER-PM contacts.

As molecular mechanisms that maintain the non-uniform distribution of neuronal ER-PM contacts were unknown, we embarked on a forward genetic screen to identify genes that are important for maintaining the unique distribution of neuronal ER-PM contacts using *C. elegans* as our model system. Through a fluorescence microscopy-based genome-wide forward genetic screen, we isolated 2 mutants with altered distribution of ESYT-2 (a major ER-PM tether in metazoan). Balancer mapping combined with next generation sequencing identified that CIL-1 and ATLN-1 (INPP5K homolog and Atlantin-1 homolog, respectively; both are proteins implicated in the maintenance of proper ER morphology) are causative genes of the phenotype. Thus, we studied how the absence of these proteins resulted in altered distribution of neuronal ER-PM contacts. Our genetic and cell biological analysis suggest that the distribution of neuronal ER-PM contacts is maintained by the balance between ER tubules and ER sheets, which are maintained by CIL-1 and ATLN-1.

I agree with the the other reviewers about nature of research, and I agree that all of the experiments here are well done.

We thank the reviewer for the positive comments on our work.

I am happy about Reviewer 1's point about quantification of ESYT-2, although I did not see it myself. This paper could be published as is if the claims were toned down to coldly describe what was found with all the necessary caveats. My feeling is that this will not happen as the caveats are quite broad.

The work here appears to me to be presented in a way that achieves a more promising impression than is merited. It is framed to unduly point to ER-PM contacts as the issue at hand. From the title onwards, the text focuses on this one readout without setting in the round the whole set of results.

As we stated above, we performed a "forward genetic screen" in order to isolate *C. elegans* mutants with altered distribution of neuronal ER-PM contacts, with the aim of identifying genes that contribute to the maintenance of the unique distribution of these contacts. Thus, we did not frame the story as this reviewer mentioned in his/her statement (highlighted in yellow above). We would like to reiterate that we found those genes (i.e., CIL-1/INPP5K and ATLN-1/Atlantin-1) from the above-mentioned un-biased approach. That is the reason why we studied CIL-1/INPP5K and ATLN-1/Atlantin-1 in the context of ER-PM contacts (and more specifically, how these two proteins are involved in the regulation of the distribution of neuronal ER-PM contacts).

We would like to quote the reviewer 1's comments from "Referees cross-commenting". Reviewer 1 stated "To follow-up on Rev#2 comment: they focused on

ER-PM contact sites because that is the essence of the split-GFP assay they used for their screen.”

The abstract implies from sentence two that the work might address mechanisms that regulate the distribution of neuronal ER-PM contacts, and yet it does not specifically address the regulation of this aspect of ER biology. Although (as Rev#1 stated in 1st comment) this was the primary read-out, it turned out to be correlated with changes in bulk ER distribution, i.e. the ER is different in many respects. The justification is missing to support honing in on this one facet.

I think there is a misunderstanding regarding “ER distribution” (relevant sentence is highlighted in yellow above). We showed in Fig S1 that the ER itself is present uniformly throughout neuronal processes of DA9 motor neuron of wild-type animals (both axon and dendrite). In Fig 6, what we showed was “rough ER proteins”. We found that rough ER proteins (visualized by endogenously-tagged SEC-61 β and overexpressed TRAP β ::mNeonGreen) were almost non-existent in axons compared to dendrites in wild-type *C. elegans* neurons (Fig 6F and 6I). In *cil-1*, *atln-1* and *ret-1* mutants, we observed significant presence of these proteins in axons compared to wild-type animals, suggesting that the axonal ER in these mutants acquired the identity that is more similar to ER sheets. To be clear, we did not show that the distribution of “ER itself” changed in these mutants but showed that the distribution of some “rough ER proteins” changed in these mutants (please also see our responses above in page 9-10 of this letter).

The penultimate sentence of the Abstract also implies causation: "In mutants of CIL-1 or ATL-1, ER sheets expand and invade into the axon, causing the ectopic formation of ER-PM contacts and defects in axon regeneration following laser-induced axotomy." but there is no experimental evidence that the ER sheets expanding and invading caused the ectopic contacts. Association does not equal causation.

As we stated above, we interpreted the ectopic presence of “rough ER proteins” in axons of *cil-1* and *atln-1* mutants as invasion of ER sheets into the axons. Yeast mutants with expansion of ER sheets are accompanied by increased formation of ER-PM contacts (e.g., PMID: 21502358), suggesting that ER sheet expansion may have led to ectopic formation of ER-PM contacts also in our model system. However, we agree with this reviewer that our studies do not conclusively establish the causative role of the expansion of ER sheets in the ectopic formation of neuronal ER-PM contacts. Thus, we toned down this statement in the abstract as shown below (page 3 of our revised manuscript).

Old sentences: “In mutants of CIL-1 or ATL-1, ER sheets expand and invade into the axon, causing the ectopic formation of ER-PM contacts and defects in axon regeneration following laser-induced axotomy. As INPP5K and Atlastin-1 have been linked to neurological disorders, the unique distribution of neuronal ER-PM contacts regulated by these proteins may support neuronal resilience during the onset and progression of these diseases.”

New sentences: “In mutants of CIL-1 or ATL-1, ER sheets expand and invade into the axon. This is accompanied by the ectopic formation of axonal ER-PM contacts and defects in axon regeneration following laser-induced axotomy. As INPP5K and Atlastin-1 have been linked to neurological disorders, the unique distribution of neuronal ER-PM contacts maintained by these proteins may support neuronal resilience during the onset and progression of these diseases.”

We would like to quote the reviewer 1' comments from "Referees cross-commenting". Reviewer 1 stated "I think Rev#2 criticisms mainly relate to the fact that the effects on ER-PM could be indirect given what is known on Atlastin and INPP5K. I would agree with that point but rather see it as one requesting to be discussed. I still think the data are solid."

*My point about Ret-1 is not that it is completely invalid, but largely so because the literature shows that reticulons (say in humans) have many functions (e.g. paper on human RTN3). The current manuscript suggests the possibility that every one of these functions is missing in the ret-1 mutant worm. **While ER shaping is most obvious RTN function**, we now know that much more is going on in this gene family in humans, and to my view this is likely the case in worms, given their 7 highly diverse isoforms. This would strongly caveat the result. Of course the authors would have the chance to rebut/revise this by explaining which RTN functions persist in their strain.*

We agree with the reviewer that the most obvious function of RTN family proteins is ER shaping (**highlighted in yellow above**). Our *ret-1* mutants carry 255bp deletion in *ret-1* gene, removing the last three common exons of all the seven isoforms of RET-1. Thus, our *ret-1* mutants most likely do not express any functional RET-1 proteins (i.e., null allele). Accordingly, we observed major changes in ER shape in the cell body of DA9 neuron in our *ret-1* null mutants (Fig 6A).

As we stated above (page 8-9 of this letter), we agree with the reviewer 3's statement on the validity of our experiments. Reviewer 3 stated, "For me, I disagree for example to call the RET-1 experiment invalid, because the experiment was done in a correct way. We can conclude there is a correlation between RET-1 ablation and ER-PM contact differences even if we don't understand the phenotype entirely, meaning why this is so, but that it does happen I believe is clearly shown."

We performed textual changes to tone down our statements in the original manuscript regarding the interpretation of these results as stated above (page 8-9 of this letter). Studying the contribution of individual RET-1 isoform in ER morphology (or other aspects of RET-1 function) is beyond the scope of this current study.

Reviewer #3 (Evidence, reproducibility and clarity (Required)):

****Summary:****

Membrane contact sites between the endoplasmic reticulum and the plasma membrane (ER-PM MCS) are structures that serve as intracellular hubs for signaling and non-vesicular lipid exchange that have been implicated in neuronal survival and growth. Although these structures have been described for several decades, it is only in the past decade or so that their molecular composition, function and importance have been increasingly recognized and described. However many aspects of their biology remain unknown. In present study the authors utilize the overexpression pattern of the ER-PM tether ESYT-2 to perform an unbiased screen for regulators of these structures in the C. elegans D9 neuron. They observe that distal axons are depleted in ER-PM MCS as compared to the soma and dendrites. Furthermore, they identify the phosphoinositol 5-phosphatase INPP5K (CIL-1 in C. elegans) and the dynamin-like ER-shaping protein Atlastin-1 (ATLN1) as negative regulators of the abundance of axonal ER-PM MCS. Furthermore they show that INPP5K acts genetically upstream of ATLN1 as ATLN1 expression rescues the INPP5K, but not vice-versa. Finally, they show that both INPP5K and ATLN1 contribute to axonal regeneration.

In general, the manuscript is well-written, the figures are clear and experiments have for the most part proper controls. However, there are several instances where the authors have overinterpreted the results and draw conclusions that are suggested rather than demonstrated definitively. For one set of the experiments (see point 2) controls are lacking and results are largely overinterpreted, and this should really be addressed.

We thank the reviewer for these very positive comments. We also appreciate all the constructive comments from this reviewer to further improve our manuscript.

****Major Comments:****

1) The authors utilize two main markers to visualize ER-PM MCS: one used the split-GFP approach and the other the ER-PM tether ESYT-2. It is important to point out to the reader the limitations of both of these tools. It is now well documented that ER-PM MCS are heterogeneous, and that different subtypes with different compositions exists (for example 10.7554/eLife.31019) Thus ESYT-2 does not label all ER-PM contacts nor does the split-GFP which relies on PI(4,5)P₂-binding tethers. Readers should be made aware that the authors are talking about ESYT-2 and/or PI(4,5)P₂-dependent MCS and that since electron microscopy (normally the gold standard for the study of MCS) is not feasible in this system, the conclusions cannot be generalized to all MCS types. In addition, the split-GFP approach or ESYT overexpression may themselves stabilize or increase the number of MCS as compared to endogenous. This should be pointed out.

We thank the reviewer for these comments. We agree with the heterogeneity of the ER-PM contacts. We also think that neither ESYT-2 nor split GFP system labels all types of ER-PM contacts.

We added the following sentence in the page 7 of our revised manuscript to describe the limitation of the split GFP approach.

New sentences: "The split GFP approach mostly labels ER-PM contacts that depend on the presence of PM PI(4,5)P₂ [as mCherry::3xPH^{PLCδ1}::splitGFP11 recognizes PM PI(4,5)P₂]. In addition, this approach may also force the formation of ectopic ER-PM contacts that may not reflect endogenous distribution of these contacts."

We also changed the following sentence on the page 8 in our revised manuscript to describe the heterogeneity of the ER-PM contacts.

Old sentence: “wrmScarlet::ESYT-2 co-localized extensively with GFP puncta, confirming that dendritic ESYT-2 localizes primarily to ER-PM contacts (**Figures 2E and 2F**).”

New sentences: “Although wrmScarlet::ESYT-2 did not co-localize perfectly with GFP puncta (i.e., split GFP-labelled ER-PM contact sites), the majority of wrmScarlet::ESYT-2 co-localized with GFP puncta, confirming that dendritic ESYT-2 localizes primarily to ER-PM contacts (**Fig 2E and 2F**). These results are consistent with the heterogeneity of the ER-PM contacts as shown in recent studies (Besprozvannaya et al., 2018; Saheki and De Camilli, 2017a)”

We also inserted the following sentences in the discussion (page 16 of our revised manuscript).

New sentences: “We acknowledge the heterogeneity of ER-PM contacts. Due to the difficulty in determining precise ultrastructure of the ER by electron microscopy in *C. elegans* neurons, we used fluorescence-based detection of neuronal ER-PM contacts (i.e., the split GFP approach and ESYT-2). However, both the split GFP approach and ESYT-2 do not label all ER-PM contacts. Further studies are needed to examine whether the proteins that we investigated in the current study have general roles in maintaining the distribution of all types of ER-PM contacts.”

2) The use of the INPP5K phosphatase domain coupled to either the PLC-PH domain or to the ER protein CP450 is quite interesting but these experiments have some major caveats, and I do not necessarily agree with the interpretation of the authors:

a) In general the substrates of INPP5K which are thought to be PI(4,5)P₂ and PI(3,4,5)P₃ are both thought to exist primarily on the plasma membrane (in some cases in other organelles), and they are not generally thought to be substantially present in the ER although some reports do suggest a small pool for ER PI(4,5)P₂ (like Watt Biochem J 2002) this is far from generally well accepted. INPP5K can also dephosphorylate soluble inositol and so influence InsP3R activity as well as the activity of TMEM16 family members which have also been implicated in MCS. So which substrate is important has not been shown and cannot be assumed to be PI(4,5)P₂ without more study. These issues are discussed a little in the Di Camilli paper Dong 2018, and need to be discussed here.

We agree with this reviewer that our studies do not completely address which substrate is regulated by INPP5K. We added the following sentences in the discussion to discuss these points.

New sentences: “There is no strong evidence that supports the presence of PI(4,5)P₂ in ER membranes. It is possible that the presence of CIL-1/INPP5K is the exact reason why PI(4,5)P₂ is not abundant in ER membranes; removal of PI(4,5)P₂ from ER membranes by CIL-1-dependent PI(4,5)P₂ dephosphorylation may be essential for ATLN-1 to regulate the balance of ER tubules and sheets. INPP5K was also reported to hydrolyze PI(3,4,5)P₃, a phosphoinositide that is normally produced by phosphoinositide 3-kinase (PI3K) at the PM (Dong et al., 2018; Gurung et al., 2003; Ijuin et al., 2000; Schmid et al., 2004). Thus, whether PI(4,5)P₂ is the major substrate of CIL-1 deserves further investigation.” (page 16 of our revised manuscript)

b) The authors claim that specifically the ER-localized PI(4,5)P₂ is the important substrate for the INPP5K but do not provide any data showing that PI(4,5)P₂ levels (on the PM or ER)

are altered in the absence of INPP5K or that these levels are rescued in the intended surfaces by their artificial constructs. There is no data to show that either the ER-tethered or PM-tethered phosphatase domains are even actually active and capable of dephosphorylation.

We thank the reviewer for these comments. We expected to see some changes in the distribution of PI(4,5)P₂ in *cil-1* mutants, but we failed to detect such changes using the PH domain of PLC δ 1 (PH^{PLC δ 1}) as a PI(4,5)P₂ marker. This is similar to what has been reported for HeLa cells lacking INPP5K (no particular changes in the distribution of PI(4,5)P₂ were detected using the same biosensor in these cells) (PMID: 30087126). It is possible that PH^{PLC δ 1} (or any other currently available PI(4,5)P₂ biosensors) may not be suitable for detecting subtle changes in the levels of PI(4,5)P₂ on ER membranes (e.g., reviewed in PMID: 25732852). We now state in the discussion "...Thus, whether PI(4,5)P₂ is the major substrate of CIL-1 deserves further investigation." (page 16 of our revised manuscript)

This is also the reason why we attempted to modulate the levels of PI(4,5)P₂ by targeting the 5-phosphatase domains of other well-studied proteins, namely OCRL-1 and UNC-26 (orthologues of human OCRL and synaptojanin-1, respectively), to the ER or to the PM. We now show in the revised manuscript that PM-targeted 5-phosphatase domains of OCRL-1 and UNC-26 can efficiently dephosphorylate PM PI(4,5)P₂ in HeLa cells, as assessed by iRFP-tagged with the PH domain of PLC δ 1 (iRFP-PH PLC δ 1) (**new Fig S4D**). We also show that ER-targeted versions of the same 5-phosphatase domains had no effects on PM PI(4,5)P₂ (**new Fig S4E**). These results suggest that the chimeric 5-phosphatase domains we used in our study are capable of removing PI(4,5)P₂ *in cis* on membranes without affecting PI(4,5)P₂ levels of other cellular compartments.

[During the revision, we found that targeting the 5-phosphatase domains of OCRL-1 and UNC-26 to the PM using PH^{PLC δ 1} as the PM-targeting motif (our original strategy) made these 5-phosphatase domains inactive [please see below the point 2-d]. Thus, we generated new versions of PM-targeted 5-phosphatase domains by using the PM-targeting motif of Lck (bioRxiv 160374 10.1101/160374) and validated their activities in HeLa cells. We replaced the old data, where we used PM-targeted 5-phosphatase domains, with the new data with our new PM targeting strategy throughout the manuscript.]

c) Moreover, assuming they are indeed active, it is unclear whether the phosphatase domains are capable of acting in cis (ie the same membrane it is tethered to), in trans (ie at an ER-PM or an ER-organelle contact site) or both. Thus, the ER-tethered phosphatase domain might actually dephosphorylate substrates on different membranes, and this needs to be assessed. The same argument applies to the OCRL and synaptojanin-1 phosphatases. It could be done in HEK cells for example if it is too difficult to do in C-elegans directly

We thank the reviewer for these suggestions. As we mentioned above, we used HeLa cells to confirm the activities of the chimeric 5-phosphatase domains. While expression of either PM-targeted OCRL-1 5-phosphatase domain (newly generated mScarlet-I-OCRL-1^{PM}) or PM-targeted UNC-26 5-phosphatase domain (newly generated mScarlet-I-UNC-26^{PM}) strongly reduced the levels of PM PI(4,5)P₂, as assessed by iRFP-PH PLC δ 1 (**new Fig S4D**), expression of ER-targeted versions of the same 5-phosphatase domains (OCRL-1^{ER}-EGFP and UNC-26^{ER}-EGFP) had no effects on PM PI(4,5)P₂ (**new Fig S4E**). These results suggest that the chimeric 5-phosphatase domains are capable of removing PI(4,5)P₂ *in cis* on membranes without affecting PI(4,5)P₂ levels of other cellular compartments.

PM-targeted CIL-1 5-phosphatase domain was not expressed at sufficiently high levels in HeLa cells, making it difficult to assess the activities of the 5-phosphatase domain of CIL-1 in cells. We added more extensive discussion on the preference/selectivity of the 5-phosphatase of INPP5K (CIL-1 homolog in mammals) in the page 16 of our revised manuscript based on literatures.

d) Finally, for the phosphatase domain tethered to the PM using the PH domain seems counterproductive because presumably the same lipid that is used to target the phosphatase domain would be cleaved by the phosphatase releasing the construct from the PM. So it seems premature to say that PM PI(4,5)P₂ is not important. At a minimum, a control showing that this construct actually decreases PM PI(4,5)P₂ would strengthen this claim.

We appreciate the reviewer for these comments. As we stated above, we could not detect efficient depletion of PM PI(4,5)P₂ by targeting the 5-phosphatase domains of OCRL-1 and UNC-26 to the PM using our original PM targeting strategy (using the PH^{PLCδ1} as the PM-targeting motif). Thus, we generated new versions of PM-targeted 5-phosphatase domains by using the PM-targeting motif of Lck and confirmed that they can induce efficient depletion of PM PI(4,5)P₂ in HeLa cells. We found that targeting the active 5-phosphatase domain of either UNC-26 or OCRL-1 to the PM (using the new strategy) did not restore the distribution of mNeonGreen::ESYT-2 in DA9 neuron in *cil-1* mutants (while the ER-targeted 5-phosphatase domain of either OCRL-1 or UNC-26 restored it to the similar levels to control animals) (**updated Fig 4D and 4E**).

We also toned down our statements as shown below:

Old sentences: "These results suggest that CIL-1 functions at the ER to modulate PI(4,5)P₂ levels of this organelle to maintain ESYT-2 distribution."

New sentences: "These results suggest that CIL-1 may function at the ER to modulate PI(4,5)P₂ levels of this organelle to maintain non-uniform distribution of ESYT-2 in neurons (see discussion), although the possibility that PM PI(4,5)P₂ plays some roles in regulating ESYT-2 distribution in neurons cannot be excluded." (page 10 of our revised manuscript)

3) It is a bit surprising to find that increased contact sites hinders axonal growth, because the opposite has been suggested in other studies (ie Petkovik et al NCB 2015). The authors clearly establish a correlation between decreased axonal regeneration capability and the ectopic presence of ER-PM contact sites, however causality is not well established - it is not clear that abnormal ER-PM contacts are hindering regeneration they may be simply be another bystander consequence of a separate problem. Does increasing contact sites with the split GFP overexpression also hinder axonal regeneration, or does detaching contact sites with the ER-tethered phosphatases improve it? It may be for example that the important parameter here is rather dynamics: tubules are dynamic structures, linked to microtubules. Indeed, the authors show that ESYT-2 is dynamic in the axon, and perhaps the ESYT-2 in the axon is not present in contact sites but rather on untethered ER. perhaps if nascent ER tubules lose their dynamics then slower reactions like ribosome docking and contact site formation can then occur spontaneously, which would explain the increase in rough ER and contact sites. Alternative explanations should at the very minimum be considered.

We appreciate the reviewer for these constructive comments.

We believe that proper compositions of ER-PM contacts rather than their abundance (i.e., quality vs. quantity) are important for regeneration or outgrowth of axons. Regarding the overexpression of the split GFP, no matter how much we overexpressed the split GFP in DA9 neurons, we could not alter the non-uniform distribution of neuronal ER-PM contacts. We did not succeed in establishing a line that stably expressed high levels of the split GFP system due to toxicity, for axotomy experiments. In our original manuscript, we performed overexpression of ATLN-1 in *cil-1* mutants to reduce ER-PM contacts in axons and observed recovery of axon regeneration defects (Fig 7F and 7G).

We agree with this reviewer that other aspects of axonal ER, including its dynamics, may also play an important role in axon regeneration. We added the following sentences in discussion to discuss other alternative explanations (page 17 of our revised manuscript).

New sentences: "Mutants lacking CIL-1 or ATLN-1 show ectopic formation of axonal ER-PM contacts. Petkovic et al., suggested that lipid delivery at neuronal ER-PM contacts promote neurite outgrowth (Petkovic et al., 2014). Thus, proper compositions of ER-PM contacts rather than their abundance maybe important for regeneration or outgrowth of axons. Our study does not rule out the possibility that the axonal ER morphology (i.e., tubular vs. sheet) and motility as well as other related functions of the axonal ER, such as maintenance of Ca²⁺ homeostasis and regulation of the functions of various other organelle contact sites, may also play a role in proper axon regeneration."

****Minor comments:****

4) In Figure 2: is the percent co-localization with the PM marker the same between ventral and dorsal areas? The mobile sites maybe ER-organelle contacts rather than ER-PM contacts.

It was very difficult to establish a stable *C. elegans* line that expressed both ESYT-2 and split GFP constructs (out of 60 injections, only 1 line was established) (perhaps due to toxicity); therefore, it has been very difficult to quantify their co-localization accurately in both ventral and dorsal processes for comparison. We added a sentence, indicating the possibility that the mobile ESYT-2 puncta may indicate its presence at other ER-organelle contacts (page 8 of our revised manuscript).

Old sentence: "These results suggest that dendritic ESYT-2 localizes to cortical ER and stably associates with ER-PM contacts."

New sentence: "These results suggest that dendritic ESYT-2 localizes to cortical ER and stably associates with ER-PM contacts, while the mobile ESYT-2 possibly indicates minor fractions of ESYT-2 that participate in membrane contact sites formed between the ER and other organelles."

5) In Figure 4: It is unclear how the authors distinguish between cortical ER sheets and ER sheets close to the membrane but not forming contacts (ie more than 30 nm from the PM)

We found that *C. elegans* ESYT-2 localize primarily at ER-PM contacts (similar to mammalian E-Syt2 and E-Syt3). Thus, we used ESYT-2 to label cortical ER sheets that form contact with the PM.

6) General comment about the figures: the x-axis labels in many of the figures can be quite cumbersome to read with very long genetic names. To increase clarity and readability the

authors should consider grouping some of the genotypes to reduce the size of some of these labels.

We modified the description of genotypes in relevant figures as much as possible to reduce the size of some labels to increase the clarity and readability.

Reviewer #3 (Significance (Required)):

Membrane contact sites (MCS) are structures that have been under intense investigation in the past decade and essentially a new branch in cell biology has emerged to study the composition and function of these structure. The paper identifies two proteins shown previously to participate in shaping the ER to contribute to the balance of PM MCS abundance, which is a novel role, though it does not provide the mechanisms through which these proteins achieve this function. The finding that contact sites are depleted in axons and inhibit axonal regeneration are a bit surprising and somewhat contradictory to other studies such as Petkovic et al NCB 2015 which suggested lipid delivery at contact sites promote rather than inhibit neurite outgrowth. Of course in the present study the context could be different, but it probably speaks to the fact that quality and not quantity is what counts. Thus I believe the manuscript would be of interest to both researchers in the field of contact site biology as well as scientists in the neural regeneration field and merits publication once the major concerns are addressed as they will surely incite further investigation into the mechanisms through which these proteins acts.

We thank the reviewer for these very nice comments on our work. We agree with this reviewer that the quality rather than the quantity of membrane contact sites may matter for their physiological functions.

My background is in cell biology and I consider myself to have significant expertise on ER and membrane contact site biology. However my experience in neuroscience and with the C. elegans model system is limited.

****Referees cross-commenting****

I agree with Rev#1 that a substantial amount of work of good quality has been performed. The authors show many correlations (INPP5K, ATL, RET-1, ablation with ER-PM contact deregulation; INPP5K, ATL ablation with axonal regrowth defect) which in my opinion are still worthwhile, interesting observations that should be published.

We thank the reviewer for supporting the publication of our manuscript. We have performed extensive textual changes and performed additional experiments to address all the concerns raised by reviewers.

However, I agree with Rev #2 that several of the conclusions of causative links needs to be softened or removed because they do not conclusively show causality. And I wholeheartedly agree that contacts may not be what is important for axonal regeneration and that other defects should be considered. But the paper could be rewritten to say this correlates with that rather than causes it, and then it would be scientifically correct - and to me even if the effect of ATL and INPP5K on contacts is indirect it is still interesting to know this, and would be interesting to eventually figure out why.

We agree with this reviewer that the conclusions of causative links should have been softened. We revised our texts throughout the manuscript and indicated all the changes with tracked changes in the revised manuscript.

Perhaps where REv2 and I differ are in what we consider enough for a complete paper - Is it acceptable to publish without mechanism or do you need mechanism to publish? Could partial mechanism be enough? Some of the experiments I suggest to strengthen the use of their targeted phosphatase constructs for example plus some of the expt Rev#2 suggested like overexpression of CLIMP, which I think is clever, would provide partial mechanism or at least further support the hypothesis. Would this be enough?

We appreciate these constructive comments. Due to limitation of available techniques in *C. elegans*, molecular mechanisms that link the functions of ATLN-1 and CIL-1 to the distribution of neuronal ER-PM contacts are not fully elucidated in the current study. We performed additional experiments outlined above in the letter to better support our hypothesis/conclusion.

Rev#2 points out 3 interesting questions the data raise that are unanswered. I agree these are the interesting questions the data raise. But if it would take another year or two to answer, perhaps it would be worth it to publish the data they have now, just adjusting the wording to admit to the questions unanswered.

We thank the reviewer for these supportive comments. We agree with this reviewer that it would take years to address the three questions that were raised by the reviewer 2. Although these questions are interesting, we think addressing them is beyond the scope of our current manuscript.

For me, I disagree for example to call the RET-1 experiment invalid, because the experiment was done in a correct way. We can conclude there is a correlation between RET-1 ablation and ER-PM contact differences even if we don't understand the phenotype entirely, meaning why this is so, but that it does happen I believe is clearly shown. What is invalid is the conclusion that this proves that ER shape regulates ER-PM contact distribution - with this I agree. The data support this hypothesis, but do not prove it. But to me it doesn't mean the data don't deserve to be out there, published.

We appreciate these supportive comments. We agree with this reviewer's statement on the validity of our experiments and believe that there is a correlation between ER shape and ER-PM contact distribution based on our results. We performed textual changes to tone down our statements regarding the interpretation of the results from the RET-1 experiments (pages 11-12 of our revised manuscript).

Old sentence: "Our results suggested that cortical ER shape is essential for maintaining the distribution of neuronal ER-PM contacts."

New sentence: "Our results suggested that cortical ER shape is potentially important for maintaining the distribution of neuronal ER-PM contacts." (page 11)

Old sentence: "Thus, cortical ER shape, and more specifically the proper balance of ER tubules and sheets, is essential for restricting the distribution of neuronal ER-PM contacts to specific regions within neurons (namely the somatodendritic region)."

New sentence: "Taken together, these data support the notion that cortical ER shape, and more specifically the proper balance of ER tubules and sheets, play a role in restricting the distribution of neuronal ER-PM contacts to specific regions within neurons (namely the somatodendritic region)" (page 12)

They did do good work for the most part, they are just a bit overzealous with the conclusions.

Thank you very much for all the helpful feedback and suggestions. We toned down our statements throughout our manuscript, so that they would not overstate the causality linkages.

September 2, 2021

RE: Life Science Alliance Manuscript #LSA-2021-01092-TR

Dr. Yasunori Saheki
Nanyang Technological University, Singapore
Lee Kong Chian School of Medicine
11 Mandalay Road
Singapore 308232

Dear Dr. Saheki,

Thank you for submitting your revised manuscript entitled "INPP5K and Atlastin-1 maintain the nonuniform distribution of ER-plasma membrane contacts in neurons". We would be happy to publish your paper in Life Science Alliance pending final revisions necessary to meet our formatting guidelines.

- please upload your main manuscript text as an editable doc file
- please upload your Tables in editable .doc or excel format
- please note that titles in the system and manuscript file must match
- please rename the "DECLARATION OF INTERESTS" section to "Conflict of interest"
- please remove the SIGNIFICANCE STATEMENT section on page 4
- please add your main, supplementary figure, and table legends to the main manuscript text after the references section
- please consult our manuscript preparation guidelines <https://www.life-science-alliance.org/manuscript-prep> and make sure your manuscript sections are in the correct order

LSA now encourages authors to provide a 30-60 second video where the study is briefly explained. We will use these videos on social media to promote the published paper and the presenting author. Corresponding or first-authors are welcome to submit the video. Please submit only one video per manuscript. The video can be emailed to contact@life-science-alliance.org

A. FINAL FILES:

B. MANUSCRIPT ORGANIZATION AND FORMATTING:

Sincerely,

Eric Sawey, PhD
Executive Editor

Reviewer #1 (Comments to the Authors (Required)):

The authors have very satisfactorily answered the reviewers' requests. This is an excellent article that should be published as it is.

September 7, 2021

RE: Life Science Alliance Manuscript #LSA-2021-01092-TRR

Dr. Yasunori Saheki
Nanyang Technological University
Lee Kong Chian School of Medicine
11 Mandalay Road
Singapore 308232
Singapore

Dear Dr. Saheki,

Thank you for submitting your Research Article entitled "INPP5K and Atlastin-1 maintain the nonuniform distribution of ER-plasma membrane contacts in neurons". It is a pleasure to let you know that your manuscript is now accepted for publication in Life Science Alliance. Congratulations on this interesting work.

DISTRIBUTION OF MATERIALS:

Again, congratulations on a very nice paper. I hope you found the review process to be constructive and are pleased with how the manuscript was handled editorially. We look forward to future exciting submissions from your lab.

Sincerely,
